

# Impact of stray light on greenhouse gas concentration retrievals and emission estimates as observed with the passive airborne remote sensing imager MAMAP2D-Light

Oke Huhs[1], Jakob Borchardt[1], Sven Krautwurst[1], Konstantin Gerilowski[1], Heinrich Bovensmann[1], Hartmut Bösch[1], and John P. Burrows[1]

[1]University of Bremen, Institute of Environmental Physics, Otto-Hahn-Allee 1, 28359 Bremen

**Correspondence:** Oke Huhs (oke.huhs@iup.physik.uni-bremen.de)

**Abstract.** MAMAP2D-Light is an airborne passive remote sensing push-broom spectrometer developed at the Institute for Environmental Physics at the University of Bremen to measure atmospheric methane ($CH_4$) and carbon dioxide ($CO_2$) column anomalies to quantify point-source emissions in the $1.6\,\mu m$-band. In its initial version, as flown in 2022, a significant stray light level of $4\,\%$ of the measured signal has been observed, causing apparent error patterns in the retrieved $CO_2$ and $CH_4$

column anomalies. In this paper, we report the successful application of a stray light correction developed for the instrument. Measurement data collected during an airborne campaign in 2022 in Canada offer the unique opportunity to investigate the end-to-end impact of stray light and its correction on the retrieved $CO_2$ and $CH_4$ column anomalies, as well as the retrieved emission rates. Stray light caused apparent error patterns in the retrieved column anomaly maps. In nearly all cases, applying the $CH_4/CO_2$ proxy method reduced the stray-light-related column errors below the column noise, leading to comparable

final emission rate estimates for proxy-only and stray-light-corrected data. In this paper, we additionally investigate the special scene contrast conditions under which the correction by applying the proxy method is no longer sufficient. Following the initial campaign in 2022, the stray light was reduced by $\sim 75\,\%$ by the implementation of a hardware modification from 2023 onward.

## 1 Introduction

Passive remote sensing has become one of the cornerstones for monitoring the most critical greenhouse gases (GHGs), carbon

dioxide ($CO_2$) and methane ($CH_4$), in the Earth's atmosphere to determine anthropogenic and natural GHG emissions. The spectral absorption features of the GHGs in reflected sunlight are exploited to retrieve the corresponding atmospheric GHG concentrations. However, depending on the instrument's spatial and spectral resolution, the distance from the source, and the source area, surface emissions introduce only minor changes in the measured absorption features compared to the absorption features due to the accumulated background concentrations in the total atmospheric column. For accurate emission estimates,

therefore, strict instrument-dependent specifications of spatial and spectral calibration accuracy of the measured spectra are required, allowing to retrieve the according atmospheric GHG columns with the required accuracy and precision.

For an instrument with a given spatial and spectral resolution, the required column precision is determined by the detection limit required for the envisaged emission estimates (Jacob et al., 2022; Pandey et al., 2023). For example, the $CH_4$ column



single-measurement precision for SCIAMACHY, the first instrument dedicated to remote sensing of GHGs from space aboard the EnVISAT satellite, was planned to achieve $1\,\%$ (Bovensmann et al., 1999). For its successor, TROPOMI (TROPOspheric Monitoring Instrument), the goal precision was tightened to $0.6\,\%$ for a single measurement (Veefkind et al., 2012). To achieve this precision, high-quality instrument characterizations, minimizing and correcting radiometric errors, are required.

A significant contributor to the radiometric error is stray light, which arises from reflections and scattering processes that are not intended in the optical design. The definition and terminology of stray light are adapted from Fest (2013). Stray light distorts the measured spectra with a continuum-dependent error (Tol et al., 2018) and is most prominent in high-contrast scenes, e.g., in mixed scenes with dark land surfaces and bright clouds. However, the stray-light-induced error signal depends on the overall intensity distribution of the light paths within the system. The spectrally, spatially, and intensity-dependent error signal introduces error patterns in the retrieved concentrations and can be misinterpreted as column enhancements and, in certain cases, even as plumes. Therefore, it is essential to mitigate stray light within the optical system. Effective mitigation of stray light involves minimizing it through an optimized optical design and correcting it during data processing. The latter uses so-called stray light kernels estimated from stray light characterization measurements.

Various methods for stray light correction have been developed. A widely used approach for non-imaging spectrometers, described by Zong et al. (2006), exploits matrix operations to correct both spectrally and spatially stable and variant stray light simultaneously. This method requires dense characterization measurements or interpolations across the entire focal plane array. For imaging spectrometers, the presence of an additional spatial axis increases the size of the correction matrix by the squared number of spatial pixels and therefore the required computational resources massively (Zong et al., 2007). For the SWIR channel of TROPOMI, Tol et al. (2018) introduced a tailored method that separates stable and variant stray light components. The stable component is corrected using an iterative deconvolution method, which has also been applied for the MethaneAIR instrument (Staebell et al., 2021). The variant component is addressed through a spatial transformation based on the variability of the stray light depending on the origin. This method leverages the fast Fourier transform (FFT) for deconvolution, enabling a near-real-time stray light correction.

This work focuses on spectra measured with MAMAP2D-Light (Methane Airborne MAPper 2D Light), a lightweight airborne remote sensing push-broom imaging grating spectrometer built at the Institute for Environmental Physics (IUP) Bremen. Besides satellite-based instruments, airborne remote sensing spectrometers offer spatially high-resolution measurements with similar spectral specifications. This allows intercomparison between both measurement platforms, airborne and spaceborne, for the used GHG concentration retrieval algorithms and the final emission rate retrievals. In contrast to satellite-based instruments, airborne spectrometers can be recalibrated and improved during their lifetime.

MAMAP2D-Light is buildung on concepts established with the MAMAP (Methane Airborne MAPper) instrument (Gerilowski et al., 2011) and is designed to measure $CH_4$ and $CO_2$ column anomalies in the $1.6\,\mu m$ band exploiting the $CO_2$ (Krings et al., 2011) or the $CH_4$ (Krings et al., 2013) proxy method. The data set used in this study was collected during the CoMet 2.0 Arctic mission, which took place in summer 2022 in Canada[1]. The GHG concentrations were retrieved using the WFM-DOAS (Weighting Function Modified Differential Absorption Spectroscopy) method Krings et al. (2011), which has been proven to

---

[1]https://comet2arctic.de/ last access: 11.02.2025



deliver reliable $CH_4$ and $CO_2$ column anomalies on the local scale in the past (Krings et al., 2011, 2013; Krautwurst et al., 2017, 2021, 2024). A post-campaign stray light characterization with a Littman/Metcalf tunable laser (Stry et al., 2006) revealed a significant stray light contamination in the campaign dataset. Therefore, a post-flight stray light correction was applied, exploiting the previously performed characterization measurements. The stray-light-contaminated campaign data provides a unique opportunity to investigate the impact of the stray light correction on real measured data and the capabilities of the proxy method in the presence of stray light in the entire processing chain from the measured spectra to the retrieved GHG concentrations and the emission rate estimates.

In Sect. 2, the instrument design of MAMAP2D-Light is introduced. The impact of stray light on the used WFM-DOAS retrieval is described in Sect. 3. The stray light characterization measurements of MAMAP2D-Light are summarized in Sect. 4. From the characterization measurements, a stray light correction algorithm is applied in Sect. 5. The correction is applied to real measured and simulated spectra, and the impact on the retrieved concentration maps is analyzed in Sect. 6. From the concentration maps, the resulting $CH_4$ emission rates of two measured landfill plumes are analyzed based on the impact of the applied stray light and the proxy correction in Sect. 7. With the post-flight stray light characterization measurements, the origin of the majority of the stray light has been localized and mitigated by a hardware improvement shown in Sect. 8.

## 2 MAMAP2D-Light instrument

MAMAP2D-Light is an airborne passive remote sensing instrument for observing atmospheric $CO_2$ and $CH_4$ columns using infrared spectroscopy. The MAMAP2D-Light instrument, shown in Fig. 1, is a push broom imaging spectrometer with a planar reflective grating. It weighs approximately $43\,\mathrm{kg}$ and fits into an underwing pod of a motor glider aircraft (e.g., Diamond HK 36-TTC ECO, (Borchardt et al., 2025)). MAMAP2D-Light covers the wavelength range from $1559\,\mathrm{nm}$ to $1690\,\mathrm{nm}$ with a spectral resolution of approximately $1.1\,\mathrm{nm}$. It comprises a front optic, an optical fiber bundle (see Fig. 2), an entrance slit, two different lenses, serving as collimator and camera optics, a planar reflection grating and an infrared detector (AIM SWIR384). The detector deployed has a Mercury Cadmium Telluride (MCT) focal plane array (FPA) comprising $384\,\mathrm{pixel} \times 288\,\mathrm{pixel}$ and a pixel pitch of $24\,\mathrm{\mu m} \times 24\,\mathrm{\mu m}$. Within the spectrometer, the FPA is oriented in a way that the spectral axis is along the $384\,\mathrm{pixel}$ axis, which results in a spectral oversampling of $\sim 3\,\mathrm{pixel}$, while the spatial axis is along the $288\,\mathrm{pixel}$ axis. The FPA is cooled to approximately $150\,\mathrm{K}$ with a linear single-piston cooler to reduce the internal thermal dark current of the MCT. The spectral cut-off was adapted from $\sim 2.5\,\mathrm{\mu m}$ to $\sim 1.8\,\mathrm{\mu m}$ by the manufacturer to reduce the sensitivity to thermal background radiation from the optical bench and mounting elements and thereby, allowing to operate the instrument at ambient temperature.

MAMAP2D-light measures scattered sunlight from the Earth's surface, which is imaged via the front optical lens onto an optical fiber bundle with 36 rectangular single fibers stacked in a ferrule, see Fig. 2. Each fiber has a fiber core of $\sim 300\,\mathrm{\mu m} \times 100\,\mathrm{\mu m}$ in spatial and spectral direction, respectively. The outer dimensions of the fibers with cladding are $\sim 315\,\mathrm{\mu m} \times 175\,\mathrm{\mu m}$. Due to the orientation of the detector, only 28 of the 36 fibers are imaged at the detector, resulting in 28 across-track ground scenes observed by the instrument. The entrance slit unit of the spectrometer comprises the ferrule on the fiber bundle's second end, an adjustable slit, an optical order sorting filter and a shutter unit. The light entering the spectrometer is collimated by a





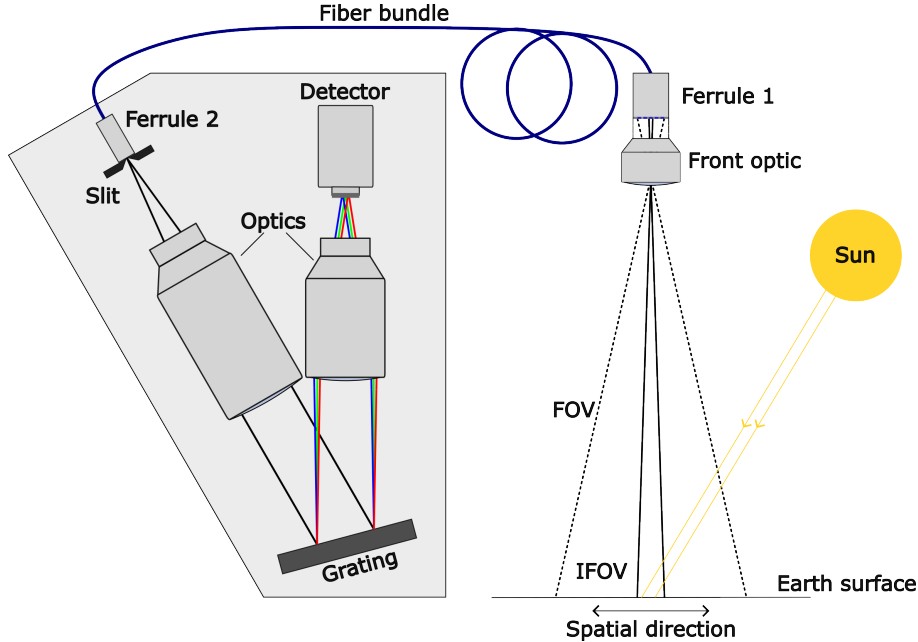

**Figure 1.** Schematic optical setup of the MAMAP2D-Light instrument. The reflected sunlight from the Earth's surface is imaged by the front optic on the input of a fiber bundle with 36 fibers stacked in Ferrule 1, where each fiber corresponds to a single spatial ground scene. The radiation enters the spectrometer block through the fibers in Ferrule 2. The fibers are stacked perpendicular to the optical bench of the spectrometer block. To adjust the linewidth of the spectrometer a Slit is placed in the entrance focal plane. The radiation is dispersed and imaged at the 2D detector with the two optics and the grating. The area of a single fiber together with the focal length of the front optics defines the instantaneous field of view (IFOV). 28 fibers are imaged at the detector, determining the field of view (FOV).

lens collimator with a focal length of $F_c = 300\,\text{mm}$ and an aperture of $F/N = 3.5$. The dispersed collimated light from the grating is then focused on the detector by the camera lens optics with $F_o = 200\,\text{mm}$ and $F/N = 2.4$. The angle of the optical axes between the lenses is $32°$. The grating deployed in MAMAP2D-Light is a ruled plane grating with $300\,\text{lines}\,\text{mm}^{-1}$ and a nominal blaze angle of $17.5°$, which is operated at the -1$^{\text{st}}$ order.

During the CoMet 2.0 campaign, the installed slit in Fig. 1 was adjustable. The slit, shown in Fig. G1 (a), consists of two

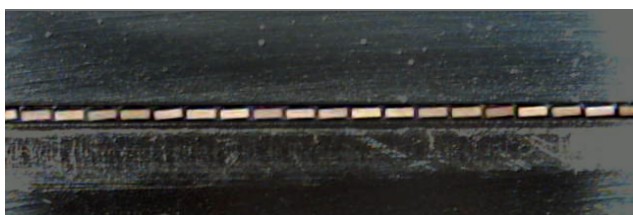

**Figure 2.** Image of a section of aligned fibers within the aluminum ferrule from Fig. 1. The fiber core dimension is $\sim 300\,\mu\text{m} \times 100\,\mu\text{m}$.

95





uncoated steel blades. Initially, it was intended to adjust the ISRF and the spectral oversampling with the slit. However, due to misaligned fibers (see Fig. 2) in the entrance ferrule, the slit was not used and was therefore left open to its maximum using only the fiber geometry as the slit.

The swath of the instrument is defined by the focal length of the input front objective $F_f = 25\,\mathrm{mm}$, in combination with the FPA spatial pixel count, the pixel size and the imaging ratio $F_o/F_c$. The total swath is defined by the fully imaged fibers on the FPA since the fiber bundle length is larger than the detector width. The resulting across-track field of view (FOV) for the full detector is about $23.3°$. However, the exact field of view is defined by the length of the input fibers fully imaged on the detector. This leads to a real FOV of $22.6°$. For the CoMet 2.0 campaign, MAMAP2D-Light was integrated on a Gulfstream G 550 (HALO, High Altitude and LOng Range Research Aircraft, operated by the DLR, Deutsches Zentrum für Luft- und Raumfahrt). With a flight altitude of $\sim 8\,\mathrm{km}$, the FOV of $22.6°$ led to a swath width of $\sim 3.5\,\mathrm{km}$ at a above ground level, with a sampling of 28 spatial fibers, corresponding to an across-track spatial resolution of $\sim 120\,\mathrm{m}$. The along-track ground scene size is dependent on the flight speed, the integration time, and the number of binned ground scenes, and was adpated to $\sim 120\,\mathrm{m}$ by binning $\sim 5$ single measurements for the flights in Canada during CoMet 2.0.

The signal-to-noise ratio (SNR) of MAMAP2D-Light is simulated from an instrument model, which has been developed initially for the MAMAP instrument by Gerilowski et al. (2011). The SNR is estimated for an albedo of $0.12$ and a sun zenith angle of $50°$ at an exposure time of $\sim 70\,\mathrm{ms}$. The considered noise contributors are the shot noise of the expected signal estimated by a radiative transfer model (RTM), the background signal including the detector and ambient dark current, and the read-out noise of the detector. Binning the $8$ spectral rows of a single fiber increases the SNR by a factor of $\sqrt{8}$. The SNR is estimated as $SNR_{single} \approx 600$ for a single measurement. Depending on the exposure time, the flight altitude, and the ground speed of the used aircraft, an along-flight track binning of 10 single measurements is applied to achieve quadratic ground scenes. For the CoMet 2.0 setup, this results in an SNR of $SNR_{quad} \approx 1900$.

## 3   The impact of stray light in the WFM-DOAS retrieval

The GHG anomalies are retrieved from the measured spectra using the WFM-DOAS method. This method does not consider any corrections for an additive error signal, which is the expected type of error resulting from stray light contamination. This section describes the WFM-DOAS retrieval and the impact of an additive offset within the WFM-DOAS retrieval.

MAMAP2D-Light measures the spectra of the sunlight passing through the atmospheric column. Depending on the depth of the absorption bands of the corresponding GHG, the anomalies of GHG concentrations are retrieved from the spectra using the WFM-DOAS method. The WFM-DOAS retrieval was initially developed for the spaceborne SCIAMACHY instrument by Buchwitz et al. (2000). The algorithm was later adapted for the airborne measurement geometry by Krings et al. (2011) for the MAMAP instrument. Krautwurst et al. (2024) describe the retrieval algorithm's latest version as applied to MAMAP2D-Light data.

Based on Lambert Beer's law, a calculated RTM at a wavelength $R_\lambda^{mod}(\overline{\mathbf{c}})$ for a state of the atmosphere represented by the model state vector $\overline{\mathbf{c}}$, can be modulated to get the RTM at the state $\mathbf{c}$ of the measurement $R_\lambda^{mod}(\mathbf{c})$. The weighting functions,





$W_{\lambda,\bar{c}_j}$, describe the change of radiance due to a change of the respective parameter $j$. An additional low-order polynomial $P_\lambda$ with a free parameter vector $\mathbf{a}$ approximates slow spectral variations due to scattering or spectral surface reflectance, which have to be considered but are not quantified. This results in the following equation:

$$\ln R_\lambda^{mod}(\mathbf{c},\mathbf{a}) = \ln R_\lambda^{mod}(\bar{\mathbf{c}}) + \sum_j W_{\lambda,\bar{c}_j} \frac{c_j - \bar{c}_j}{\bar{c}_j} + P_\lambda(\mathbf{a}) + \varepsilon_\lambda \tag{1}$$

The state vector of interest $\mathbf{c}$, where each element represents a parameter $j$, e.g., GHG concentrations, is retrieved from a real measured spectrum $R_\lambda^{mea}$ by a least squares fit with the fit parameters $\mathbf{c}$ and $\mathbf{a}$. Thereby, the residuum $\epsilon_\lambda$ approaches a minimum:

$$\left\| \ln R_\lambda^{mea} - \ln R_\lambda^{mod}(\mathbf{c},\mathbf{a}) \right\|^2 \equiv \|\epsilon_\lambda\|^2 \longrightarrow min \tag{2}$$

Stray light is radiation deviating from the intended light path and illuminating the FPA at unintended positions. The position of the intended path in the focal plane is called the origin position, and the unintended position is called the target position. In this work, the stray light terminology is adapted from Fest (2013) and described in detail in Appendix A. Stray light causes an additive error signal $e$ at the focal plane, also called a zero-level offset. The error signal occurs in the target spectrum and, by being absent, also in the origin spectrum. The fitting in Eq. 2 is then performed to a real measured spectrum of

$$\ln R_\lambda^{mea,real} = \ln(R_\lambda^{mea} + e). \tag{3}$$

However, the polynomials $P_\lambda(\mathbf{a})$ are additive components to the logarithm of the radiance in Eq. 1, and therefore, scalable multiplicative factors of the radiance. Consequently, in WFM-DOAS, the additive offset $e$ is tried to be compensated for by a multiplicative scaling factor of the polynomial. This introduces a signal level-dependent scaling error, which leads, in the case of a positive error signal, to a shrinking of the absorption line depths relative to the continuum. The corresponding fitting parameter $c_j$ then "sees" shallower trace gas absorption bands, which leads to an underestimation of the retrieved column anomaly. Therefore, an additive offset can not be observed in the spectral residuals $\epsilon_\lambda$ of the fit, except for areas in the spectral window without any trace gas-related absorption bands, e.g., pure Fraunhofer-Lines.

MAMAP2D-Light is designed to quantify GHG anomalies relative to the background concentrations. Due to the normalization of the retrieved columns to the background in post-processing, described in detail in Sect. 6, a constant additive offset would not impact the precision of the retrieved column anomalies. However, the impact of stray light depends on the radiation of the source and the amount of the intended radiation within the target spectrum. Thus, scenes with inhomogeneous albedo, spectral surface reflectance and aerosol scenario result in decreased precision in the retrieved column anomalies.

## 4 Stray light characterization in MAMAP2D-Light

The stray-light-related error signal introduces errors in the retrieved and not further corrected GHG column anomalies, as deduced above. It is, therefore, essential to characterize the stray light in the instrument. For MAMAP2D-Light, this was performed by dedicated characterization measurements. These measurements were used to identify the origin of the stray light



and mitigate it by design and to use the measurements for a post-flight stray light correction. The following stray light charac-
terization measurements were performed in MAMAP2D-Light in the configuration flown during the CoMet 2.0 campaign in
2022, which introduced substantial amounts of stray light.

In this paper, the stray light is quantified by the methodology described by Tol et al. (2018), whereby a spatially and spectrally
minimal spot is illuminated, and the corresponding light at the detector (defined as point response function, PRF) is measured.

The spot area of the PRF is limited spectrally by the instrumental spectral response function (ISRF) and spatially by the point
spread function (PSF) of the spectrometer optics convoluted with the fiber geometry.

The optical setup for the stray light characterization measurements is shown in Fig. 3. A Littman/Metcalf laser system (Lion
System by Sacher Germany, Stry et al., 2006), with a tunable wavelength range from $1600\,\mathrm{nm} - 1750\,\mathrm{nm}$ at a movement preci-
sion of $0.05\,\mathrm{nm}$ and a power of $\sim 20\,\mathrm{mW}$ was used as a tunable monochromatic light source. The laser diode's side modes are

suppressed by the Littman/Metcalf configuration, which is wavelength-dependent. The manufacturer determined the side-mode
suppression for several wavelengths. As an example, it is $55.4\,\mathrm{dB}$ at $1625\,\mathrm{nm}$, measured with a spectral resolution of $0.05\,\mathrm{nm}$,
see Fig. B1.

The actual wavelength of the laser was observed using a laser wavelength meter (671A, by Bristol) with an accuracy of $\pm 0.2\,\mathrm{pm}$
at $1000\,\mathrm{nm}$ for the range from $520\,\mathrm{nm} - 1700\,\mathrm{nm}$. The laser was fed to an integrating sphere with an inner diameter of 5.3"

(IS6-C, by Ophir). An adjustable slit was imaged by a relay optic consisting of two lenses (FO1 and FO2), shown in Fig. 3,
on a single fiber of the entrance ferrule to illuminate a single fiber of the entrance fiber ferrule (F) of MAMAP2D-Light. By
moving the slit in the direction of the stacked fibers, different fibers were illuminated.

For a flat field correction, which accounts for pixel response non-uniformity (PRNU) errors, measurements of a fully illumi-
nated slit and FPA were performed with a spectrally calibrated sphere (UMBB-500 by Gigahertz-Optik GmBH, diameter of

20") with four integrated $50\,\mathrm{W}$ broadband Quartz Tungsten Halogen lamps. This white light measurement was corrected by
the dark current and divided by the corresponding spectral radiance derived from the calibration curve of the sphere and the
generated wavelength grid from Appendix H1 for each pixel.

The stray light was quantified at 21 positions across the FPA (3-4 spectral at 6 spatial positions). At each position, 100 frames at

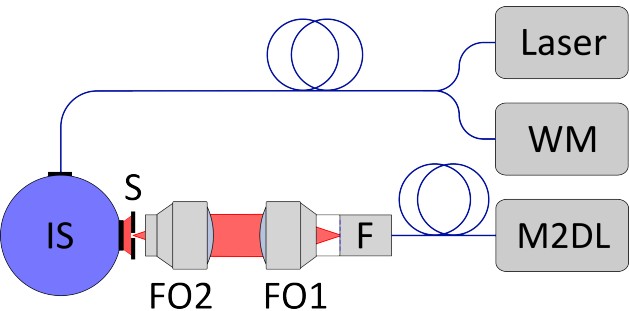

**Figure 3.** Optical setup for stray light measurements. A Tuneable Littman/Metcalf laser emits laser light which is fed into a Wavemeter
(WM) and an integrating sphere (IS) via a y-fiber. At the output port of the sphere, an adjustable slit (S) is assembled, which is imaged with
two objectives (FO1 and FO2) on a single fiber of the input fiber bundle ferrule (F) of MAMAP2D-Light (M2DL).





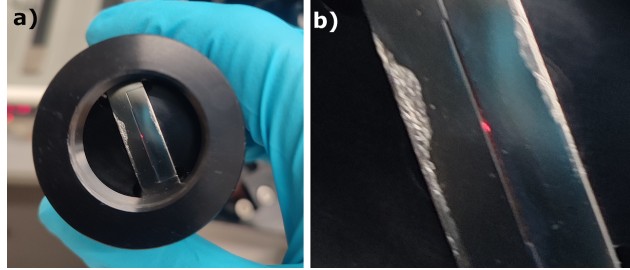

**Figure 4.** (a) Photograph of the entrance aluminium fiber ferrule, Ferrule 2 in Fig. 1, with a black anodized mount. A single fiber is illuminated with the setup shown in Fig. 3 and a white light source. The illuminated fiber is visible as a red dot. (b) zoom in on the fiber stack.

10 different exposure times were recorded. The exposure times were increased from $10\,\mathrm{ms}$ to $3000\,\mathrm{ms}$ to increase the dynamic range of the measured signal. The dark signal level, increasing due to thermal radiation, constrained the highest exposure time. The dark signal for each point was measured for each exposure time after a complete set of exposure times with illumination, by shutting off the laser. The measurements were flat-field corrected, where the fibers' cladding areas (displayed as dark lines in the spectral direction in Fig. D1) were interpolated by fitting a 2-dimensional $3^{\mathrm{rd}}$-order polynomial to the fiber core signal. The dark current corrected data showed patterns related to a detector effect which were most prominent for higher exposure times. The patterns were corrected using a data-driven approach, which is shown in detail in Appendix C.

The measured signals at one position for all exposure times were merged into a single two-dimensional frame. Therefore, saturated and blooming-contaminated pixels had to be filtered out. Blooming[2] occurred in the measurements at higher exposure times due to the saturation of the directly illuminated pixels. Due to the CMOS-based read-out electronics of the detector, only the directly neighbouring pixels of a saturated pixel are affected by blooming. For merging, each non-saturated and non-blooming-contaminated pixel value at the highest exposure time was selected. The full merged frame was finally normalized to the integral of the signal over all pixels. The merged frames of the stray light characterization measurements for MAMAP2D-Light in the CoMet 2.0 configuration for four different spot positions are shown in Fig. 5. The measurements revealed several stray light and non-stray-light-related artifacts which are discussed in the next section.

## 4.1 Stray light contributors

The stray light contributors were separated into a spectrally and spatially *invariant* part independent of and a spatially *variable* part depending on the position of the illuminated spot. Within the measurements, the relative spectral position of the *variable* stray light was constant. The description for the stray light sources uses the terminology defined in Appendix A, where the stray light is classified in different orders. With each stray light process (e.g., scattering, reflection, etc.) in a light path, the order is increased, starting with the intended light path as the $0^{\mathrm{th}}$ order.

---

[2]Blooming occurs due to photogenerated charges within a saturated pixel, which are not fully collected by the pixel's read-out electronics. The leftover charges are then collected by the neighbouring pixels.




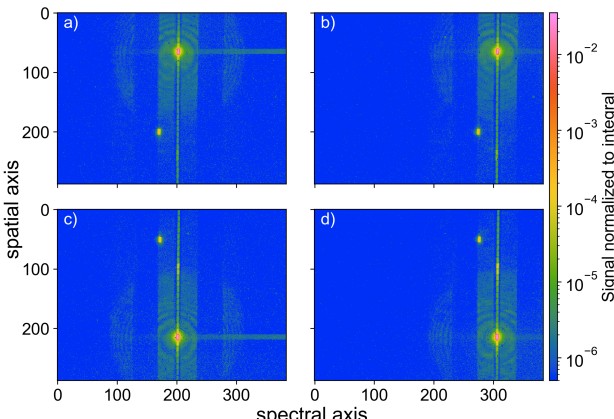

**Figure 5.** Different spectral and spatial spots from the stray light characterization of MAMAP2D-Light in the CoMet 2.0 configuration. (a) and (c) were recorded at $\sim 1628\,\mathrm{nm}$, (b) and (d) at $\sim 1661\,\mathrm{nm}$. The horizontal line at the right-hand side of the illuminated spot is due to the not completely suppressed side modes of the laser used. The vertical line through the illuminated spots is caused by light from outside the instrument. A sharp ghost appears spatially mirrored but spectrally in a constant offset from the initial spot. Further, the spectral and spatial invariant stray light cone around the illuminated spot is shown in all images.

The *invariant* stray light forms a wide-spreading cone around the illuminated spot. The cone is split up by two $\sim 40\,\mathrm{pixel}$ wide vertical stripes. The size of the lines matches the size of the blade edges of the unused adjustable slit in Fig. G1 (a). This leads to the conclusion, that the cone originates from scattered radiation described by the Bidirectional Reflection Distribution Function (BRDF) of all optical components, which then illuminates the critical high reflective surfaces in the object plane of the entrance slit, namely the aluminum ferrule and the steel blades of the adjustable slit. This results in at least $2^{\mathrm{nd}}$ order stray light at the detector. Within the areas of the blade edges, the BRDF-originating radiation is reflected outside of the intended light paths due to the angle of the blade surface relative to the optical path. Thus, it is expected that the $1^{\mathrm{st}}$ order scattering processes described by the Bidirectional Transmittance Distribution Function (BRTF) of the refractive optics and the BRDF of the grating are dominating in this area. This stray light is five orders of magnitude lower than the signal, and, therefore, only detectable in the merged frames (see Sect. 5 Fig. 6).

The *variable* stray light occurs as a sharply imaged ghost, which moves spatially mirrored relative to the spatial position of the illuminated spot. In the spectral direction, the distance between the ghost and the illuminated spot is constant (i.e. always $31.8\,\mathrm{pixel}$). The ghost is a sharp image and therefore must originate from reflected stray light whose light path is focused on the detector. Analyses depending on the reflections from the entrance focal plane mentioned above have shown that the ghost vanished after inserting a blackened slit aperture, see Sect. 8. This leads to the conclusion that stray light paths are focused at the entrance focal plane, which is then reflected and imaged at the detector. The ghost is not originating from the focal plane of the detector, since the spatial variations are mirrored at the FPA.

Another potential stray light contamination in the measurements occurs as a dashed line in a vertical direction from the illu-




minated spot. The single line segments occur due to radiation passing through the non-intendedly illuminated fibers and are
therefore the result of stray light from in front of Ferrule 1 in Fig. 1. In this stray light measurement configuration, it was not
possible to distinguish between the stray light originating from the paths from the front optics to the ferrule and the stray light
originating from the optical stray light measurement optical relay (FO1 and FO2) in front of the instrument.

The horizontal line on the right-hand side of the illuminated spot is a consequence of the already mentioned side modes of the
used laser, described in Appendix B.

The stray light within the light path from Ferrule 1 to the FPA of MAMAP2D-Light is $(3.9 \pm 0.32)\,\%$, see Sect. 8.

## 5   Post-flight stray light correction

The stray light characterization measurements following the CoMet 2.0 mission revealed the presence of a significant amount
of stray light, $\sim 4\,\%$, see Sect. 8. Consequently, a post-flight stray light correction was implemented based on the procedure
described by Tol et al. (2018), utilizing the characterization measurements outlined in Sect. 4.

The two types of stray light shown in Sect. 4.1 were corrected by separate methods. The invariant or stable stray light was
represented by a stable kernel $\mathbf{K}_{stable}$. The variable stray light was represented by a reflection kernel $\mathbf{K_{refl}}$. The terminology
is adapted from Tol et al. (2018), although the majority of the stable stray light had its origin from a reflection process in at
least the $2^{nd}$-order.

### 5.1   Stable kernel $\mathbf{K}_{stable}$

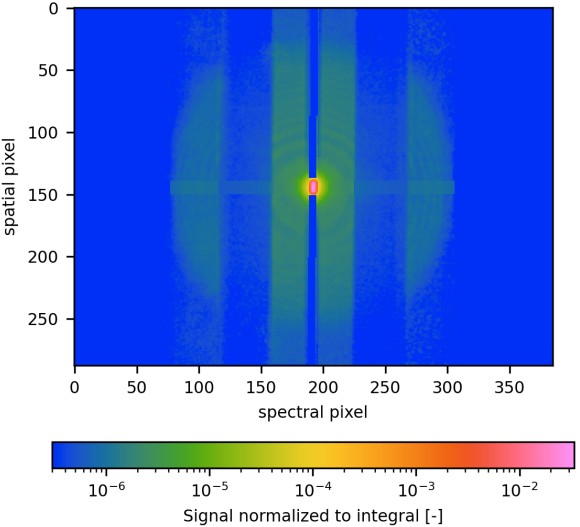

**Figure 6.** Stable Kernel for stray light correction after filtering vertical and horizontal non-stray-light related artifacts



All the measured spots from the stray light characterization measurements were shifted to the center. The position of the spots was derived with the python function "ndimage.center_of_mass" (version 1.13.1) due to the non-Gaussian ISRF and PSF. For best overlap, the "shift" function from the "scipy.ndimage" python package (version 1.13.1) was used for a linear interpolation to shift on a sub-pixel level. The median of all shifted measurements formed $\mathbf{K}_{stable}$. Due to the median, the variant stray light vanished.

The laser used had insufficient side-mode suppression, leading to unreliable data in the horizontal direction. The resulting data gap was interpolated by a method described in Appendix E. Furthermore, the vertical line consisting of stray light from the optical setup in front of the entrance fiber ferrule was set to zero. This, however, did not take into account pure spectral stray light induced from shape irregularities of the grating itself. Following this, $\mathbf{K}_{stable}$ was normalized to the integrated signal over all columns $k$ and rows $l$, such that $\sum_{k,l}(\mathbf{K}_{stable})_{k,l} = 1$, see Fig. 6.

The stable kernel comprises the PRF and the stable stray light. The stray light is defined to be in the far field of $\mathbf{K}_{stable}$; the near field of $\mathbf{K}_{stable}$ comprises the PRF. Consequently, $\mathbf{K}_{stable}$ was split into $\mathbf{K}_{far}$ and $\mathbf{K}_{near}$. The stray light was corrected using an iterative deconvolution approach described by Tol et al. (2018):

$$\mathbf{J}_i = \frac{\mathbf{J}_0 - \mathbf{K}_{far} * J_{i-1}}{1 - \sum_{k,l}(\mathbf{K}_{far})_{k,l}} \tag{4}$$

The ideal frame $\mathbf{J}_n$ was derived after $n = 3$ iterations, as described Tol et al. (2018), further iterations showed sub-DN changes, 255    starting with the measured, dark current and flat field corrected frame as $\mathbf{J}_0$. By this method, the stray light was redistributed in $\mathbf{J}_n$.

## 5.2    Reflection kernel $\mathbf{K}_{refl}$

The spatial variable stray light contaminated the measured spectrum with a spectrally shifted image of the corresponding spatial 260    spectrum. The corrected frame $J_{corr}$ was derived from the measured frame $J$, the relative intensity variability of the ghost spot $\mathbf{E}_{refl}$, a spatial and spectral transformation through convolution with the reflected kernel $\mathbf{K}_{refl}$, and a mirroring operation of the y-axis $R$. The reflected stray light should be redistributed instead of subtracted, similar to $\mathbf{K}_{stable}$. Therefore, the term $(\mathbf{E}_{refl} \cdot J)$ was added in the correction:

$$\mathbf{J}_{corr} = \mathbf{J} - \mathbf{K}_{refl} * (\mathbf{E}_{refl} \cdot \mathbf{J})^R + (\mathbf{E}_{refl} \cdot \mathbf{J}). \tag{5}$$

The reflection Kernel $\mathbf{K}_{refl}$ was determined from the relative positions of the ghost spot to the originally illuminated spot, see Fig. 5. In the spectral direction, the relative offset $x_{refl}$ was constant. In the spatial direction, the ghost spot was mirrored and shifted by $y_{refl}$ from the center. A spot search algorithm defined $x_{refl}$ and $y_{refl}$ based on the relative distances between ghost and origin spots' barycenters. $\mathbf{K}_{refl}$ shifted the frame to the ghost position. Since the ghost spot was a sharp image, $\mathbf{K}_{refl}$ would be ideally a single pixel with the value 1 at $x_{refl}$ and $y_{refl}$. However, due to floating values, the signal pixel was initially 270    set to the nearest integer value and afterward shifted by the decimal points using the "shift" function of the "scipy.ndimage" package (version 1.13.1) in python. Thus, the signal in $\mathbf{K}_{refl}$ had an area of $2\,\text{pixels} \times 2\,\text{pixels}$.





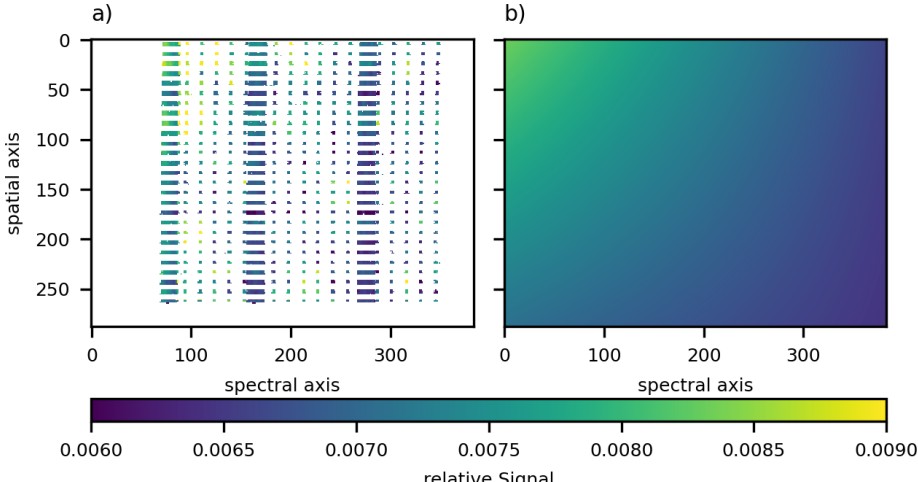

**Figure 7.** Relative intensity distribution of the reflected stray light. (a) Data extracted from measurements. (b) two-dimensional first-order polynomial fit.

The relative intensity variability of the ghost spot and the origin spot is represented by $\mathbf{E}_{refl}$ and was generated from the wavelength grid, instrumental response function (Appendix H1 and H2) and the stray light characterization measurements using the equation:

$$\mathbf{E}_{refl} = \frac{\mathbf{S}^{R}_{refl}(x - x_{refl}, y - y_{refl})}{\mathbf{S}_{origin}(x, y)}. \qquad (6)$$

$\mathbf{S}_{origin}$ represents the signal of the origin spot and $\mathbf{S}_{refl}$ is the corresponding signal of the reflected spot, which is shifted by the corresponding $x_{refl}$ and $y_{refl}$ values. The respective signal levels within a fiber were determined by the mean intensity of the spot, defined by a half-maximum threshold. The $R$-operator is mirroring the y-axis. Due to the sparse data availability for $\mathbf{E}_{refl}$, a two-dimensional first-order polynomial fit was deployed to fill the data gaps, shown in Fig. 7. Higher orders in the fit function led to a stronger variability of the values in the unknown edges. The RMS of the relative fit residuals was $\sim 8\,\%$. A more accurate $\mathbf{E}_{refl}$ estimation would either require a denser grid of stray light measurements or, e.g., wavelength grid measurements with an increased dynamical range, as done for the stray light characterization measurements, see Sect. 4.

The second term in Eq. 5 ($\mathbf{K}_{refl} * (\mathbf{E}_{refl} \cdot \mathbf{J})^{R}$) represents the amount of reflected stray light in the frame. However, the slit was not perfectly aligned vertically, and due to the smile effect[3] slightly curved. This distortion needed to be corrected before the mirroring operation was performed and reversed before subtraction. The correction was achieved by shifting each row by a value $x_{smile,\,row}$. This value was determined by the difference between the barycenter of each row from a measurement of a full slit and the median of all barycenters from the same measurement. The resulting $\mathbf{x}_{smile}$ array for all rows was the median

---

[3]The smile effect is occurring at planar gratings due to geometric differences of the dispersion angle in the spatial direction, this is causing a spectral deformation of the imaged slit at the FPA





for each row from the wavelength grid and instrumental response function (refer to Appendix H1 and H2) measurements. The correction is only valid due to the relatively small wavelength dependency of the diffraction angle defined by the groove frequency of the grating with $300\,\mathrm{lines\,mm^{-1}}$.

### 5.3 Out-of-field stray light in MAMAP2D-Light

Due to the extension of $\mathbf{K}_{stable}$ and the offset of $\mathbf{K}_{refl}$ from the center, the out-of-field stray light (OFSL) contaminated the edges of the measured frame. To consider the spectral OFSL (or out-of-band stray light) in the correction, the spectral axis of the measured frame was extrapolated with an extended RTM, as used in Sect. 6. The RTM was fitted to each row of the dark current and flat field corrected frame, scaled with a polynomial ($3^{\mathrm{rd}}$-order) and spectral shift parameter within the spectral range of MAMAP2D-Light. The extended spectra were then derived from scaling and shifting the full RTM range with the derived fit parameters. This method of extrapolation gives only an estimation of the signal level of the spectral OFSL and the expected impact is discussed in Appendix D3. It is important to note that the surface spectral reflectance and the aerosol scenario have an impact on the signal level of the spectral OFSL and would affect the correction quality even with a perfectly characterized system.

The spatial OFSL was neglected within the correction for two reasons. First, getting reasonable information about the spectral surface reflectance near the flight track post-flight is challenging. Second, the entrance ferrule consists of 36 fibers, from which 28 fibers were fully and a $29^{\mathrm{th}}$ fiber partially imaged at the detector, limiting the source area for spatial stray light to approximately 3.5 fibers, equivalent to 35 pixels on each side. Simulations considering the full spectral and spatial OFSL showed only a minor impact of the spatial OFSL on the column noise in the retrieved data, see Fig. 13.

### 5.4 Applied stray light correction

The correction for the stable and the reflected stray light from Sect. 5.1 and 5.2 was applied to a laser measurement with a fully illuminated entrance slit at a given wavelength. The measurement was dark current and flat-field corrected. Further, the bad pixels were linearly interpolated before the correction. The measured and the corrected frame are shown in Fig. 8. The ghost is visible as a dashed line left from the Laser signal in the measured frame. In the correction, the shade from the stable stray light vanishes nearly completely. The intensity of the ghost is decreased and at some pixels, it is overcorrected. Due to the spatial shift of the reflected stray light ($x_{refl}$ in Sect. 5.2), the lower two fibers of the ghost are not corrected. The standard deviation of the measurement is derived by excluding the slit (i.e. direct signal) area. The stray light correction is reducing the measured standard deviation $\sigma_{meas} = 0.060\,\%$ to $\sigma_{corr} = 0.025\,\%$.

### 6 Impact of stray light on retrieved concentrations

The post-flight stray light correction from Sect. 5 was applied to the data collected during the CoMet 2.0 Arctic mission. The stray-light-corrected and uncorrected frames were retrieved with the WFM-DOAS retrieval individually, and the resulting sin-





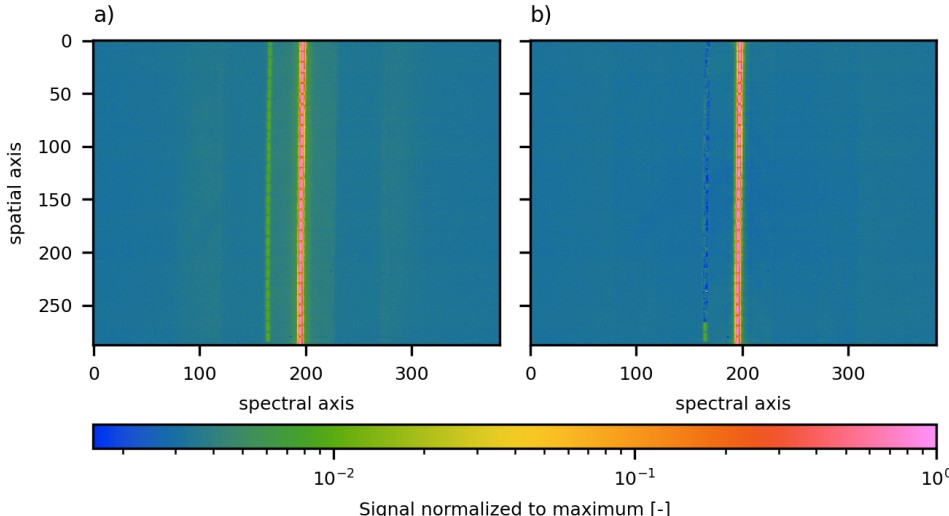

**Figure 8.** Stray light correction applied to a laser measurement performed at $1625.88\,\text{nm}$. Bad detector pixels are linearly interpolated. (a) dark current and flat-field corrected data. (b) with applied stray light correction.

gle $CH_4$, $CO_2$ columns and proxy-corrected column anomalies are compared. As discussed in Sect. 3, it is expected that the
stray light has an impact on the column noise of the retrieved column anomalies. To separate the noise contribution of the stray
light from other noise sources, simulated synthetic measurements are included in the analysis.

## 6.1 Data processing

The column anomalies were retrieved with the airborne WFM-DOAS method, which is described briefly in Sect. 3 and in detail
by Krautwurst et al. (2024). The retrieval delivers column anomalies from the trace gases of interest as profile scaling factors
(PSF) of atmospheric profiles at the mean state of the atmosphere during the measurements using an RTM calculated with
SCIATRAN 3.8 (Rozanov et al., 2014). The spectra were dark current corrected, radiometric calibrated by a calibrated sphere
measurement, see Sect. 4, and wavelength calibrated. The retrieved data was filtered using a root-mean-squared (RMS, see in
Sect. 3) threshold of the fit residuals to assess the quality of the fit. To account for signal intensities exceeding the linearity
range of the detector and to keep a sufficient signal-to-noise ratio, a maximum and minimum signal threshold was applied.
The retrieved column data showed a nonlinear dependency on the detector filling. This phenomenon has already been observed
for MAMAP data and is discussed by Krautwurst et al. (2017). For MAMAP2D-Light, the nonlinear dependency for each
spatial sample was corrected with a data-driven approach analogous to that developed for MAMAP. A low-order polynomial



($2^{nd}$ - $3^{rd}$ order) was fitted to the column data over the detector filling for one spatial fiber over a single flight leg. The column
data was then normalized by the fit result.

Typically (Krings et al., 2013; Krautwurst et al., 2017, 2024), the proxy method is used to minimize the impact of light-path
errors, like multi-scattering or instrumental error. The $CH_4$ proxy is the ratio of the retrieved $CH_4$-PSF and the $CO_2$-PSF,
assuming a constant $CO_2$ concentrations over the measurements area:

$$CH_{4,proxy} = CH_{4,psf}/CO_{2,psf}. \tag{7}$$

However, the proxy method underestimates mixed plume signals. Therefore, in this work, the non-proxy corrected single
columns are also analyzed in more detail.

Depending on the altitude at which the $CH_4$ plume and therefore the concentration perturbation is located, the WFM-DOAS
retrieval has varying sensitivities. This sensitivity is described by the altitude-dependent averaging kernel $AK(z)$ (Krings
et al., 2011). For the CoMet 2.0 data, it was computed for each ground scene, considering its respective surface elevation and
assuming that all enhancements are located below the aircraft. Based on the $AK(z)$, conversion factors $c_f$ were derived used
for correction of the retrieved PSFs:

$$CH_{4,rel} = (CH_{4,psf} - 1) \cdot c_f \tag{8}$$

The column data was georeferenced using the aircraft position and attitude and the surface elevation, described in detail by
Krautwurst et al. (2024).

**6.2    Stray light in high contrast scenes**

Initial results of the CoMet 2.0 campaign dataset revealed an error pattern in the proxy-corrected $CH_4$ column anomalies for
the scene shown in Fig. 9. The data was processed using the retrieval, RTM, and orthorectification parameters shown in Tab. F2.
In the non-stray-light-corrected concentration map in Fig. 9 (c), significantly enhanced $CH_4$ column anomalies are shown. The
intensity map in Fig. 9 (b) revealed a high contrast scene, where the surfaces consist of highly reflective sand, low-reflective
vegetation, and a nearly non-reflective lake. The $CH_4$ column anomaly pattern resembles the mirrored sand surface, which
aligns with the mirrored ghost seen in Fig. 5. After the stray light correction, the structures in the $CH_4$ column anomalies were
reduced, but not erased. This is related to the not accurately known reflection intensity distribution $\mathbf{E}_{refl}$ in Fig. 7 and Sect.
5.2.

The stray light correction also reduced further negative column anomalies, which were located in ground scenes with low
intensity compared to the across-track neighbouring ground scenes. In this scene, with applied proxy correction, the total
column noise was reduced from $0.40\,\%$ to $0.33\,\%$ by the stray light correction.

The impact of the stray light mainly depends on the signal level and distribution of the origin and the signal level of the
target spectrum. In Fig. 10, the reflected and the stable error signal for a target spectrum are shown. The reflected stray light
introduces a more structured and different curved error signal, whereas the stable stray light is smoother and follows the curve
of the target spectrum. The proxy method is unable to correct imbalanced error contamination in the $CO_2$ and $CH_4$ bands.




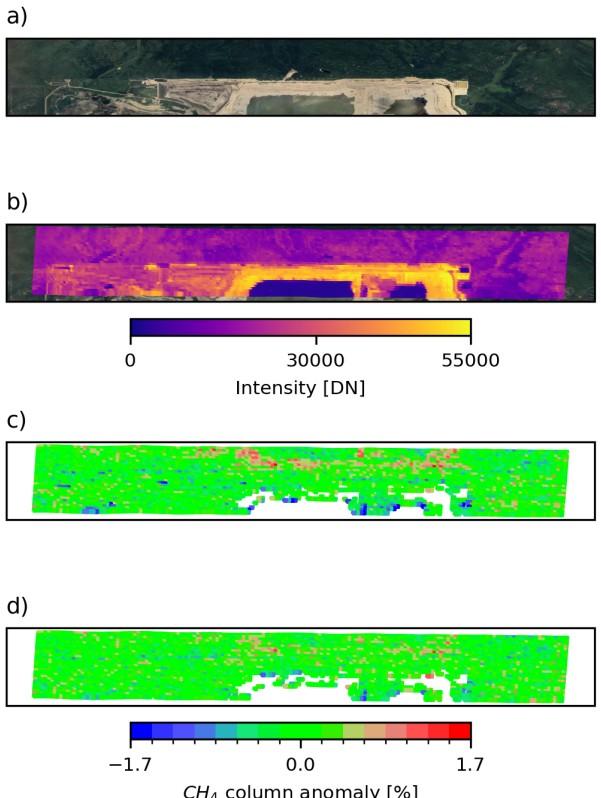

**Figure 9.** Measured scene over high reflective sand and low reflective vegetation. The surface in RGB is shown in (a). (b) shows the intensity in the SWIR measured with MAMAP2D-Light. The non-stray-light-corrected and proxy-corrected processed data is shown in (c), with a column noise of $0.40\,\%$. The stray-light-corrected data in (d) has a reduced column noise of $0.33\,\%$. The RGB map is provided by © OpenStreetMap, accessed using Cartopy.

Due to the different absorption line depths, a general imbalance of the sensitivity to a zero-level offset is given; if the zero-level offset varies spectrally, the imbalance can be compensated or amplified. The shown target spectrum is the corresponding synthetic spectrum, which is generated as described in Appendix D, of an enhanced pixel in Fig. 9 (b), which is caused by the contamination of the reflected stray light.

## 6.3 Stray light as source for pseudo-noise

The stray light introduced error patterns in the concentration maps can be observed as pseudo-noise in the column noise estimate of the real measured column anomalies. Therefore, in the following, the variation of the column anomalies is analyzed based on a flight leg, shown in Fig. 16, over an area dominated by urban and agricultural surfaces, where plume signatures were masked. The flight leg was chosen due to the strong variations in surface reflectance. Further, based on the real measured



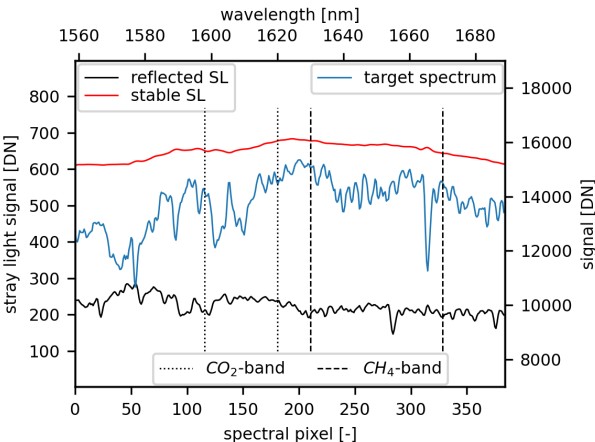

**Figure 10.** Separated stray light (SL) error signal from the sharp ghost reflex (reflected SL) and the stable kernel (stable SL) for a target spectrum in a simulated frame.

frames, synthetic frames were generated and artificially contaminated with stray light and random noise to simulate the different error types individually in the processing chain. The concentration anomalies were retrieved using the parameters shown in Tab. F1.

### 6.3.1 Column noise in real measured data

The column noise of the real measured column anomalies was estimated from the standard deviation of the source-free back-
ground area. In Fig. 11, the distribution and the column noise of the non-proxy-corrected single columns and proxy-corrected columns with and without applied stray light correction are shown. The column noise of the non-proxy-corrected single columns is significantly improved after the stray light correction. However, after the proxy correction, the stray light correction has no significant impact on the column noise. When comparing the standard deviations of the single $CH_4$ column with the stray light correction to the proxy corrected column, the noise of the single $CH_4$ column is marginally lower. The
increased column noise after the proxy correction is associated with the division of two independent quantities contaminated with random noise.

The impact of spatial stray light is depicted through a correlation of the mean retrieved column anomalies with the mean intensity of a measured frame, as shown in Fig. 12. The intensity of each wavelength-calibrated and dark-current-corrected spectrum is derived from the continuum between $1620.5\,\mathrm{nm}$ and $1623.0\,\mathrm{nm}$, in digital numbers [DN]. Similar to the column
noise in Fig. 11, the correlation of the mean column enhancements with the mean intensities is corrected by the proxy method. However, after the stray light correction, the correlation in the single $CO_2$ and $CH_4$ columns decreases significantly. The effectiveness of the stray light correction differs between the $CH_4$ and $CO_2$ columns, impacting the shown correlation of the proxy and the stray-light-corrected data. This variance may be linked to the OFSL correction outlined in Sect. 5. Due to the





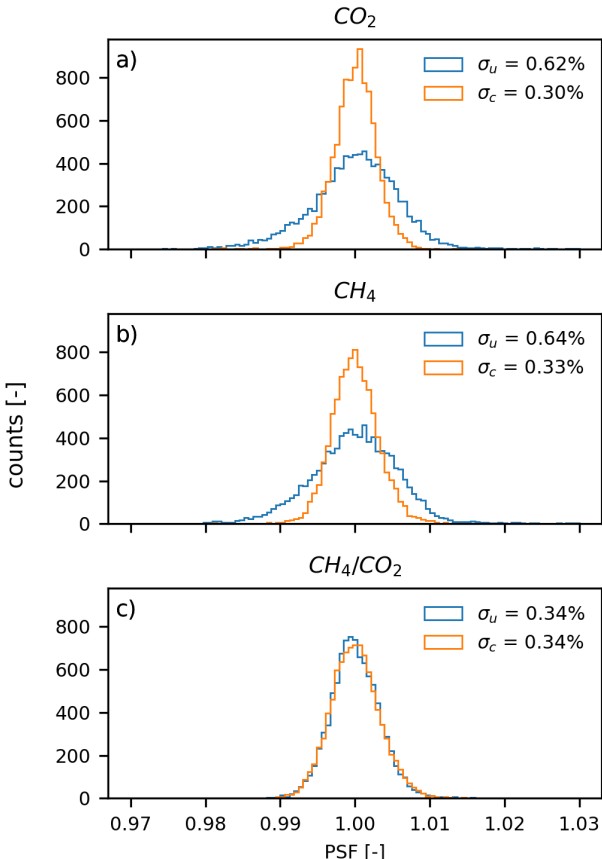

**Figure 11.** Histograms of the retrieved single $CO_2$ (a) and $CH_4$ (b) and the proxy ($CH_4/CO_2$) corrected (c) column data as profile scaling factors (PSFs). The Distributions show data without (blue) and with (orange) stray light correction.

location of the used fit-window (1575 nm - 1677.5 nm) on the detector, the $CH_4$ band is more affected by the OFSL than the

$CO_2$ band. The position of the $CO_2$ and $CH_4$ bands are marked in Fig. 10.

### 6.3.2    Column noise in simulated data

The column noise in Fig. 11 after the stray light correction and after the proxy correction stays in the same range of $0.34$ with along-track binning to get quadratic ground scenes. Further, the histograms have a normal-distribution-like shape, which is an

indication that the total noise is dominated by random noise. To have the possibility to separate the stray light introduced error from random noise error sources, synthetic spectra were generated and contaminated with different error signals, as described in Sect. D.

The different cases and resulting single read-out column noise values are shown in Fig. 13. The first two cases are for comparing





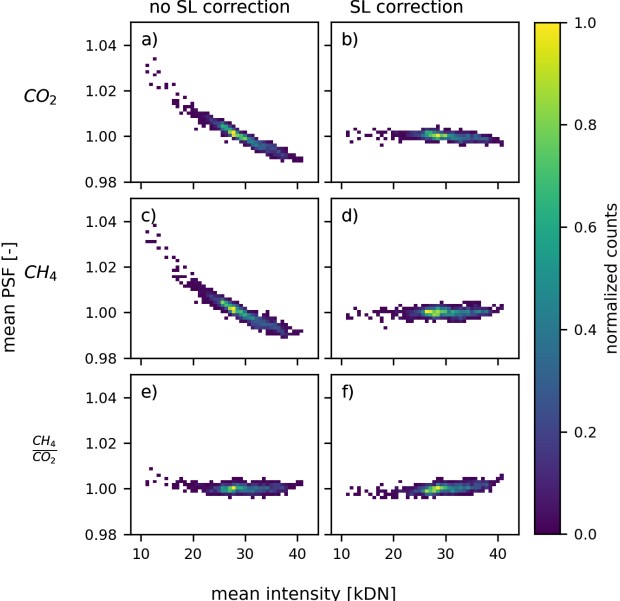

**Figure 12.** 2D-histograms (color) of the average retrieved single $CO_2$ and $CH_4$ and the proxy ($CH_4 / CO_2$) corrected column data dependent of the profile scaling factors per frame on the y-axis and the average intensity of the frame on the x-axis. (a) mean $CO_2$ PSF with no applied stray light (SL) correction shows a strong correlation with the mean intensity. After the stray light correction in (b), the correlation vanishes. The correlation is also visible in the mean $CH_4$ PSF data in (c) and vanishes after the stray light correction (d). (e) and (f) show the correlation for the proxy-corrected data, where the stray light correction has only a minor impact compared to the single columns.

the real measured results with the simulated results and show a close correlation regarding the stray light introduced error,

which is discussed in detail in Appendix D2. The differences between the real non-stray-light corrected data and the synthetic frames with artificially added stray light and random noise contamination highlight that the simulated data is close but not fully accurate. However, a more accurate model would require precise knowledge of the scene's aerosol scenario, spectral surface reflectance and the out-of-field signal. Nevertheless, the synthetic frames can show the impact of the different error contributors.

The stray light correction for the simulated data is the stray light correction of the real measured data, with the differences described in the following. The stable stray light is corrected with a perfectly known stable kernel, from Fig. 6. The spatial OFSL is neglected, and the spectral OFSL is perfectly known during the correction. Due to the spatial shift, see Fig. 8 (b), of the reflected stray light, the lower two fibers are not considered in the column noise estimation.

The stray light correction for the simulated data with random noise contamination shows a similar impact as for the real

measurement; the noise of the single $CO_2$ and $CH_4$ columns is significantly reduced, and the proxy corrected column is only marginally affected by the stray light correction. In general, the proxy method is effective for reducing the stray-light-introduced pseudo-noise, which can be seen in the stray-light-only contaminated data. The column noise of the proxy method





is higher than for the single $CO_2$ and $CH_4$ columns in non-stray-light-contaminated data (whether after stray light correction or a priori without stray light) when random noise was added to the data. This is expected due to the division of two independent

noise-contaminated values. By contaminating the synthetic spectra only with the random noise, the resulting proxy single read-out column noise is at $\sim 0.32\,\%$. This is the theoretically achievable column precision for the analyzed measurement. Without considering the spectral OFSL in the stable stray light correction, the $CH_4$ column noise is $\sim 0.22\,\%$. With perfect knowledge of the spectral OFSL, this value decreases to $\sim 0.11\,\%$. The leftover column noise is related to the spatial OFSL. For the real measured data, the spectral OFSL is approximated, and the impact is discussed in detail in Appendix D3.

The proxy-corrected single read-out column noise of the real data is $\sim 0.46\,\%$, which is $\sim 44\,\%$ higher than the theoretically achievable minimum. The lower column noise in the real single $CO_2$ and $CH_4$ columns, compared to the proxy column noise, is a hint that the discrepancy with the simulation is caused by (pseudo-)noise originating from e.g., unknown features in the surface spectral reflectance or the unknown real OFSL.

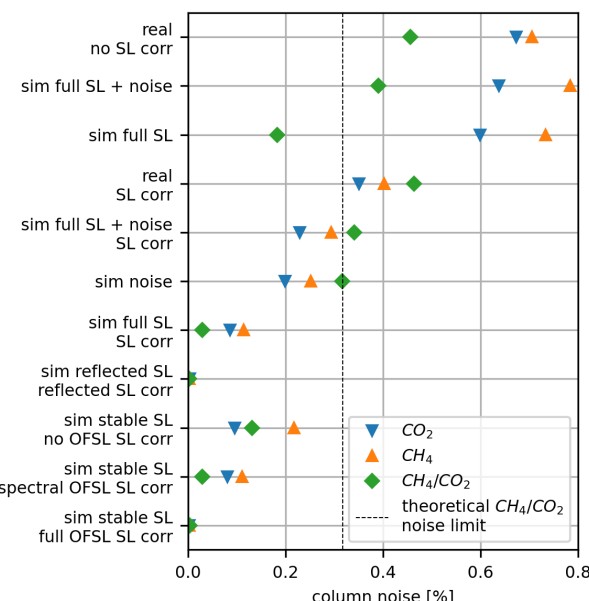

**Figure 13.** Single read-out column noise for retrieved column anomalies for the $CO_2$ (blue triangle), $CH_4$ (orange triangle) and the proxy corrected ($CH_4/CO_2$) (green diamond) column, for different cases (y-axis). The cases are labeled in two lines: the first line contains the data setup with real measured (real) or simulated (sim) spectra and the type of contamination, which is full stray light (SL), random noise (noise), only reflected stray light (reflected SL), or only stable stray light (stable SL). The bottom line of each label indexes the case of applied correction (corr); which is (no) stray light (SL) corrected, only reflected stray light corrected (reflected), and the impact of knowledge of the out-of-field stray light (no/spectral/full OFSL) in the stray light correction.



## 7 Impact of stray light on emission rate estimations

The primary objective of MAMAP2D-Light is to quantify GHG emission rates from point sources by exploiting the retrieved GHG anomaly maps. Here, the column noise of the anomaly maps, and therefore the impact of the stray light induced patterns, especially for the single columns, becomes important for the quality of the retrieved GHG emission rates.

The emission rates were retrieved using a mass balance approach and using the corresponding wind data (Krautwurst et al., 2024; Borchardt et al., 2025). This work focuses on the impact of the stray light on the retrieved emission rates, which means

that the error estimation in this paper solely includes the error due to stray light, and atmospheric uncertainties (e.g., wind speed uncertainty) are neglected. However, the wind values are chosen from real wind measurements for the analysis to get realistic values for the emission rates, but those emission rates are not meant to be compared with inventories or discussed regarding their environmental impact.

### 7.1 Emission rate estimates with error estimations


The emission rate $F$ of a trace gas was estimated with a mass balance approach similar to Krautwurst et al. (2024). Within the georeferenced concentration data, $n$ cross-sections are defined. For each cross-section, the emission rate $F_c$ is estimated as:

$$F_{cs} = f \cdot \sum_{j}^{m} u_j \cdot \cos\left(90° - \alpha_j\right) \cdot \Delta V_j \cdot \Delta x_j. \tag{9}$$

where $m$ is the number of ground scenes inside the plume area, $f$ converts the emission rate from $\mathrm{molec\,s^{-1}}$ to $\mathrm{t\,h^{-1}}$, $u_j$ and

$\alpha j$ are the wind speed and wind direction, $\Delta x_j$ is the distance element along a cross-section with a concentration enhancement $\Delta V_j$. The concentration enhancement is calculated by:

$$\Delta V_j = \frac{CH_{4,rel,j}}{\overline{CH_{4,rel,bg}}} \cdot CH_4^{abs\ col} \tag{10}$$

where the relative enhancement $CH_{4,rel,j}$ is normalized with the local relative background $\overline{CH_{4,rel,bg}}$ and scaled with the assumed background column of $CH_4$ $CH_4^{abs\ col}$ in $\mathrm{molec\,cm^{-2}}$ from the RTM. The relative background is estimated from the

local background around the plume.

The total emission rate of one flight leg $F_{leg}$ is calculated by averaging the emission rates of all cross-sections:

$$F_{leg} = \frac{\sum_{i=1}^{n} F_{cs,i}}{n} \tag{11}$$

The total error $\delta F_{total}$ of the emission rate estimation is derived by Krautwurst et al. (2024). In this work, only the error contributors, which are affected by the stray light correction, are considered, leading to a reduced equation:

$$\delta F_{total} = \sqrt{\left(\delta F_{css}^2 + \delta F_{atm}^2 + \delta F_{bg}^2\right)}. \tag{12}$$

$\delta F_{css}$ is the combined error of all $n$ single cross-sections of a single flight leg:

$$\delta F_{css} = \frac{\sqrt{\sum_{i=1}^{n} \delta F_{cs,i}^2}}{n}. \tag{13}$$





The error for a single cross-section $\delta F_{cs,i}$ is calculated from the column precision $\delta F_{col-pr}$. For a single cross-section, the random column precision is reduced by the number of enhanced ground scenes $m$:

$$\delta F_{cs,i} = \sqrt{\frac{\delta F_{col-pr,i}^2}{m}}. \tag{14}$$

Uncertainties of the measured plume due to atmospheric variabilities or turbulences are considered by $\delta F_{atm}$, which is calculated from the 1-sigma standard deviation (SD) from the calculated emission rates for all cross-sections in one flight leg by:

$$\delta F_{atm} = \frac{SD(F_{cs,i})}{\sqrt{n_{eff}}}, \tag{15}$$

where $n_{eff}$ is the number of temporal and spatial independent cross-sections. For the comparison, $n_{eff}$ is set to 1 for all cases since the stray light correction should have a neglectable impact on the correlation estimation.

The background error $\delta F_{bg}$ is estimated by the standard deviation of emission rate estimates, with variations of the background area up to $50\,\%$ from the initial background.

## 7.2 Retrieved $CH_4$ emission rates

The impact of the stray light correction on the retrieved $CH_4$ emission rates was analyzed based on two detected plumes from the Brady Road Landfill and the Prairie Green Landfill near the city of Winnipeg in Manitoba, Canada. However, to deal with realistic emission rate values, the wind speed was determined from historical wind data from the Winnipeg Airport GoC (2025). Further, the wind was assumed to be constant over the full boundary layer height, and the plume was assumed to be well mixed in the boundary layer, even in the near field. The detected $CH_4$ plumes are shown in Fig. 14 and 16, and the parameters for the emission rate retrieval are shown in Tab. F1.

The results in presence and absence of applied proxy and stray light corrections for the two landfills are shown in Fig. 14 and 16, and the resulting emission rate estimates in these four cases are shown in Fig. 15 and 17 with the relevant error contributors as described in Sect. 7.1. In all cases, the total error is dominated by the error for atmospheric variability or turbulence $\delta F_{atm}$. However, this error is likely overestimated since it is assumed, that all cross-sections in the swath are correlated ($n_{eff} = 1$ in Eq. 15).

For both landfills, the emission rates derived from the proxy-corrected column data and the stray-light-corrected single $CH_4$-column data are relatively close and within the error resulting from the background definition $\delta F_{bg}$. However, due to the light-path-correction within the proxy-method, compensating e.g. light path elongations due to aerosol scattering, and with the assumption of $CH_4$ only plumes[4], the proxy-corrected data is more reliable. As in Sect. 6.3.1, the column noise differs slightly and causes small variations in the determined emission rates and corresponding errors for the three corrected cases.

For the Brady Road Landfill in Fig. 15, the non-stray-light corrected single $CH_4$ column differs significantly from the other

---

[4]While also $CO_2$ is emitted from landfills, the single column $CO_2$ data indicate no emission strong enough to influence the proxy and mask parts of the $CH_4$ emissions



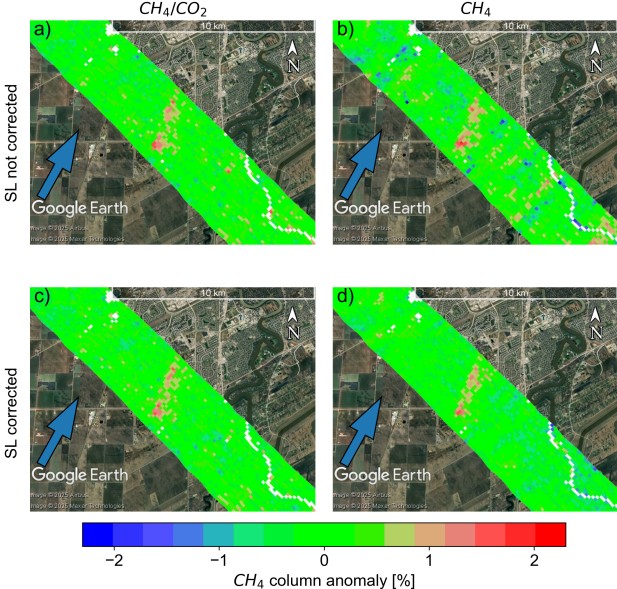

**Figure 14.** Retrieved $CH_4$ anomalies at the Brady Road Landfill. The results with applied proxy correction ($CH_4/CO_2$) are shown in the left column, and the single $CH_4$ column results are shown in the right column. Non-stray-light-corrected results are shown in the top row, and stray-light-corrected results in the bottom row. The blue arrow marks the wind direction. The map underneath is provided by Google Earth (Image © Airbus 2025, © Maxar Technologies 2025).

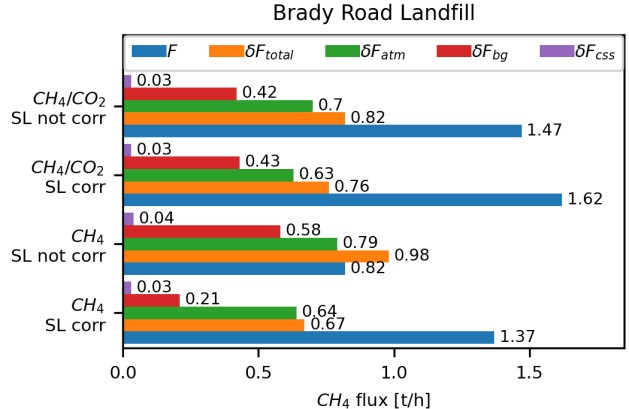

**Figure 15.** Retrieved $CH_4$ emission rates (F, blue) for the scene shown in Fig. 15 for different cases of applied proxy ($CH_4/CO_2$) and stray light (SL) correction. The total error ($\delta F_{total}$, orange) is calculated from the different single error contributors due to turbulences $\delta F_{atm}$ (green), $\delta F_{bg}$ (red), and $\delta F_{css}$ (purple), which are described in Sect. 7.1. The shown emission rates are not supposed to be compared with emission inventories.



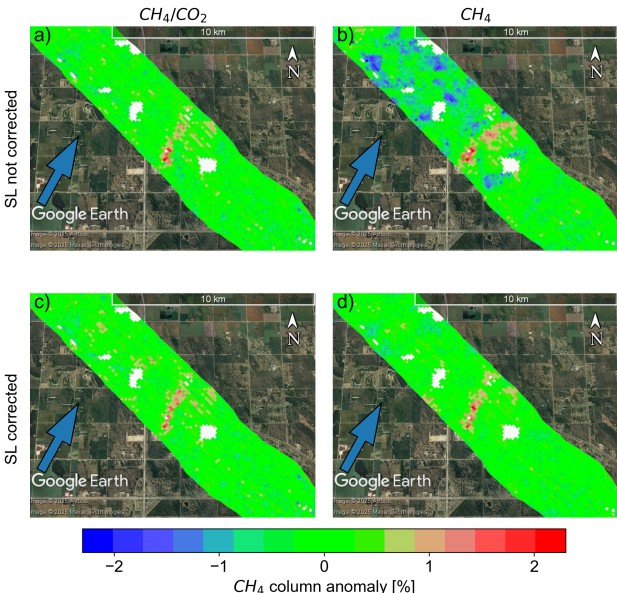

**Figure 16.** Similar to Fig. 14 but for the Prairie Green Landfill. The map underneath is provided by Google Earth (Image © Airbus 2025, © Maxar Technologies 2025).

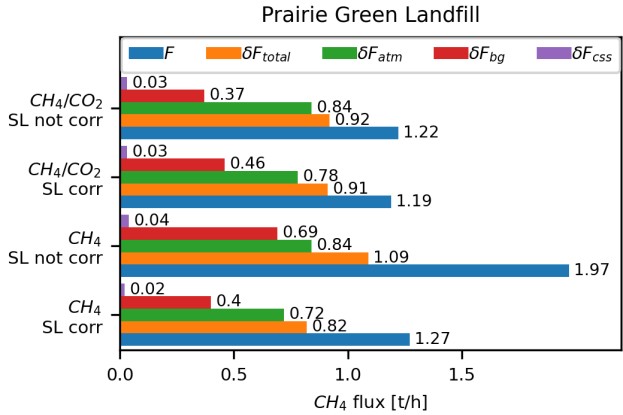

**Figure 17.** Similar to Fig. 15 but for the scene shown in Fig. 16.

cases. The derived $CH_4$ emission rate is $\sim 55\,\%$ lower than the mean of the other cases. The concentration map for the Brady Road Landfill in Fig. 14 (b), shows strong variations of the background column due to small scale (in the region of the MAMAP2D-Light ground scene size) inhomogeneous surface reflectance; these small variations seem to have no significant impact on the error from the background definition $\delta F_{bg}$. However, the resulting error from the standard deviation of the emission rates from the single cross-sections $\delta F_{atm}$ is slightly increased. The emission rate for the non-stray-light corrected



$CH_4$-column for the Prairie Green Landfill, in Fig. 17, is increased by $\sim 65\,\%$ compared to the mean of the other cases, which can be explained in the corresponding concentration map in Fig. 16 (b), where the plume signal is displaced compared to plumes of the other cases. This leads to the conclusion that a stray-light-introduced pattern is causing an additional false plume
signal. The overall column anomalies in the background are disturbed by patches of decreased column anomalies, which are related to inhomogeneous surface reflectance scenes due to agricultural land use covered by multiple adjacent MAMAP2D-Light ground scenes. The error estimates are increased for the background error $\delta F_{bg}$, whereas the relative standard deviation of the estimated emission rate for the single cross sections $\delta F_{atm}$ is relatively constant.

## 8 Stray light reduction by hardware improvement

During the CoMet 2.0 mission, an adjustable slit, shown in Fig. G1 (a), with uncoated blades, was installed in front of the ferrule. The edge of the blades was visible as an area where the stable stray light was decreased. By exchanging the adjustable slit with a black-coated fixed-width slit aperture, shown in Fig. G1 (b) and (c), the sharp ghost vanished completely, and the stable stray light cone was decreased significantly, as shown in Fig. 18.

This reduction of stray light was determined by single spot measurements, where the stray light cone is fully imaged at the
detector, as illustrated in Fig. 18 (a) and (c). Both the non-stray-light-related horizontal laser artifact and the vertical line originating from in front of the fiber bundle were masked, see Sect. 5. Furthermore, a noise threshold was applied, calculated by the mean and the standard deviation in an illumination-free area ($\overline{Signal_{dark}}$). All values below the noise threshold were set to zero. The prepared image was normalized to the integrated signal of the entire FPA. The stray light was separated from the origin spot with a threshold value relative to the maximum intensity of the frame. The spot-size threshold is the average relative
minimum value of the instrumental response functions ($\overline{ISRF_{min}}$), as described in Appendix H2. The relative stray light is the ratio of the integrated stray light to the total integrated signal.

The uncertainty was calculated by the quadratic addition of the uncertainties for the spot size, the noise threshold, and the size and position of the horizontal and vertical masks. The single uncertainties were calculated by disturbing the variables by the values stated in Tab. 1.

The stray light in the Comet 2.0 configuration, shown in Fig. 18 (a), was $(3.9 \pm 0.32)\,\%$. The stray light after the hardware

**Table 1.** Parameters for stray light quantification and absolute uncertainties for the relative stray light of MAMAP2D-Light in the Comet 2.0 (Comet) configuration and for the post-campaign hardware improvement (HWI). The total error is calculated by a quadratic addition of the single components. $\sigma$ represents the standard deviation.

| Uncertainty source | start value | disturbance | $\Delta_{Comet}$ | $\Delta_{HWI}$ |
|---|---|---|---|---|
| Spot size threshold | $\overline{ISRF_{min}}$ | $\pm 3\sigma_{ISRF_{min}}$ | $\pm 0.104\,\%$ | $\pm 0.144\,\%$ |
| Noise threshold | $\overline{Signal_{dark}} + 3\sigma_{Signal_{dark}}$ | $\pm \sigma_{Signal_{dark}}$ | $\pm 0.293\,\%$ | $\pm 0.233\,\%$ |
| Mask size and position | defined by hand | $\pm 1\,$pixel | $\pm 0.076\,\%$ | $\pm 0.072\,\%$ |


improvement, as depicted in Fig. 18 (c), was $(1.0 \pm 0.28)\,\%$. The total reduction in stray light from the hardware improvement





is approximately $(74 \pm 10)\,\%$.

The uncertainties in Table 1 show a primary influence of the noise threshold on the total uncertainty. This is directly correlated to the weak stray light signal, particularly in the case of the stray light measurement with the hardware improvement.


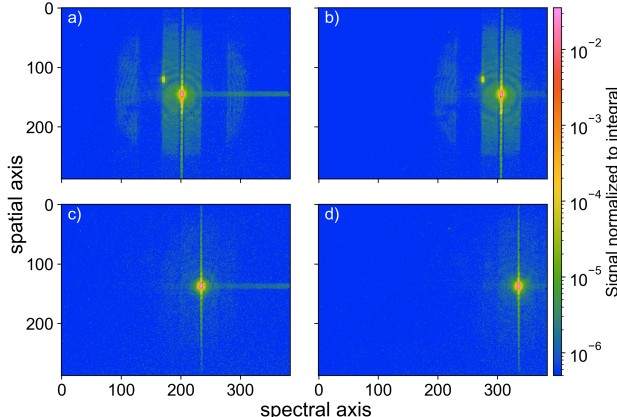

**Figure 18.** Different spectral spots for stray light characterization. (a) and (b) images show two spots similar to Fig. 5 without hardware optimization. (c) and (d) images show measurement results after a blackened slit aperture was inserted in front of the spectrometer's entrance slit. The sharp ghost vanishes nearly completely, and the stable stray light cone is decreased significantly. The images (a) + (c) are measured at $\sim 1628\,\text{nm}$ and (b) + (d) at $\sim 1661\,\text{nm}$. The spectral offset is related to a turned grating during a readjustment of the MAMAP2D-Light system.

## 9 Conclusions

Stray light is causing a varying additive error signal in the spectra measured with the push-broom imaging spectrometer MAMAP2D-Light. In the WFM-DOAS retrieval, this varying error leads to pseudo-noise in the retrieved GHG columns. Based on stray light characterization measurements, a stray light correction was applied to a stray-light-contaminated cam-

paign dataset. This allowed insights into the impact of stray light in the whole processing chain, from measured spectra to retrieved GHG fluxes.

The stray light characterization measurements were performed with a relatively low-cost measurement setup, with a Metcalf/Littman laser as a tunable monochromatic light source with insufficient side-mode suppression. To apply the measured data in the stray light correction, occurring detector effects and occurring side modes from the laser system were corrected or

interpolated.

The stray light correction showed a substantial improvement in the column precision of the retrieved single $CO_2$ and $CH_4$ column concentration anomalies. Within the $CH_4/CO_2$ proxy-corrected data, the stray light correction had an impact on single scenes with high-intensity contrast and strong varying spectral surface reflectance. However, in the majority of the scenes, the



proxy method was able to correct stray-light-related errors. Analyses on artificial spectra showed that the column precision for
stray-light-corrected or proxy-corrected data is limited by random noise sources.

Within the flux estimates for two measured landfill $CH_4$ emissions, the stray-light- and the proxy-corrected concentrations
provided similar emission rates. However, the non-stray-light and non-proxy corrected data show error patterns, which are
highly affecting the flux estimates.

In MAMAP2D-Light, stray light correction is a crucial step to eliminate the need for the proxy method and, therefore, to
differentiate between individual components in mixed plumes. However, the proxy method is also utilized for light path and
other instrumental corrections, which must be taken into account during data interpretation. The proxy method is only effective
against stray light if both retrieved concentrations are measured in the same optical path and the trace-gas bands are closely
spaced, as is the case for $CH_4$ and $CO_2$ in the $1.6\,\mu m$ channel. However, for passive remote sensing instruments that offer additional spectral channels, e.g., NIR or SWIR-2, reducing and correcting stray light is essential to retrieve reliable atmospheric
data.

The impact of stray light was analyzed based on the WFM-DOAS retrieval. For other retrieval algorithms the impact of stray
light might be different, since there are retrieval algorithms like FOCAL (Fast atmOspheric traCe gAs retrieval) (Reuter et al.,
2017a, b), UoL-FP (The University of Leicester Full Physics) (Cogan et al., 2012) and the $CH_4$ retrieval for the MethaneAIR
instrument (Chan Miller et al., 2024), which consider an additive offset in their atmospheric state vector.

*Data availability.* All level 1 and level 2 data can be provided by the corresponding authors upon request.





## Appendix A: Stray light terminology

For this work, the stray light terminology is adapted from Fest (2013). Stray light is a collective term for unwanted redirected radiation that reaches the focal plane of an optical instrument. It occurs in all optical systems and can only be mitigated by
design and manufacturing processes or corrected based on exact calibration measurements. The types of stray light can be described by their physical origin mechanisms.

*Ghost reflections* occur due to reflections and refraction, whose light paths obey Snell's law or the grating equation. Depending on the divergence of the resulting light path, ghost reflections can occur as sharply focused images.

*Scatter stray light* results from scattering on rough or particulate contaminated surfaces; since there are no perfectly smooth
surfaces, all surfaces scatter light. Scatter stray light is described by the Bidirectional Scatter Distribution Function (BSDF), which is often referred to in terms of the scatter direction as the Bidirectional Reflection Distribution Function (BRDF) or the Bidirectional Transmittance Distribution Function (BTDF). The most common way to describe the BSDF of one or a series of surfaces is the Harvey model (described, e.g., in Peterson (2004) and Fest (2013)), which uses two to three parameters to describe a surface. Depending on the accuracy of the analytical model, it is rather complex to describe those surface parameters.
*Internal stray light*, also called thermal background, originates from the thermal emission of the optical system itself. This becomes crucial in infrared applications, where the thermal radiation of the instrument results in stray light at the focal plane. The internal stray light is corrected by subtracting a background measurement, which is recorded with a turned-off or blocked intended light source.

*Out-of-field stray light* originates from sources outside of the intended light path. However, the resulting stray light reaches the
focal plane and contaminates the measured irradiances at the focal plane.

A surface is called *critical* if the detector sees it; this counts for optical elements like lenses and housing surfaces. A surface illuminated by stray light is called an *illuminated surface*.

The stray light paths are characterized by their order. The intended light path is the zeroth order. Any stray light event adds a new unintended light path, in which the order is increased. The intensity of the stray light decreases with each stray light event,
resulting in higher-order stray light usually being of lower intensity.

## Appendix B: Laser side modes

The stray light measurements in Fig. 5 showed contamination in spectral direction, which was related to the used laser. The measurements in the of Fig. 5 (a) and (c) were done at $\sim 1628\,\text{nm}$. With the maximum peak value and the maximum of the
row-wise median for the area from the horizontal pixel 315 to 384, a side-mode suppression of $43.0\,\text{dB}$ for the top and $43.3\,\text{dB}$ for the bottom measurement was determined. The side-mode suppression determined by the manufacturer at $1625\,\text{nm}$ was $54.96\,\text{dB}$, see Fig. B1. However, MAMAP2D-Light has a coarser spectral resolution compared to the Laser characterization measurements; convolving the curve in Fig. B1 with the ISRF of MAMAP2D-Light led to a side-mode suppression of $42.92\,\text{dB}$ (red curve).



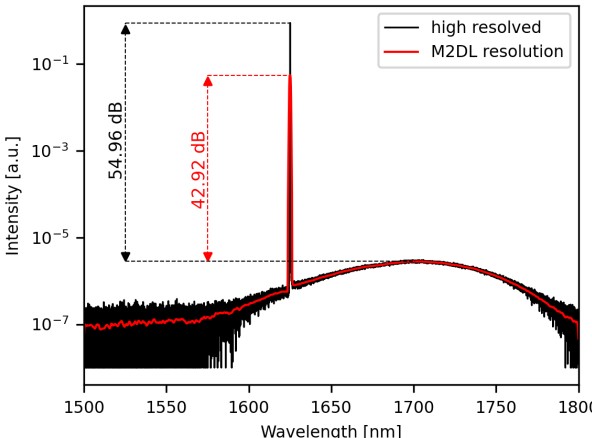

**Figure B1.** Spectrum of the laser signal at $1625\,\mathrm{nm}$ recorded with a spectral resolution of $0.05\,\mathrm{nm}$ by the manufacturer. The side-mode peak is at $\approx 1700\,\mathrm{nm}$ and is $54.96\,\mathrm{dB}$ suppressed compared to the laser main peak. After cnonvolving the high resolved spectrum with the MAMAP2D (M2D) ISRF the value of side-mode-supression is reduced to $42.92\,\mathrm{dB}$

## Appendix C: Detector Dark Signal Shift

During the stray light characterization measurements of the MAMAP2D-Light system, a reproducible detector effect occurred. In some areas, the measured signal of a partially illuminated frame was lower than the measured dark signal. This led to a negative shift in the dark current corrected measurements. Similar effects defined as pedestal shift were also observed by Chapman et al. (2019) for the Next Generation Airborne Visible Infrared Spectrometer (AVIRIS-NG) system, where it is corrected using non-illuminated reference pixels covered by a mask at the edges of the detector.

For MAMAP2D-Light, the effect caused a negative signal horizontal to the initially illuminated and by blooming widened spot, see C1 (b). The size of the illuminated spot during the stray light characterization measurements was three spectral pixels by ten spatial pixels. The laser power was constant for each measurement, while the detector's exposure times were increased from $10\,\mathrm{ms}$ to $3000\,\mathrm{ms}$.

No pixel was saturated for exposure times up to $20\,\mathrm{ms}$. By increasing the exposure time from $10\,\mathrm{ms}$ to $20\,\mathrm{ms}$, the negative offset was also increased (Fig. C1 (a)). For exposure times larger than $20\,\mathrm{ms}$, the pixels started saturating, and blooming occurred; the negative offset increased also. The spatial distribution of $35\,\mathrm{pixel}$ for $3000\,\mathrm{ms}$ correlated with the spatial extent of the blooming-related saturation area. However, the spatial extent was constant for lower exposure times, even for the measurements without saturation.

The offset has to be correlated to the collected charges in the read-out electronics since blooming-only signals influence it. Further dependencies need to be characterized, which was out of scope for this work. However, separating the offset from other sources of additive offsets, like stray light, is challenging. The negative offset is corrected by determining an illumination-free area. Within that area, the row-wise mean of the illumination-free area (Fig. C1 (b) is subtracted from the measured frame.



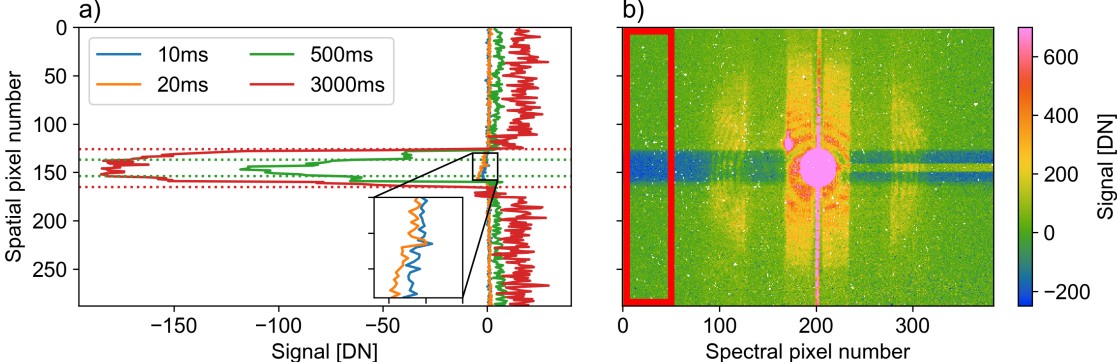

**Figure C1.** Negative offset after dark signal correction. (a) row-wise mean in the non-illuminated area from spectral pixel 0 to 50 (red rectangle in (b)), for different exposure times. Dotted lines show spatial borders of corresponding saturated areas for $500\,\mathrm{ms}$ and $3000\,\mathrm{ms}$. (b) Dark current corrected signal at $3000\,\mathrm{ms}$. Stripe of negative signal in rows of saturated pixels from the spot signal. The area left from horizontal pixel 50 in the red frame is non-illuminated, used for (a). The noise for the measurement performed at an exposure time of $3000\,\mathrm{ms}$ is increased due to additional shot noise of thermal photons in the dark current measurement.

## Appendix D: Stray light simulation

The impact of the measured stray light on the retrieved column anomalies was analyzed by contaminating synthetic calculated spectra with the corresponding stray light signal from the total frame. This offered the advantage of analyzing the introduced stray light error separately in order of its origin and evaluating the correction constraints, e.g., knowledge of the OFSL and other error contributions like the detector's read-out noise and the shot noise.

### D1 Generating synthetic frames

The analysis was based on synthetic spectra, as they would be measured by MAMAP2D-Light, with known atmospheric properties. For simplification, the same RTM and instrumental spectral properties, ISRF, and wavelength grid were used for the synthetic spectra and the retrieval. Therefore, retrieving the synthetic spectra without any error signal contamination results in profile scaling factors (PSFs) equal to 1. To consider the spectral OFSL, the RTM was calculated for the wider wavelength range of $1500\,\mathrm{nm}$ - $1750\,\mathrm{nm}$ compared to the wavelength range of approximately $1559\,\mathrm{nm}$ - $1689\,\mathrm{nm}$ imaged at the MAMAP2D-

Light detector. The range was chosen due to the size of the stable stray light kernel in Fig. 6. The RTM was convolved with the measured ISRF of the instrument.

The stray light signal per pixel depends on the surrounding signal. Therefore, two-dimensional frames with spectral and spatial directions were generated. The signal levels for the synthetic frames were determined from real measured frames. The measured frames were corrected for the background signal, flat-field corrected, and rescaled with the median of the flat-field correction

frame. The rescaling was applied to keep a signal level within the range of the detector's $16\,\mathrm{bit}$ output values.

For generating the synthetic frame, the slowly varying curve related to the spectral surface reflectance and aerosol scenario




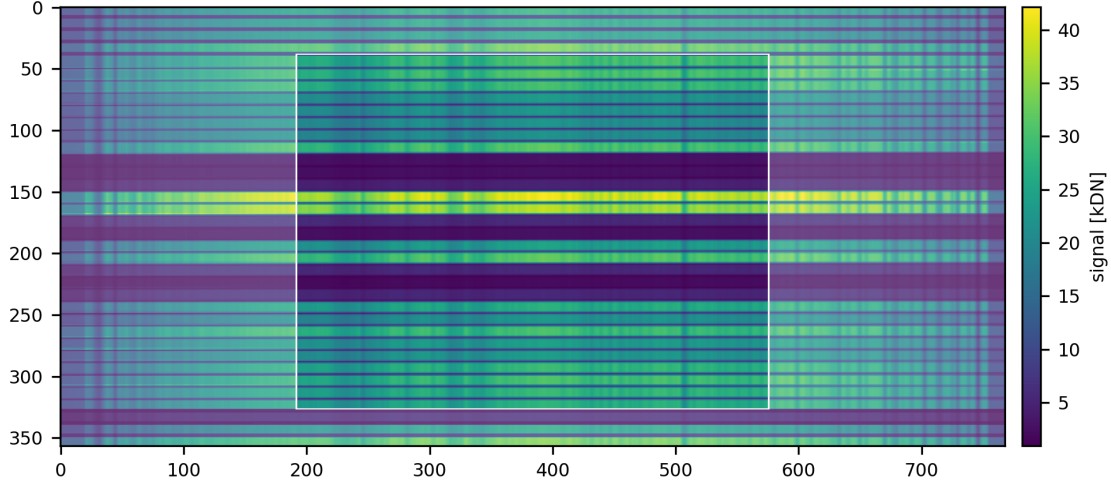

**Figure D1.** A generated synthetic frame with the fiber structure of MAMAP2D-Light. The highlighted middle represents the dimensions of the MAMAP2D-Light detector. The pale surrounding is the out-of-field signal, which contributes to the corresponding stray light error signal based on the straylight kernel.

of the spectra had to be be taken into account (Fig. D2). In the WFM-DOAS retrieval, this was done by fitting a low-order polynomial $P_\lambda(\mathbf{a})$ (Sect. 3). For the synthetic frames, the calculated RTM was first flattened and then rescaled with the curve of the real measured spectra. The slowly varying curve in the radiance of the RTM and the real spectra was determined by

fitting $exp(P_\lambda(\mathbf{a}))$ to the mean value of several areas with minor absorption-band features in the spectra, see Tab. D1. For the calculated RTM, all five areas were used for the fit. The measured spectra were limited to the spectral range of MAMAP2D-Light. Therefore, only the values from area 2 to area 4 could be used. The wavelength for the measured data was used from the wavelength grid described in Appendix H1. The retrieval fit window was chosen to keep the requirement for the retrieved PSFs equal to one.

  Within a fiber core, the measured spectra are binned spatially to form a single spectrum. The corresponding synthetic spectrum

**Table D1.** Wavelength ranges of low absorption-band features in the absorption spectra.

| Area | Wavelength [nm] |
|------|-----------------|
| 1 | 1506.3 - 1506.6 |
| 2 | 1584.6 - 1588.2 |
| 3 | 1620.5 - 1623.0 |
| 4 | 1681.0 - 1683.0 |
| 5 | 1717.4 - 1719.6 |


was repeated to the spatial extent of the fiber core. The areas of cladding also have a significant signal level; therefore, for each



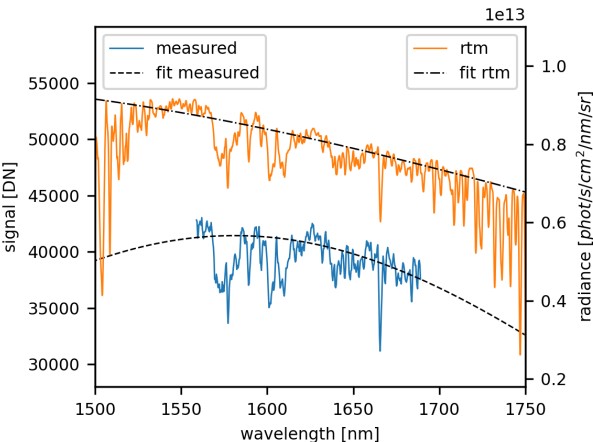

**Figure D2.** A measured spectrum and a calculated model spectrum. For both spectra, a least squares fit was applied.

cladding row, an individual synthetic spectrum was generated, as done for the binned spectra. The spatial out-of-field signal was considered by adding seven extra fibers, four on top and three at the bottom of the frame. The fiber core size of the added fibers was 7 pixels, and the cladding was 3 pixels in spatial direction. The signal level of the added fibers was randomly chosen
from the 29 real measured fibers. The signal level of the claddings between the added fibers was determined by the minimum signal of the two adjacent fibers times a conversion factor, which was the mean ratio of the signal in the cladding and the signal in the fiber cores of the flat-field correction measurement. A full synthetic frame is shown in Fig. D1.

The synthetic frames for the following analysis were generated from real measured frames of a flight leg whose parameters are shown in Tab. F1. This real measured data had already been used in Sect. 6.3.1. It consisted of approximately 1800 frames
recorded over agricultural and urban-dominated surfaces.

## D2    Synthetic frames with stray light contamination

The synthetic frames were contaminated artificially with the corresponding stray light signal. The stray light was generated by the inverse correction processes described in Sect. 5. The ideal frame $\mathbf{F}$ was contaminated with the stable stray light using the following equation, which considered the redistribution of the stray light, from Tol et al. (2018):

$$\mathbf{J}_0 = \left( 1 - \sum_{k,l} (\mathbf{K}_{far})_{k,l} \right) \cdot \mathbf{F} + \mathbf{K}_{far} * \mathbf{F}, \tag{D1}$$

with $\mathbf{J}_0$ as the measured contaminated frame and $\mathbf{K}_{far}$ the far-field of the $\mathbf{K}_{stable}$, see Sect. 5.1.

The stray light resulting from the sharp ghost reflection was considered as described in Eq. 5. For the synthetic frames, the two-dimensional fit for $\mathbf{E}_{refl}$, shown in Fig. 7, was expanded to the full frame shown in Fig. D1.

Stray light is causing a pseudo-noise in the retrieved column anomalies. However, there is also random noise, which is intro-
duced by the shot noise $N_{phot}$ of the measured photons and the read-out noise $N_{ro}$ of the detector electronics, which had to



be considered in the synthetic spectra, too. The shot noise is introduced by the intended signal photons as well as from the unwanted thermal photons in the background correction and was calculated by the signal in electrons $S_{el}$ with $N_{phot} = \sqrt{S_{el}}$. The thermal signal was estimated from the background measurements dependent on the exposure time. The slope of a first-order polynomial fit of a pixel value per exposure time was used as the background signal introduced by thermal photons. The pixel values for the thermal and the intended signal were converted with the fraction of the detector's full-well-capacity ($0.34\,\mathrm{Me^-}$) and the corresponding bit-depth ($16\,\mathrm{bit}$) to the signal in electrons. The total noise $N_{full}$ is calculated by:

$$N_{full} = \sqrt{S_{el,intended} + S_{el,thermal} + N_{ro}^2}. \tag{D2}$$

The synthetic frames were contaminated by a noise frame, containing a noise value for each frame pixel. The value for each pixel was a random normal distributed value with a standard deviation of the calculated noise converted into binary units. Two noise frames are generated, one for the non-stray-light-contaminated synthetic frame and one for the full-stray-light-contaminated synthetic frame.

The column anomalies were retrieved from the synthetic spectra as described in Sect. 3 for the real measured spectra. In Fig. D3, the resulting $CO_2$, $CH_4$ and the proxy corrected ($CH_4/CO_2$) columns are compared. The overall column noise for all three columns of the real and the simulated data is very similar. Differences are expected due to several factors, namely the not perfectly matched fitting of the low-order polynomial to adapt the simulated spectra to the measured spectra, see Fig. D2, pseudo-noise introduced by a more complex structure than a low-order polynomial, spectral surface reflectance, the unknown real out-of-field signal, and residual uncertainties in the measured stray light kernels. However, a Pearson correlation factor of $0.80$ in the $CO_2$ and $0.77$ in the $CH_4$ column was calculated in the direct comparisons. The correlation factor is close to zero after the proxy correction.

### D3 Spectral out-of-field stray light extrapolation

In the stray light correction in Sect. 5, the measured spectra were extrapolated using a $3^{rd}$-order polynomial to scale an extended RTM. In reality, the signal beside the FPA, and therefore the OFSL, is unknown. However, the simulated spectra provide the opportunity to apply the extrapolation with different orders ($1^{st}$ to $4^{th}$) of the polynomial. In Fig. D4, the standard deviations of the retrieved column anomalies are compared to the case where the spectral OFSL is fully known and with no considered OFSL, similar to Fig. 13. However, due to the presence of spatial OFSL, only data from the middle fiber is analyzed. The $CH_4$-band is close to the border of the FPA, see Fig. 10. Therefore, it is expected that the spectral OFSL has the biggest impact on the $CH_4$-band. The introduced pseudo noise is decreased by $\sim 90\,\%$ to $< 0.018\,\%$ by applying any extrapolation. In this example, the higher-order extrapolations in the fit meet well with the initial out-of-field light. However, in reality, the spectral intensity distribution depends mainly on the spectral surface reflectance, which is usually not fully covered by a low-order polynomial for a wider spectral range. Further, the higher-order fits for extrapolation can highly under- or overestimate the signal level in the areas for extrapolation.




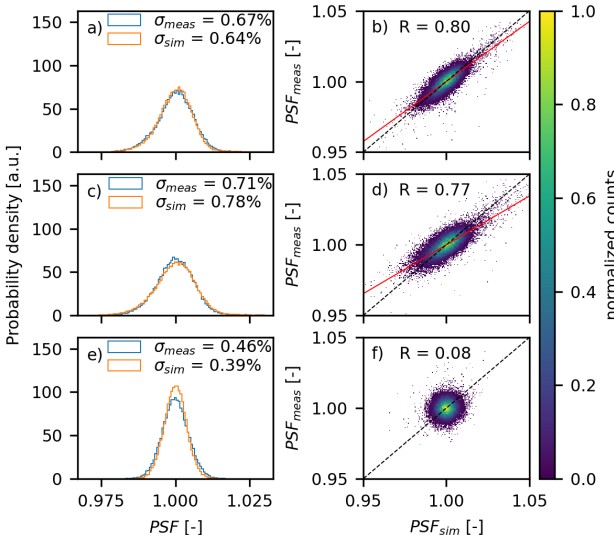

**Figure D3.** Retrieved profile scaling factors (PSF) for the $CO_2$, $CH_4$ and the proxy corrected ($CH_4/CO_2$) columns for simulated and real measured data. The simulated data is artificially contaminated with stray light and random noise. The left column shows histograms representing the column noise. The right column shows the correlation of the simulated and real measured column anomalies. The red line shows a linear fit through the data, and the dashed black line marks a Pearson correlation coefficient (R) of 1.

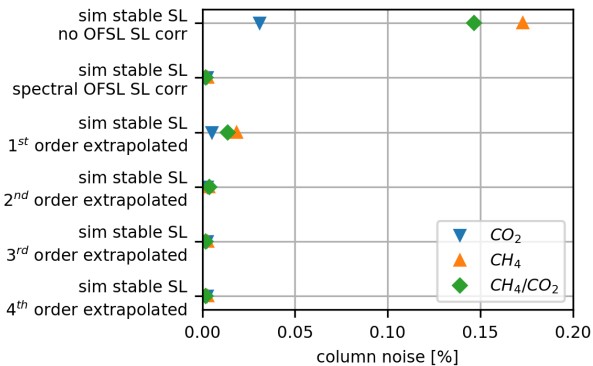

**Figure D4.** Single read-out column noise for retrieved column anomalies for the $CO_2$ (blue triangle), $CH_4$ (orange triangle) and the proxy corrected ($CH_4/CO_2$) (green diamond) column, for different cases (y-axis). The first two cases are from Fig. 13 and only the middle fiber (number 14) is considered.

## Appendix E: Stable Kernel optimization

The measured stable stray light in Fig. 5 in Sect. 4 shows contamination of the used laser, which is related to insufficient side-mode suppression. For the stable kernel creation, shown in Fig. 6, those areas were corrected. Therefore, the horizontal signal



of the raw stable kernel was masked. Afterward, the stable kernel was defined in several sections, depending on the surface type in the slit object plane. The stray light in the outer regions from spectral pixels $75 - 116$ and $267 - 303$ was reflected from a steel surface from the adjustable slits blades. Within the areas from spectral pixel $117 - 158$ and $225 - 268$, the light was

reflected from the blade edges; due to the angle, the light was not reflected into the optical path of the useful signal. The area from the spectral pixel $159 - 224$ was reflected from the aluminum ferrule and the aligned fibers. The stray light signal from the relay optics used for single fiber illumination can not be separated into instrumental and non-instrumental stray light, and therefore, the fiber area was set to zero.

The signal of each region was remapped into the polar coordinate space using "warp_polar" function of the "skimage" Python

package (version 0.18.1). The rows of the resulting 2D image represented the rotation angles, and the columns the radii. The signal in dependency of the radius for all angles is shown in Fig. E1. At this point, a generalized scattering theory, like the Harvey scatter model described by Peterson (2004), could be fitted. However, due to the signal steps in the aluminum and steel areas, the fitting did not describe the kernel sufficiently. Therefore, the median value along the rotation angle axis was used as a numerical function to describe the scattering. By rotating the function, the observational gaps were filled.

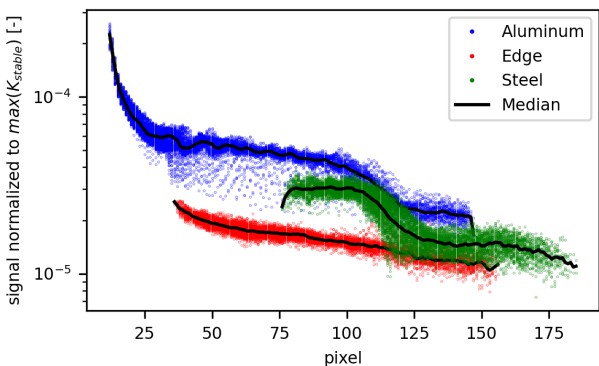

**Figure E1.** Signal of the scattering surfaces from the stable kernel in order of the radius after a polar coordinate transform.



## Appendix F: Parameters for WFM-DOAS and flux retrieval

**Table F1.** Parameters for RTM simulation, WFM-DOAS and flux retrieval for landfill scene

| | |
|---|---|
| Date | 11.09.2022 |
| Time | 15:30 - 16:00 UTC |
| Wavelength | 1500 nm - 1750 nm |
| Wavelength resolution | 0.01 nm |
| Flight altitude | 29000 feet |
| Background $CH_4$ | 1906 ppb |
| Background $CO_2$ | 413.7 ppm |
| Sun zenith angle | 55.7° |
| Surface evaluation | 172 m - 305 m |
| Albedo | 0.20 |
| Aerosol scenario | urban |
| Wind speed | 5.3 $ms^{-1}$ (GoC, 2025) |
| Wind direction | 209° |
| Fit window | 1575 nm - 1677.5 nm |
| Mean $c_f$ $CH_4$ | 0.80 |
| Mean $c_f$ $CO_2$ | 0.76 |
| Spatial resolution | $\sim 120 \times 120\,m^2$ |
| Plume area | 1.5 km from source |
| Background area | 2 km from plume area |





**Table F2.** Parameters for RTM simulation and WFM-DOAS for oil sand scene

| | |
|---|---|
| Wavelength | 1555 nm - 1730 nm |
| Wavelength resolution | 0.01 nm |
| Flight altitude | 26000 feet |
| Background $CH_4$ | 1894 ppb |
| Background $CO_2$ | 411.7 ppm |
| Sun zenith angle | 46.3° |
| Surface evaluation | $185 - 839$ m |
| Albedo | 0.20 |
| Aerosol scenario | urban |
| Fit window | 1575 nm - 1677.5 nm |
| Mean $c_f$ $CH_4$ | 0.78 |
| Mean $c_f$ $CO_2$ | 0.75 |
| Spatial resolution | $\sim 120 \times 120$ m$^2$ |

**Appendix G:  Entrance slit exchange**

During the CoMet 2.0 mission, the adjustable slit shown in Fig. G1 (a) with uncoated blades was installed in the spectrometer in front of the entrance fiber ferrule, in Fig. 1. The variable slit was exchanged with a fixed 200 µm slit (THORLABS S200ULK) consisting of blackened stainless steel, shown in Fig. G1 (b) and (c). The slit was wider than the fiber and is therefore acting as an additional aperture, as blackening the ferrule was not feasible. The slit was glued on an anodized aluminum support. The side that shows in the direction of the optics is painted with NEXTEL Velvet Coating 811-21.

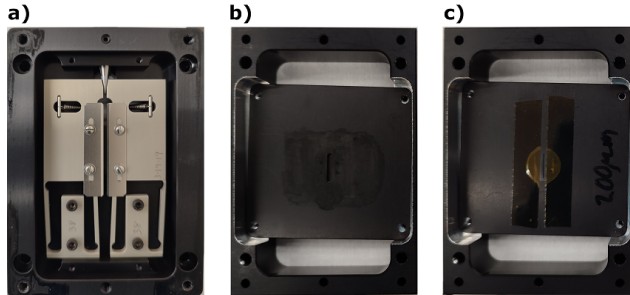

**Figure G1.** (a) adjustable slit, which was assembled during the CoMet 2.0 mission at the entrance fiber ferrule (ferrule 2, in Fig. 1) of the spectrometer. (b) fixed 200 µm slit as replacement for (a). (c) side of slit, that shows in the direction of the ferrule.





## Appendix H: Characterization measurements

In order to retrieve trace gas column enhancements from the measured spectra, it is necessary to have a very good character-
ization of the instrument. The wavelength calibration and the instrumental spectral response function (ISRF) were measured
with a Littman/Metcalf laser system (Lion System, by Sacher Germany, (Stry et al., 2006)), with a tunable wavelength from
$1600\,\text{nm} - 1750\,\text{nm}$ at a precision of $0.05\,\text{nm}$ and a power of $\sim 20\,\text{mW}$. The actual wavelength of the laser was observed with a
laser wavelength meter (671A, by Bristol) with an accuracy of $\pm 0.2\,\text{pm}$ at $1000\,\text{nm}$ for the NIR range from $520\,\text{nm} - 1700\,\text{nm}$.
Flat field corrections were applied to account for PRNU errors and losses from the grating. These corrections were performed
using a broadband Quartz Tungsten Halogen lamp as a WLS. To achieve a homogeneous illumination, all sources were con-
nected to an input port of an integrating sphere, with an inner diameter of 5.3" (IS6-C, by Ophir).

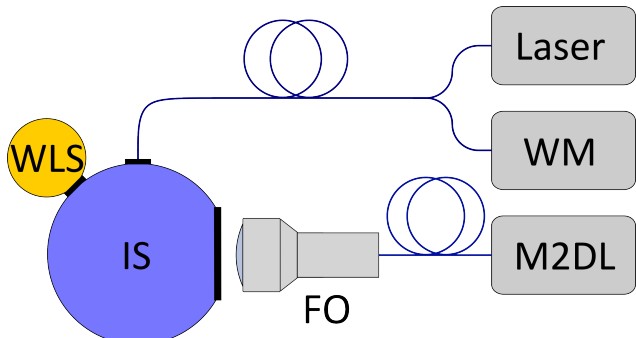

**Figure H1.** Optical setup for characterization measurements. Tuneable Littman/Metcalf laser observed with a Wavemeter (WM) and led via
a fiber to an integrating sphere (IS). The second input port of the integrating sphere is occupied by a white light source (WLS). The front
optic (FO) of MAMAP2D-Light (M2DL) views the output port of the sphere.

## H1 Wavelength calibration

The laser was adjusted to 18 different positions on the detector for the wavelength calibration. Due to the wavelength restriction
of the laser, which could not be tuned to wavelengths lower than $\sim 1600\,\text{nm}$, it was not possible to measure the corresponding
wavelength for the first $\sim 100$ pixels. In order to overcome this limitation and the additionally coarse measurement resolution,
the pixel-to-wavelength conversion was generated via a $2^{\text{nd}}$-order polynomial fit for the barycenter of each binned fiber. The
overall wavelength range covered was $\sim 1558\,\text{nm} - 1689\,\text{nm}$.

## H2 Instrumental spectral response function

The high accuracy of the wavelength meter ($\pm 0.2\,\text{pm}$ at $1000\,\text{nm}$) and the low step size of the tunable laser, permit the de-
termination of a high resolved ISRF even with a low spectral pixel sampling, based on the measurements performed by van
Hees et al. (2018) for the TROPOMI instrument. Therefore, the binned value of a spectral pixel was observed while tuning the



wavelength at a high resolution of $\sim 0.05\,\mathrm{nm}$. The full width at half maximum (FWHM) is about $1.00\,\mathrm{nm} - 1.08\,\mathrm{nm}$. Some fibers had a systematically increased ISRF FWHM, which was attributed to the presence of not perfectly aligned fibers in the

ferrule during manufacturing, see Fig. 2.

*Author contributions.* OH planned and performed the stray light characterization measurements; OH developed the correction algorithm; KG, with support of SK, JB developed the MAMAP2DL instrument; JB, OH developed the operating software; SK developed and maintained the retrieval algorithms, with support from JB and OH; OH retrieved and analyzed the data; H. Bovensmann acquired funding, contributed

to the HALO COMET 2.0 campaign planning and execution and the conceptional planning of the work; JPB and H. Boesch support the funding acquisition as well the conceptional planning of the work; OH wrote the draft version; All the authors critically assessed the results and corrected and improved.

*Competing interests.* The authors declare that they have no conflict of interest.

*Acknowledgements.* MAMAP2DL was in parts funded by BMBF within the project AIRSPACE (01LK1701B), by the University of Bremen

and by the State of Bremen (APF). HALO flights during the CoMet 2.0 Arctic mission have been supported by the State of Bremen, the Max Planck Society (MPG), and by the German Research Foundation (Deutsche Forschungsgemeinschaft, DFG) within the DFG Priority Program (SPP 1294) Atmospheric and Earth System Research with the Research Aircraft HALO (High Altitude and LOng Range Research Aircraft) under grant BO 1731/2-1. Part of the work on stray light characterization is supported by the ESA project CAMAP (ESA Contract No. 4000137866/22/NL/SD).

Writing the manuscript was supported by the AI tool Grammarly.

This research was also partly made possible by the team behind the CoMet 2.0 Arctic mission, especially the mission PI, Andreas Fix.



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
