# Peer review of "Impact of stray light on greenhouse gas concentration retrievals and emission estimates as observed with the passive airborne remote sensing imager MAMAP2D-Light"

_EGUsphere, 2025_

## Referee Comment (RC3)

**AMT REVIEW: Impact of stray light on greenhouse gas concentration retrievals and emission estimates as observed with the passive airborne remote sensing imager MAMAP2D-Light**

**General Summary**

Huhs et al. present measurements from a new airborne hyperspectral pushbroom spectrometer targeting CO2 and CH4 in the 1.6 micron band. During their maiden campaign they discover that their data suffers from substantial stray light contamination, caused by reflections from the entrance slit. They implement a correction using the TROPOMI stray light correction algorithm to salvage their observations.

The subject matter is within the scope of AMT. I believe it is of use to publish as a reference for future campaigns where the instrument is deployed. I think the organisation could be improved and the paper could be made more concise. Since the stray light correction method is not new, I would prefer the authors to shift the focus more towards the stray-light impacts on the retrieval, as weirdly the novel aspect could be the unintended experiment created by the adjustable slit. I am surprised by the reaction from the other reviewers - the paper is similar in quality to an average paper I have seen in AMT.

**Comments**

Pg 5 L 96: I believe the first reviewer stated that the slit configuration was different during the campaign than during the stray light measurements. But it does say later that the measurements later were taken using the CoMet campaign conditions (Pg 7, L162), which I assume means the slit was in its fully open position. If this is the case perhaps it should be stated explicitly to avoid confusion.

Pg 6 L132: The notation of Eq. 1,2 is a bit weird. First, since the error term appears on the RHS, I believe the LHS of Eq 1 actually corresponds to the authors $R^{mea}_\lambda$. Secondly the function $R^{mod}_\lambda$ appears on the LHS and RHS. The one on the right should be denoted something like $R^{RTM}_\lambda$ or something like that because it corresponds to the simulated spectrum from the radiative transfer simulation. Then $R^{mod}$ would be equation 1 without the error term, making the objective function in Equation 2 make sense. A normal way to write equation 2 would be

$$\arg\min_{\mathbf{x},\mathbf{c}} \ \|\ln R^{mea}_\lambda - \ln R^{mod}_\lambda(\mathbf{c},\mathbf{a})\|^2$$

Pg. 6 ln 136: Is there any particular reason why the measured radiances are log-transformed? In terms of radiance, the noise errors are approximately normally distributed, so the errors in this

case will be log-normally distributed. This means that the least squares estimate is no longer the maximum likelihood one.

Pg 9, L206 - It might be helpful to add some arrows to Figure 5 labeling the features due to the sources of stray light discussed in Section 4.1

Pg 12, L278 - There seems to be quite a bit more variability in the intensity of the ghost spot than what is modeled using the first order polynomial fit. It also looks real, in the sense that there is correlation between the calibration measurements. Is there a reason to believe this is an error that needs to be smoothed, or could some non-parametric method like radial basis function interpolation be used instead? It might be worth testing if something like this would improve the test case in Fig. 9.

Pg 14, L325: PSF is already being used for point spread function. The acronym for profile scaling factor should be changed to something else.

Pg 16, L372: As a general rule it is best to organize figures sequentially in the order that they are first referenced in the paper.

Pg 17, L380: The proxy uncertainty seems too close to the CH4 uncertainty in the stray light corrected case. If the error is completely uncorrelated with the CO2 column it should be ~0.44. 0.34 implies that the errors in CH4 and CO2 columns retrieved are almost perfectly correlated. That is really weird to me - is the instrument random noise really that low? It could be a typo as the value for the ratio looks to be about 0.44 in Fig. 13.

Pg 23: Perhaps Fig. 14/15 and Fig 16/17 can be combined into  two panel figures.

**Minor Corrections**

L53: buildung -> building

L80: "and thereby, allowing to operate the instrument at ambient temperature." -> and thereby allowing the instrument to operate at ambient temperature.

L107: adpated -> adapted

L 115 and L399:: Quadratic sounds weird in this context. I think saying "square" is fine.

L 402 Sect. D -> Appendix D.

---

## Author Comment (AC1)

Author comment on "Impact of stray light on greenhouse gas concentration retrievals and emission estimates as observed with the passive airborne remote sensing imager MAMAP2D-Light" by Oke Huhs et al., Anonymous Referee #2

We thank Referee #2 for the comments regarding our manuscript. In this document, we provide our reply to the comments. The original comments made by the referee are typeset in red. The authors' answers are typeset in black and *italic*. A revised version of the manuscript, with highlighted changes, will be sent with this reply. The changes in the revised manuscript regarding the referee's comments are highlighted. If not stated differently, the figure numbers and sections refer to those of the initial manuscript.

The manuscript assesses straylight of a custom-built imaging spectrometer and its influence on retrievals of GHG point source emissions using the recorded, straylight-contaminated spectral data. The general topic matches the scope of AMT, but I have substantial concerns accepting it for publication.

General Authors' answers, for clarification:

Stray light, which is common in all optical instruments, is a complex issue within the additive radiometric errors relevant for optical remote sensing instruments for trace gases (Veefkind, et al., 2012), especially for high accuracy and precision applications such as remote sensing of greenhouse gas distributions. To the best of our knowledge, the tolerable amount of stray light is so far only determined using simulated data, as for example done for the CO2M mission (Buchwitz, et al., 2019). The strict requirements for stray light present challenges in mitigating it within the optical design. Stray light characterisation and correction offer the potential to overcome hardware limitations (Clermont, C.Michel, Chouffart, & Zhao, 2024).

The straylight level of 5.6% reported in the manuscript for the MAMAP2D-Light low-cost airborne sensor in the CoMet 2.0 configuration is comparable to the amount of 4.4% correctable stray light in the TROPOMI SWIR channel, a high-tech satellite sensor (Tol, et al., 2018). The 2.4% estimated stray light for the MethaneAIR instrument (Staebell, et al., 2021) is also in this range and similar to the level of stray light in MAMAP2D-Light after the hardware improvement.

It has to be noted that the estimated stray light levels for the different MAMAP2D-Light configurations were reported lower in the initially submitted manuscript due to underestimating the treatment of noise. This has been improved and is described in detail in Sect. 6 of the revised manuscript. This change does not affect the conclusions of the paper.

The hardware improvement of the instrument configuration highlights a general issue affecting all spectrometers, specifically concerning reflective parts in the object plane of the spectrometer, e.g., an adjustable slit used during the CoMet 2.0 campaign (Acton Research, Model SPS-716-S1). For remote sensing of GHGs, the use of a fiber-based 2D-slit homogenizer leads to an issue regarding the ferrule in the object plane; a similar slit homogenizer is also planned for the CO2I instrument of the CO2M mission (Hummel, et al., 2022; Dussaux, et al., 2025).

The manuscript, to the best of our knowledge, is the first publication to address the impact of stray light contamination on greenhouse gas measurements based on measured solar backscattered absorption spectra, especially investigating the effect on the Level 2 CH4/CO2 proxy method and derived emissions. For TROPOMI, an analysis was conducted based on

simulated spectra with different surface types, including a retrieval method application, demonstrating the correction method's feasibility (Tol, et al., 2018). For the OCO-2 instrument, it is only assumed that stray light is the cause of an intensity offset in the measured spectra, which led to better-fitting XCO2 data compared to collocated TCCON data, after considering a spectrally constant offset in the retrieval algorithm (Wu, et al., 2018). For MethaneAIR, the stray light was only characterized within the radiometric calibration (Staebell, et al., 2021) and not further analyzed regarding the impact on the retrieved concentrations. The airborne remote sensing data from MAMAP2D-Light offers a smaller ground pixel size compared to satellite-based sensors with similar spectral properties, allowing for the distinction between concentration enhancements and falsely introduced (pseudo) noise. This is a clear benefit of the airborne remote sensing data used in the manuscript, as it enables the analysis of the impact of stray light on the retrieved Level 2 and Level 4 products.

The adapted and applied stray light correction presents a new opportunity for data analysis of the MAMAP2D-Light dataset, which was previously constrained by stray light contamination in the retrieved single CO2 and CH4 columns. Furthermore, the detection limit for the single-read-out data has decreased after the stray light correction. For a broader purpose, these results are relevant for instruments that exploit the 1.6 micron band for GHG observations and apply the proxy method: MethaneAIR (Chan Miller, et al., 2024), TANGO (Brenny, et al., 2023), GOSAT-GW (Tanimoto, et al., 2025), Sentinel-5 (Landgraf, Butz, Hasekamp, & aan de Brugh, 2019). To our knowledge, this is the first publication that demonstrates the robustness of the proxy method against stray light in measured spectra.

**Specific Authors' answers:**

The manuscript treats a stray light problem, which was caused by reflections at the entrance slit of the spectrometer. This is mentioned in Section 8 on page 25, at the end of the paper. At the beginning of page 5, the authors mention that this slit was not in use during the measurements. Consequently, the manuscript treats an issue that was caused by one specific instrument component, which was not necessary for operation in the first place.

The scope of the manuscript is described in the "General Authors' answers, for clarification". The issue is a large, stray-light-contaminated CoMet 2.0 campaign data set. A post-flight stray light correction reduced the stray light introduced errors. The required stray light characterization measurements were performed in the instrument's CoMet 2.0 campaign configuration, which included the adjustable slit aperture. From the characterization measurements, 62% of the stray light was attributed to a general issue with reflective parts in the spectrometer's object plane. Even after the hardware improvement, which involved replacing the adjustable slit aperture with a fixed blackened slit aperture, approximately 2% of the signal was correctable stray light. The scope of the manuscript is clarified in the introduction and conclusions of the revised version

The manuscript does not discuss to what extent the stray light is specific to this instrument component and how general it can be transferred to straylight problems in other instruments and their impact. The relevance of the remaining straylight on measurements taken after the hardware correction (i.e. the removal of the slit, which was not in use) is not discussed.

The whole Comet 2.0 campaign data set is contaminated with an amount of stray light, which is comparable to the correctable stray light in the TROPOMI SWIR channel. We clarified this

in the introduction and conclusions. A significant amount of stray light was attributed to reflective parts in the object plane built in during the CoMet 2.0 campaign. Namely, the non-coated adjustable slit aperture and the fiber ferrule, described in Sect. 4.1, which can be an issue for all spectrometers. The description of the origins of stray light in MAMAP2D-Light is generalized to optical processes occurring in optical instruments, like BRDF, BDTF, and reflections.

For future campaigns, we performed a hardware improvement by exchanging the adjustable slit with a blackened fixed slit aperture, which is done in Sect. 8. For the revised manuscript, we added an estimation of the impact of the stray light after the hardware improvement on the retrieved GHG concentrations in Sect. 8 (Sect. 6 in the revised manuscript). Furthermore, an estimation of the stray light level after the stray light correction is added.

Consequently, besides highlighting the importance of a careful spectrometer design, it remains unclear to me, what scientific question is addressed by the paper.

The scope of the manuscript is described in the "General Authors' answers, for clarification".

Without substantial revisions that address the points listed below, I must strongly recommend against publication of this manuscript in AMT.

(1) It should be clearly stated at the beginning of the manuscript (i) who is addressed by the paper, (ii) what is the general scientific question, and (iii) what is the added value of the new stray light correction compared to the existing procedures.

*The introduction, conclusions, Sect. 6.3.2 and Sect. 8 are adapted to match the suggestions:*

(i, ii) The manuscript provides insight into the impact of stray light throughout the entire processing chain, i.e., answering the question: "What is the impact of stray light on the processed L2 and L4 data based on in-field measured spectra?" This is a unique advantage of airborne remote sensing, as the smaller ground scene size allows for distinguishing between concentration enhancements and false-introduced pseudo-noise features, maintaining similar spectral properties to those of comparable satellite instruments. This is clarified in the introduction.

It was chosen to describe the origin of the stray light in general optical terms (BRDF, BDTF, reflections) to clarify the general issue of stray light in optical instruments. Furthermore, a minor focus is placed on the critical design of the object plane of the spectrometer when using a fiber-based 2D slit homogenizer, with a link to the setup of CO2M.

(iii) The added value of the manuscript is focused on the impact of stray light in the retrieved Level 2 and Level 4 data. It is stated in the manuscript that the procedure is not new. It was initially applied to the TROPOMI SWIR-Channel, based on a point spread correction for a 1D Spectrometer in 1932. Assuming a stable stray light kernel over the entire imaging plane is significantly less power-consuming than other methods for 2D spectrometers. This was mentioned in the introduction. However, to clarify the focus of the work on the retrieved data, the description of different stray light corrections is reduced in the revised version. However, there are still difficulties not considered in the TROPOMI stray light correction, such as out-of-band stray light (OBSL), detector effects due to saturation beyond typical usage during characterization, and the laser system used, all of which affect the stray light correction and

are therefore mentioned. Further, the impact of OBSL is analyzed in Sect. 4.4.2 of the revised manuscript.

(2) The cause of the stray light has to be clarified from the beginning of the manuscript. It should be evaluated how general or component-specific the treated stray light issue is and how general the proposed procedures can be applied to lower stray light levels.

The cause of stray light is described in Sect. 3.2 of the revised manuscript in generalized optical terms. The issue with reflective parts in the object plane of the spectrometer is clarified in the conclusions and mentioned in the abstract of the revised version. Furthermore, a link to other spectrometers that might be affected by this issue is established in the conclusions. The scope of the manuscript is the impact of stray light on the retrieved data based on the applied correction

The stability of the stray light constrains the convolution-based method. The ghost correction is valid for imaged ghosts.

As long as the stray light can be characterized (limited by the light source, intensity/dark signal, and the used detector) and the stability condition is fulfilled, the method is applicable.

An analysis of the impact of the stray light level after hardware improvement is added to the manuscript.

(3) Based on your analysis, do you expect other similar scientific instruments to face the same straylight issues? Do their straylight correction schemes require improvement and why?

Generally, an optical design will always generate stray light. That is why a stray light correction becomes mandatory at a certain point of required precision; the chosen method relies on the type and behaviour of the stray light. Due to similarities in the observed stray light in MAMAP2D-Light, the approach by (Tol, et al., 2018) applied to the TROPOMI-SWIR channel was chosen in this paper. As demonstrated for MAMAP2D-Light, which had a stray light level comparable to TROPOMI and MethaneAIR without an applied stray light correction, it is possible to deliver scientifically relevant concentration measurements using only the data-driven proxy correction. However, this approach sacrifices single-read-out column precision and is limited to small scales where the background concentrations can be assumed constant. This manuscript is the first to demonstrate the effectiveness of the proxy method in mitigating stray-light-induced errors. However, the applied stray light correction algorithm enables a new field of analysis of the retrieved data, allowing for the distinction of mixed plumes, which was not initially a scientific objective of the instrument. Other instruments, such as MethaneAIR, TANGO, Sentinel-5 or GOSAT-GW, are designed to use the proxy method in the 1.6 micron band. In the first order, a stray light correction of the SWIR-1 data will enhance the detection limit for single-read-out measurements of the single columns. Mentioned MethaneAIR, TANGO, Sentinel 5 and GOSAT-GW in the conclusions

Except for strict band-pass filtering, which is challenging to apply in non-collimated light paths, spectrometers will always suffer from out-of-band stray light. Due to the lack of knowledge of the source illumination level, it is challenging to correct for it, as shown in the measurements with MAMAP2D-Light. The OBSL has a minor impact, since the introduced pseudo-noise from other unknowns dominates. However, this may be different for other

instruments and retrieval methods that consider further sources for pseudo noise. Added OBSL analysis in Sect 6.3.2 (Sect. 4.4.2 of the revised manuscript)

The 2D-slit homogeniser assembly in the object plane of the spectrometer offers several advantages in terms of spectrometer performance, as introduced first in MAMAP and further in AirmapDOAS, MAMAP2D, and CAMAP (Gerilowski, et al., 2025). It is also planned to be implemented in the optical design of the CO2M mission. However, this manuscript demonstrates that a critical part of the light path will lead to an increased stray light level if not carefully treated to reduce reflectivity. So, regarding stray light, it is essential to consider the object plane during instrument design. Established link to CAMAP and CO2M in the introduction and conclusions

Another aspect of the chosen stray light correction approach is the computational power required for correction and the corresponding time it takes. An iterative FFT-based deconvolution is less power-consuming than handling matrices with 12 billion elements (288x384 pixels in M2DL) as required by the method introduced by (Zong, Brown, Meister, Barnes, & Lykke, 2007). To establish clarity for the focus of the manuscript, this aspect isn't present anymore in the introduction

For all highly precise optical instruments, it is worth performing stray light characterization measurements. Based on the received information, the instrument's performance can be enhanced. The stray light estimates obtained through simulations are only as accurate as the dependencies in the underlying model are defined.

(4) The proposed straylight correction should be applied to the data recorded after the hardware modification and evaluated against established stray light correction approaches. How does the remaining straylight influence impact the GHG retrievals?

Agree, an analysis based on the simulated spectra is added to the hardware improvement chapter Sect. 8. Unfortunately, there is no measured data available showing that the leftover stray light after hardware improvement leads to a significant decrease in data quality.

The manuscript is focused on the impact of stray light on the retrieved data. The method is already established for the TROPOMI SWIR channel and for MethaneAIR. Another established method for spectrometers (Zong, Brown, Meister, Barnes, & Lykke, 2007) is applied, for example, to the hyperspectral EnMAP (Baumgartner, et al., 2025). The method would require characterization measurements in a denser grid, which were not performed in the CoMet 2.0 configuration. A comparison based on measured spectra with MAMAP2D-Light would be questionable, as shown in the manuscript, since other (pseudo) column noise sources become dominant after stray light correction. These sources can only be made visible by addressing them in the retrieval algorithm.

(5) The authors mention an order sorting filter in the setup. This is an essential component when it comes to straylight. Straylight is minimized by a filter, which only transmits the spectral range used for the spectral retrieval. Is this the case?

The order sorting filter is an edge filter transmitting radiance above 1500 nm, so the near field OBSL is not affected; however, the stray light from different refractive orders is reduced. A customised band-pass filter for the spectrometer's given spectral range would indeed

suppress the out-of-band stray light, even with challenges due to a non-collimated light path at the filter position. Added "1500 nm cut-on" in line 98 of the revised manuscript.

**Summarized and further applied changes in the revised manuscript:**

- Rewritten the introduction and conclusions with a focus on the retrieved data Highlight similarities with CO2M, MethaneAIR, GOSAT-GW, Sentinel 5 and TANGO
- Differentiate clearly between OFSL and OBSL
- Description of the stray light correction for the stray light components already addressed in the TROPOMI correction, as well as minor adaptations regarding detector effects, and laser side-modes, are moved to the Appendix. The newly developed approach for OBSL stays in the main text.
- The structure of the manuscript is improved by shifting Sect. 3 to Sect. 6 (Sect. 4.1 in the revised manuscript) and combining the stray light characterization with the correction to Sect. 3 in the revised manuscript.
- Changed "real" measurements to measurements
- Made Sect. 6.3.2 more consistent to clear results regarding single read-out column noise, OFSL and OBSL, including Fig. 13 (14 in the revised manuscript).
- Added analysis of the impact of the stray light level after hardware improvement on the simulated data, in Sect. 8, Sect. 6 in the revised manuscript. Furthermore, the handling of the noise in the stray light level estimation in Sect. 8 has been improved, which leads to increased stray light levels. We added an estimation of the remaining stray light level after the applied stray light correction for the CoMet2.0 configuration to this section.
- A Bug in the stray light correction code was found and eliminated, which is leading to non-significant minor differences in Fig. 13, 15,17 (in the revised manuscript: Fig. 13, 15, 16)

**20 1 Introduction**

Passive remote sensing has become one of the cornerstones for monitoring the most critical greenhouse gases (GHGs), carbon dioxide (CO2) and methane (CH4), in the Earth's atmosphere to determine anthropogenic and natural GHG emissions. The spectral absorption features of the GHGs in reflected sunlight are exploited to retrieve the corresponding atmospheric GHG

concentrations. However, depending on the instrument's spatial and spectral resolution, the distance from the source, and the source area, surface emissions introduce only minor changes in the measured absorption features compared to the absorption features due to the accumulated background concentrations in the total atmospheric column. For Therefore, the spectra have to be measured very precisely to enable accurate emission estimates, therefore, which is translated into strict instrument-dependent specifications of for the accuracy of the spatial and spectral calibration accuracy of the measured spectra required, allowing to retrieve the according atmospheric GHG columns with the required accuracy and precision.

For an instrument with a given spatial and spectral resolution, the required column precision is determined by the detection limit required for the envisaged emission estimates (Jacob et al., 2022; Pandey et al., 2023). For example, the CH4 column single-measurement precision for SCIAMACHY, the first instrument dedicated applying solar backscatter absorption spectroscopy to remote sensing of GHGs from space aboard the EnVISAT ENVISAT satellite, was planned to achieve 1% (Bovensmann et al., 1999). For its successor, TROPOMI (TROPOspheric Monitoring Instrument), the goal precision was tightened to 0.6% for a single measurement (Veefkind et al., 2012). To achieve this precision, high-quality instrument characterizations, minimizing and correcting radiometric errors, are required the calibration measurements, therefore, need to characterize radiometric errors precisely to implement corrections minimizing their impact on the measured spectra.

A significant contributor to the radiometric error is stray light, which arises from reflections and scattering processes that are not intended in the optical design. The definition and terminology of stray light are adapted from Fest (2013). Stray light distorts the measured spectra with a continuum-dependent error (Tol et al., 2018) and is most prominent in high-contrast scenes, e.g., in mixed scenes with dark land surfaces and bright clouds. However, the stray-light-induced error signal depends on the overall intensity distribution of the light paths within the system. The spectrally, spatially, and intensity-dependent error signal introduces error patterns in the retrieved concentrations and can be misinterpreted as column enhancements and, in certain cases, even as plumesemission plumes from point source emitters. Therefore, it is essential to mitigate stray light within the optical system. Effective mitigation of stray light involves minimizing it through an optimized optical design, usually via simulations during the design phase of an instrument, and correcting it during data processing. The latter uses so-called based on stray light kernels estimated from stray light characterization measurements.

Various methods for For the stray light correctionhave been developed. A widely used approach for non-imaging spectrometers, described by Zong et al. (2006), exploits matrix operations to correct both spectrally and spatially stable and variant stray lightsimultaneously. This method requires dense characterization measurements or interpolations across the entire focal plane array. For imaging spectrometers—, the presence of an additional spatial axis increases the size of the correction matrix by the squared number of spatial pixels and therefore the required computational resources massively (Zong et al., 2007). For the SWIR channel of TROPOMI, it is essential to analyze the origin of the stray light. The origin and behaviour is highly dependent on the instrument type (Clermont et al., 2024; Baumgartner et al., 2025). For grating spectrometers with a relatively narrow spectral range, Tol et al. (2018) introduced a tailored method that separates stable and variant stray light components. The stable component is corrected using an iterative deconvolution method, which has also been applied for the MethaneAIR instrument (Staebell et al., 2021). The variant component is addressed through a spatial transformation based on the variability of the stray light depending on the origin. This method leverages the fast Fourier transform (FFT) for deconvolution, enabling

50

55

a near-real-time the globally stable stray light correction. from the variable stray light due to, e.g., ghosts.

This work focuses on spectra measured with For this work, the stray light correction from Tol et al. (2018) is adapted to MAMAP2D-Light (Methane Airborne MAPper 2D Light), a lightweight airborne remote sensing push-broom imaging grating spectrometer built at the Institute for Environmental Physics (IUP) Bremen. Besides satellite-based instruments, airborne remote sensing spectrometers offer spatially high-resolution measurements with similar spectral specifications. This allows intercomparison between both measurement platforms, airborne and spaceborne, for the used GHG concentration retrieval algorithms and the final emission rate retrievals. In contrast to satellite-based instruments, airborne spectrometers can be recalibrated and improved during their lifetime.MAMAP2D-Light is buildung built on concepts established with the MAMAP (Methane Airborne MAPper) instrument (Gerilowski et al., 2011) and. It is designed to measure CH4 and CO2 column anomalies in the 1.6 µm band, exploiting the CO2 (Krings et al., 2011) or the CH4 (Krings et al., 2013) proxy method. The data set used in this study was collected during the CoMet 2.0 Arctic mission, which took place in summer 2022 in Canada1. The GHG concentrationswere, which is also established for MethaneAIR (Chan Miller et al., 2024), and planned for the GOSAT-GW (Observing SATellite for Greenhouse gases and Water cycle) (Tanimoto et al., 2025) and Sentinel-5 (Landgraf et al., 2019) missions.

For remote sensing of GHGs, airborne remote sensing spectrometers provide smaller ground scenes compared to satellite-based observations with similar spectral properties. This offers the opportunity to distinguish between real enhancements and concentration anomalies introduced by instrument, atmospheric, or surface-related error sources (Gerilowski et al., 2011). Therefore, airborne demonstrators, such as the MAMAP2D (Methane Airborne MAPper 2D) and the CAMAP (CO2 And Methane Airborne maPper) (Gerilowski et al., 2025) instrument currently developed as the airborne demonstrator for the CO2M-Mission (Sierk et al., 2021), are a valuable complement for satellite missions to understand the impact of instrumental error signals, as stray light, on the retrieved data products with real measurements.

This work focusses on CH4 concentrations, retrieved using the WFM-DOAS (Weighting Function Modified Differential Absorption Spectroscopy) method Krings et al. (2011), in combination with the CH4/CO2 proxy method, in the following abbreviated as proxy method, which has been proven to deliver reliable CH4 and column anomalies on the local scale in the past (Krings et al., 2011, 2013; Krautwurst et al., 2017, 2021, 2024). A post-campaign stray light characterization with a Littman/Metcalf tunable laser (Stry et al., 2006) revealed a significant stray light contamination in the campaign dataset.

Therefore, a post-flight (Krings et al., 2013; Krautwurst et al., 2017, 2021, 2024; Borchardt et al., 2025). For airborne remote sensing, the sensitivities of the proxy method to deviations in the atmospheric state and observation geometry have been analyzed through simulations by Krings et al. (2011). Besides the advantages of the proxy method, it is only feasible if the proxy concentration (in this case, the CO2) remains constant; otherwise, it either underestimates or overestimates the concentrations by definition.

In this work, the adapted stray light correction was applied, exploiting the previously performed characterization measurements.

The stray-light-contaminated campaign data is applied to measured spectra collected with MAMAP2D-Light during the CoMet

<sup>1https://comet2arctic.de/ last access: 11.02.2025

- 2.0 Arctic mission, which took place in summer 2022 in Canada1. The campaign dataset is contaminated with 5.6% of stray light, which is only slightly higher than the estimated correctable stray light of 4.4% in the TROPOMI SWIR channel (Tol et al., 2018).
- Applying the stray light correction to the CoMet 2.0 data set provides a unique opportunity to investigate the impact of the stray light correction on real measured data and the capabilities of the proxy method in the presence of stray light in the stray light and its correction in the entire processing chain from the measured spectra to the retrieved GHG concentrations and the derived emission rate estimates -with and without the applied proxy method. This especially examines the capabilities of the proxy method to correct stray-light-induced errors in the single CH4 column.
- In Sect. 2, the instrument design of MAMAP2D-Light is introduced. The impact of stray light on the used WFM-DOAS retrieval is described in Sect. 4.1. The stray light stray light characterization measurements of MAMAP2D-Light are summarized in Sect. 3.1. From the characterization measurements, a and the applied stray light correction algorithm is applied are summarized in Sect. 3.13. The correction is applied to real-measured and simulated spectra, and the impact on the WFM-DOAS method as well as the impact on the retrieved concentration maps is analyzed in Sect. 4. From the concentration maps, the resulting CH4 emission rates of two measured landfill plumes are analyzed based on in Sect. 5 with respect to the impact of the applied stray light correction and the proxy correctionin Sect. 5. With the post-flight stray light characterization measurements, the origin of the majority of the stray light has been localized and mitigated finally improved by a hardware improvement shown modification summarized in Sect. 6. The expected error in column noise after the hardware improvement is determined.

**2 MAMAP2D-Light instrument**

MAMAP2D-Light is an airborne passive remote sensing instrument for observing atmospheric CO2 and CH4 columns using 110 infrared spectroscopy solar backscatter absorption spectroscopy in the short-wave infrared around 1.6 µm. The MAMAP2D-Light instrument, shown in Fig. 1, is a push broom imaging spectrometer with a planar reflective grating. It weighs approximately 43 kg and fits into an underwing pod of a motor glider aircraft (e.g., Diamond HK 36-TTC ECO, (Borchardt et al., 2025)). MAMAP2D-Light covers the wavelength range from 1559 nm to 1690 nm with a spectral resolution of approximately 1.1 nm. It comprises a front optic, an optical fiber bundle (see Fig. 2), an entrance slit unit (ESU), two different lenses, serving as 115 collimator and camera optics, a planar reflection grating and an infrared detector (AIM SWIR384). The detector deployed has a Mercury Cadmium Telluride (MCT) focal plane array (FPA) comprising 384 pixel × 288 pixel and a pixel pitch of  $24 \, \mu \text{m} \times 24 \, \mu \text{m}$ . Within the spectrometer, the FPA is oriented in a way that the spectral axis is along the 384 pixel axis, which results in a spectral oversampling of  $\sim 3$  pixel, while the spatial axis is along the 288 pixel axis. The FPA is cooled to approximately 150 K with a linear single-piston cooler to reduce the internal thermal dark current of the MCT. The spectral cut-off 120 was adapted from  $\sim 2.5\,\mu m$  to  $\sim 1.8\,\mu m$  by the manufacturer to reduce the sensitivity to thermal background radiation from the optical bench and mounting elements and thereby, allowing to operate the instrument allowing the instrument to operate at ambient temperature.

<sup>1https://comet2arctic.de/ last access: 11.02.2025

**Figure 1.** Schematic optical setup of the MAMAP2D-Light instrument. The reflected sunlight reflected from the Earth's surface is imaged by the front optic on onto the input of a fiber bundle with 36 fibers stacked in Ferrule 1, where each fiber corresponds to a single spatial ground scene. The radiation enters the spectrometer block through the fibers in Ferrule 2. The fibers are stacked perpendicular to the optical bench of the spectrometer block. To adjust the linewidth of the spectrometer, a Slit aperture is placed in the entrance focal plane. The radiation is dispersed and imaged at the 2D detector with the two optics and the grating. The area of a single fiber together with the focal length of the front optics defines the instantaneous field of view (IFOV). 28 fibers are imaged at the detector, determining the field of view (FOV).

MAMAP2D-light measures scattered sunlight from the Earth's surface, which is imaged via the front optical lens onto an optical fiber bundle with 36 rectangular single fibers stacked in a ferrule, see Fig. 2, acting as a 2D-slit-homogenizer (Hummel et al., 2022; Gerilowski et al., 2025). Each fiber has a fiber core of  $\sim 300 \, \mu \text{m} \times 100 \, \mu \text{m}$  in spatial and spectral direction, respectively. The outer dimensions of the fibers with cladding are  $\sim 315 \, \mu \text{m} \times 175 \, \mu \text{m}$ . Due to the orientation of the detector, only 28 of the 36 fibers are imaged at onto the detector, resulting in 28 across-track ground scenes observed by the instrument. The entrance slit unit ESU of the spectrometer comprises the ferrule on the fiber bundle's second end, an adjustable slit, an uncoated adjustable slit aperture (Acton Research, Model SPS-716-1S), an 1500 nm cut-on optical order sorting filter and a shutter unit. The light entering the spectrometer is collimated by a lens collimator system (the collimator) with a focal length of  $F_c = 300 \, \text{mm}$  and an aperture of F/N = 3.5. The dispersed collimated light from the grating is then focused on the detector by the camera lens optics with  $F_o = 200 \, \text{mm}$  and F/N = 2.4. The angle of the optical axes between the lenses is  $32^\circ$ . The grating deployed in MAMAP2D-Light is a ruled plane grating with  $300 \, \text{lines mm}^{-1}$  and a nominal blaze angle of  $17.5^\circ$ , which is operated at the  $-1^{\text{st}}$  order.

125

130

135

During the CoMet 2.0 campaign, the installed slit aperture in Fig. 1 was adjustable. The slit, an uncoated adjustable slit

Figure 2. Image of a section of aligned fibers within the aluminum ferrule from Fig. 1. The fiber core dimension is  $\sim 300 \, \mu m \times 100 \, \mu m$ .

aperture, as shown in Fig. H1 (a), consists 3, consisting of two uncoated steel blades. Initially, it was intended to adjust the ISRF and the spectral oversampling with the slit aperture. However, due to misaligned fibers (see Fig. 2) in the entrance ferrule, the slit was not used and was therefore uncoated adjustable slit aperture was left open to its maximum using only the fiber geometry as the entrance slit.

140

The swath of the instrument is defined by the focal length of the input front objective  $F_f = 25 \,\mathrm{mm}$ ; in combination with the

**Figure 3.** Uncoated adjustable slit aperture as mounted during the CoMet 2.0 mission at the entrance fiber ferrule (ferrule 2, in Fig. 1) of the spectrometer. (a) side of uncoated adjustable slit aperture showing in direction of the ferrule. (b) side of uncoated adjustable slit aperture showing in direction of the collimator lens.

FPA spatial pixel count, the pixel size and the imaging ratio  $F_o/F_c$ . The total swath is defined by the fully imaged fibers on the FPA since the fiber bundle length is larger than the detector width. The resulting across-track field of view (FOV) for the full detector is about 23.3°. However, the exact field of view is defined by the length of the input fibers fully imaged on the detector. This leads to a real FOV of 22.6°. For the CoMet 2.0 campaign (this work), MAMAP2D-Light was integrated on a Gulfstream G 550 (HALO, High Altitude and LOng Range Research Aircraft, operated by the DLR, Deutsches Zentrum für Luft- und Luft- Raumfahrt). With a flight altitude of  $\sim 8 \, \mathrm{km}$  above ground level, the FOV of 22.6° led to a swath width of  $\sim 3.5 \, \mathrm{kmat}$  a above ground level, with a sampling of 28 spatial fibers, corresponding to an across-track spatial resolution of  $\sim 120 \, \mathrm{m}$ . The along-track ground scene size is dependent on the flight speed, the integration exposure time, and the number of binned ground scenes, and was adpated adapted to  $\sim 120 \, \mathrm{m}$  by binning  $\sim 5 \, \mathrm{single}$  measurements for the flights in Canada during CoMet 2.0. Due to MAMAP2D-Lights' compact dimensions and weight of approximately  $43 \, \mathrm{kg}$ , the system also fits

into an underwing pod of a motor glider aircraft (e.g., Diamond HK 36-TTC ECO) and was successfully deployed in this configuration for an airborne campaign in Australia (Borchardt et al., 2025).

**3 The impact of stray light in the WFM-DOAS retrieval**

165

175

The GHG anomalies are retrieved from the measured spectra using the WFM-DOAS method. This method does not consider any corrections for an additive error signal, which is the expected type of error resulting from stray light contamination. This section describes the WFM-DOAS retrieval and the impact of an additive offset within the WFM-DOAS retrieval.MAMAP2D-Light measures the spectra of the sunlight passing through the atmospheric column. Depending on the depth of the absorption bands of the corresponding GHG, the anomalies of GHG concentrations are retrieved from the spectra using the WFM-DOAS method. The WFM-DOAS retrieval was initially developed for the spaceborne SCIAMACHY instrument by Buchwitz et al. (2000). The algorithm was later adapted for the airborne measurement geometry by Krings et al. (2011) for the MAMAP instrument. Krautwurst et al. (2024) describe the retrieval algorithm's latest version as applied to MAMAP2D-Light data.Based on Lambert Beer's law, a calculated RTM at a wavelength  $R_{\lambda}^{mod}(\bar{\mathbf{c}})$  for a state of the atmosphere represented by the model state vector  $\bar{\mathbf{c}}$ , can be modulated to get the RTM at the state  $\bar{\mathbf{c}}$  of the measurement  $R_{\lambda}^{mod}(\bar{\mathbf{c}})$ . The weighting functions,  $W_{\lambda,c_j}$ , describe the change of radiance due to a change of the respective parameter j. An additional low-order polynomial  $P_{\lambda}$  with a free parameter vector  $\bar{\mathbf{c}}$  approximates slow spectral variations due to scattering or spectral surface reflectance, which have to be considered but are not quantified. This results in the following equation:

$$\ln R_{\lambda}^{mod}(\mathbf{c},\mathbf{a}) = \ln R_{\lambda}^{mod}\left(\overline{\mathbf{c}}\right) + \sum_{j} W_{\lambda,\overline{c}_{j}} \frac{c_{j} - \overline{c}_{j}}{\overline{
[revised manuscript text omitted]
 PRF area of MAMAP2D-Light is larger than in the TROPOMI SWIR spectral band, leading to larger areas of saturation during the stray light characterization, surrounded by a 1-pixel wide area of blooming2 around the saturated pixels. To get reliable data in the saturation area, the exposure time was increased in specifically adapted

<sup>2Blooming occurs due to photogenerated charges within a saturated pixel, which are not fully collected by the pixel's read-out electronics. The leftover charges are then collected by the neighbouring pixels.

**Figure 5.** (a) Photograph of the entrance aluminium fiber ferrule, Ferrule 2 in Fig. 1, with a black anodized mount. A single fiber is illuminated with the setup shown in Fig. 4 and a white light source. The illuminated fiber is visible as a red dot. (b) zoom in on the fiber stack.

smaller steps. The dark signal level, increasing linearly with the exposure time due to thermal radiation, constrained the highest exposure time to 3000 ms. The dark signal for each point was measured for each exposure time after a complete set of exposure times with illumination, by shutting off the laser. The measurements were flat-field corrected, where the fibers' cladding areas (displayed as dark lines in the spectral direction in Fig. E1) were interpolated by fitting a 2-dimensional 3rd-order polynomial to the fiber core signal. The dark current corrected data showed patterns related to a detector effect, which were most prominent for higher exposure times with increased saturation. The patterns were corrected using a data-driven approach, which is shown in detail in Appendix D.

The measured signals at one position for all exposure times were merged into a single two-dimensional frame —by the following procedure: Therefore At each exposure time, saturated and blooming-contaminated pixels had to be were filtered out. Blooming³ occurred in the measurements at higher exposure times due to the saturation of the directly illuminated pixels. Due to the CMOS-based read-out electronics of the detector, only the directly neighbouring pixels of a saturated pixel are affected by blooming. For merging, each non-saturated and non-blooming-contaminated pixel value at the highest exposure time was selected. The full merged frame was finally normalized to the integral of the signal over all pixels. The merged frames of the stray light characterization measurements for MAMAP2D-Light in the CoMet 2.0 configuration for four different spot positions are shown in Fig. 6. The measurements revealed several stray light and non-stray-light-related artifacts which are discussed in the next section.

**3.2 Stray light contributors components**

245

250

The stray light contributors were separated into a spectrally and spatially *invariant* part independent of and a spatially *variable* part depending on the position of the illuminated spot. Within the measurements, the relative spectral position of the *variable* stray light was constant. The description for the stray light sources uses the terminology defined in Appendix A, where the stray light is classified in different orders. With each stray light process (e.g., scattering, reflection, etc.) in a light path, the

<sup>3Blooming occurs due to photogenerated charges within a saturated pixel, which are not fully collected by the pixel's read-out electronics. The leftover charges are then collected by the neighbouring pixels.

Figure 6. Different spectral and spatial spots from the stray light characterization of MAMAP2D-Light in the CoMet 2.0 configuration. (a) and (c) were recorded at  $\sim 1628$  nm,(b) and (d) at  $\sim 1661$  nm. The horizontal line at the right-hand side of the illuminated spot is due to the not completely suppressed laser side modes (LSM) of the laser used. The vertical line through the illuminated spots is caused by light from outside the instrument entering the spectrometer through the fibers. A sharp ghost appears spatially mirrored but spectrally in a constant offset from the initial spot. Further, the spectral and spatial invariant stray light (stable stray light; SSL) cone around the illuminated spot is shown in all images.

order is increased, starting with the intended light path as the 0th order.

260

265

The observed stray light was separated into two components based on their position relative to the illuminated spot, dependent on the position of the illuminated spot on the detector: The spectrally and spatially *invariant* stray light part always occurred at the same relative position, while the spatially *variable* part changed its relative position. Nevertheless, the relative spectral position of the *variable* stray light was constant.

[revised manuscript text omitted]

The reflection Kernel  $\mathbf{K}_{refl}$  was determined from the relative positions of the ghost spot to the originally illuminated spot, see Fig. 6. In the spectral direction, the relative offset  $x_{refl}$  was constant. In the spatial direction, the ghost spot was mirrored and shifted by  $y_{refl}$  from the center. A spot search algorithm defined  $x_{refl}$  and  $y_{refl}$  based on the relative distances between ghost and origin spots' barycenters.  $\mathbf{K}_{refl}$  shifted the frame to the ghost position. Since the ghost spot was a sharp image,  $\mathbf{K}_{refl}$  would be ideally a single pixel with the value 1 at  $x_{refl}$  and  $y_{refl}$ . However, due to floating values, the signal pixel was initially set to the nearest integer value and afterward shifted by the decimal points using the "shift" function of the "scipy.ndimage" package (version 1.13.1) in python. Thus, the signal in  $\mathbf{K}_{refl}$  had an area of 2 pixels  $\times$  2 pixels. The relative intensity variability of the ghost spot and the origin spot is represented by  $\mathbf{E}_{refl}$  and was generated from the wavelength grid, instrumental response function (Appendix II and I2) and the stray light characterization measurements using the equation:

$$\mathbf{E}_{refl} = \frac{\mathbf{S}_{refl}^{R}(x - x_{refl}, y - y_{refl})}{\mathbf{S}_{origin}(x, y)}.$$

335  $S_{origin}$  represents the signal of the origin spot and  $S_{refl}$  is the corresponding signal of the reflected spot, which is shifted by the corresponding  $x_{refl}$  and  $y_{refl}$  values. The respective signal levels within a fiber were determined by the mean intensity of

the spot, defined by a half-maximum threshold. The R-operator is mirroring the y-axis. Due to the sparse data availability for  $\mathbf{E}_{reft}$ , a two-dimensional first-order polynomial fit was deployed to fill the data gaps, shown in Fig. B2. Higher orders in the fit function led to a stronger variability of the values in the unknown edges. The RMS of the relative fit residuals was  $\sim 8\,\%$ . A more accurate  $\mathbf{E}_{reft}$  estimation would either require a denser grid of stray light measurements or, e.g., wavelength grid measurements with an increased dynamical range, as done for the stray light characterization measurements, see Sect. 3.1. The second term in Eq. B2  $(\mathbf{K}_{reft}*(\mathbf{E}_{reft}-\mathbf{J})^R)$  represents the amount of reflected stray light in the frame. However, the slit was not perfectly aligned vertically, and due to the smile effect3 slightly curved. This distortion needed to be corrected before the mirroring operation was performed and reversed before subtraction. The correction was achieved by shifting each row by a value  $x_{smile}$ , row. This value was determined by the difference between the barycenter of each row from a measurement of a full slit and the median of all barycenters from the same measurement. The resulting  $\mathbf{x}_{smile}$  array for all rows was the median for each row from the wavelength grid and instrumental response function (refer to Appendix II and I2) measurements. The correction is only valid due to the relatively small wavelength dependency of the diffraction angle defined by the groove frequency of the grating with 300 lines mm $^{-1}$ .

**3.2 Out-of-field stray light in MAMAP2D-Light**

340

345

350

355

360

365

Due to the extension of  $K_{stable}$  and the offset of  $K_{reft}$  from the center, the out-of-field stray light (OFSL) contaminated the Out-of-field stray light (OFSL) and OBSL contaminated the spatial and spectral edges of the measured frameframes. To consider the spectral OFSL (or out-of-band stray light) OBSL in the correction, the spectral axis of the measured frame was extrapolated with an extended RTM, as used in Sect. 4. The RTM was fitted to each row of the dark current and flat field corrected frame and scaled with a polynomial ( $3^{rd}$ -order) and spectral shift parameter within the spectral range of MAMAP2D-Light. The extended spectra were then derived from scaling and shifting the full RTM range with the derived fit parameters. This method of extrapolation gives only an estimation of the signal level of the spectral OFSL provides only an estimate of the OBSL signal level, and the expected impact of estimation uncertainty is discussed in Appendix E3. It is important to note that the surface spectral reflectance and the aerosol scenario have an impact on the signal level of the spectral OFSL OBSL and would affect the correction quality even with in a perfectly characterized system.

[revised manuscript text omitted]

$$\quad \ln R_{\lambda}^{mod}(\mathbf{c}, \mathbf{a}) = \ln R_{\lambda}^{mod}(\overline{\mathbf{c}}) + \sum_{j} W_{\lambda, \overline{c}_{j}} \frac{c_{j} - \overline{c}_{j}}{\overline{c}_{j}} + P_{\lambda}(\mathbf{a}) + \varepsilon_{\lambda}$$

$$\tag{1}$$

The values of the parameters j (e.g. the GHG concentrations) building the state vector of interest c are retrieved from a measured spectrum  $R_{\lambda}^{mea}$  by a least squares fit with the fit parameters c and a. To separate the noise contribution of the stray light from other noise sources, simulated synthetic measurements are included in the analysis

$$\underset{\mathbf{a}, \mathbf{c}}{\operatorname{arg\,min}} \left\| \ln R_{\lambda}^{mea} - \ln R_{\lambda}^{mod}(\mathbf{c}, \mathbf{a}) \right\|^{2} \tag{2}$$

Stray light is radiation deviating from the intended light path and illuminating the FPA at unintended positions. The position of the intended path in the focal plane is called the origin position, and the unintended position is called the target position (For terminology, see again Appendix A). Stray light causes an additive error signal (or zero-level offset) *e* at the focal plane. The error signal occurs in the target spectrum and, by being absent, also in the origin spectrum. The fitting in Eq. 2 is then performed to a measured spectrum of

410
$$\ln R_{\lambda}^{mea,real} = \ln(R_{\lambda}^{mea} + e).$$
 (3)

While the polynomials  $P_{\lambda}(\mathbf{a})$  are introduced to catch, among others, instrumental error signals, they are in fact additive components to the logarithm of the radiance in Eq. 1, and therefore, scalable multiplicative factors of the radiance. Consequently,

in WFM-DOAS, the additive offset e is tried to be compensated for by a multiplicative scaling factor of the polynomial. This introduces a signal level-dependent scaling error, which leads, in the case of a positive error signal, to a shrinking of the absorption line depths relative to the continuum. The corresponding fitting parameter  $c_i$  then "sees" shallower trace gas absorption bands, which leads to an underestimation of the retrieved column anomaly. Therefore, an additive offset can not be observed in the spectral residuals  $e_{\lambda}$  of the fit, except for areas in the spectral window without any trace gas-related absorption bands, e.g., pure Fraunhofer-Lines.

MAMAP2D-Light is designed to quantify GHG anomalies relative to the background concentrations. As the normalization of the retrieved columns to the background is performed in the post-processing, described in detail in Sect. 4.2, a constant additive offset would not impact the precision of the retrieved column anomalies. However, the impact of stray light depends on the radiation of the source and the amount of the intended radiation within the target spectrum. Thus, scenes with inhomogeneous albedo, spectral surface reflectance, or aerosol scenario result in decreased precision in the retrieved column anomalies.

**425 4.2 Data processing**

430

435

The column anomalies were retrieved with the airborne WFM-DOAS method, which is described briefly in Sect. 4.1 and in detail by Krautwurst et al. (2024). The retrieval delivers column anomalies from the trace gases of interest as profile scaling factors (PSF) of atmospheric profiles at the mean state of the atmosphere during the measurements using an RTM calculated with SCIATRAN 3.8 (Rozanov et al., 2014). The spectra were dark current corrected, radiometric calibrated by a calibrated sphere measurement, see Sect. 3.1, and wavelength calibrated. The retrieved data was filtered using a root-mean-squared (RMS, see in Sect. 4.1) threshold of the fit residuals to assess the quality of the fit. To account for signal intensities exceeding the linearity range of the detector and to keep a sufficient signal-to-noise ratio, a maximum and minimum signal threshold was applied.

The retrieved column data showed a nonlinear dependency on the detector filling. This phenomenon has already been observed for MAMAP data and is discussed by Krautwurst et al. (2017). For MAMAP2D-Light, the nonlinear dependency for each spatial sample was corrected with a data-driven approach analogous to that developed for MAMAP. A low-order polynomial (2nd - 3rd order) was fitted to the column data over the detector filling for one spatial fiber over a single flight leg. The column data was then normalized by the fit result.

Typically (Krings et al., 2013; Krautwurst et al., 2017, 2024), the proxy method is used to minimize the impact of light-path errors, like multi-scattering or instrumental error. The CH4 proxy is the ratio of the retrieved CH4-PSF and the CO2-PSF, assuming a constant CO2 concentrations over the measurements area:

$$CH_{4,proxy} = CH_{4,psf}/CO_{2,psf}. (4)$$

However, the proxy method underestimates mixed plume signals either underestimates or overestimates plume signals if the  $CO_{2,psf}$  is not constant, e.g., due to  $CO_{2}$  emissions nearby or background changes due to large-scale gradients in the  $CO_{2}$  concentration. Therefore, in this work, the non-proxy corrected single columns are also analyzed in more detail.

Depending on the altitude at which the  $CH_4$  plume and therefore the concentration perturbation is located, the WFM-DOAS retrieval has varying sensitivities. This sensitivity is described by the altitude-dependent averaging kernel AK(z) (Krings et al., 2011). For the CoMet 2.0 data, it was computed for each ground scene, considering its respective surface elevation and assuming that all enhancements are located below the aircraft. Based on the AK(z), conversion factors  $c_f$  were derived used for correction of the retrieved PSFs:

$$CH_{4,rel} = (CH_{4,psf} - 1) \cdot c_f \tag{5}$$

The column data was georeferenced using the aircraft position and attitude and the surface elevation,—. The procedure is described in detail by in Krautwurst et al. (2024).

**4.3 Stray light in high contrast scenes column anomalies**

450

465

Initial results of the CoMet 2.0 campaign dataset revealed an error pattern in the proxy-corrected CH4 column anomalies for the scene shown in Fig. 8. The data was processed using the retrieval, RTM, and orthorectification parameters shown in Tab. G2. In the non-stray-light-corrected concentration map in Fig. 8 (c), significantly enhanced CH4 column anomalies are shown. The intensity map in Fig. 8 (b) revealed a high contrast scene, where the surfaces consist of highly reflective sand, low-reflective vegetation, and a nearly non-reflective lake. The CH4 column anomaly pattern resembles the mirrored sand surface, which aligns with the mirrored ghost seen in Fig. 6. After the stray light correction, the structures in the CH4 column anomalies were reduced, but not erased. This is related to the not accurately known reflection intensity distribution Erefl shown in Fig. B2 and Sect. discussed in Appendix B2.

The stray light correction also reduced further negative column anomalies, which were located in at ground scenes with low intensity compared to the across-track neighbouring ground scenes. In this scene, with applied proxy correction, the total column noise was reduced from 0.40% to 0.33% by the stray light correction.

The impact of the stray light mainly depends on the signal level and distribution of the origin and the signal level of the target spectrum. In Fig. 9, the reflected and the stable error signal for a target spectrum are shown. The reflected stray light introduces a more structured and different curved error signal, whereas the stable stray light is smoother and follows the curve of the target spectrum. The proxy method is unable to correct imbalanced error contamination in the  $CO_2$  and  $CH_4$  bands. Due to the different absorption line depths, a general imbalance of the sensitivity to a zero-level offset is given; if the zero-level offset varies spectrum, the imbalance can be compensated or amplified. The shown target spectrum is the corresponding synthetic spectrum, which is generated as described in Appendix E, of an enhanced pixel in Fig. 8 (b), which is caused by the contamination of the reflected stray light.

**4.4 Stray light as source for pseudo-noise in column anomalies**

475 The stray light introduced stray-light-introduced error patterns in the concentration maps can be observed as pseudo-noise in the column noise estimate of the real measured retrieved column anomalies. Therefore, in the following, the variation of the column anomalies is analyzed based on a flight leg, shown in Fig. 10, over an area dominated by urban and agricultural

**Figure 8.** Measured scene over high reflective sand and low reflective vegetation. The surface in RGB is shown in (a). (b) shows the intensity in the SWIR measured with MAMAP2D-Light. The non-stray-light-corrected and proxy-corrected processed data is shown in (c), with a column noise of 0.40 %. The stray-light-corrected data in (d) has a reduced column noise of 0.33 %. The RGB map is provided by © OpenStreetMap, accessed using Cartopy.

surfaces, where plume signatures were masked. A plume signal extending from a landfill was masked for the calculation of the column noise. The flight leg was chosen due to the strong variations in surface reflectance. Further, based on the real measured frames, synthetic frames were generated and artificially contaminated with stray light and random noise to simulate the different error types individually in the processing chain. The concentration anomalies were retrieved using the parameters shown in Tab. G1.

**4.4.1 Column noise in real measured data**

480

485

The column noise of the real measured column anomalies was estimated from the standard deviation of the source-free background area. In Fig. 11, the distribution and the column noise of the non-proxy-corrected single columns and proxy-corrected columns with and without applied stray light correction are shown. The column noise of the non-proxy-corrected single

Figure 9. Separated stray light (SL) error signal from the sharp ghost reflex (reflected SL) and the stable kernel (stable SL) with y-axis on the left for a target spectrum with y-axis on the right in a simulated frame.

**Figure 10.** Retrieved CH4 anomalies at the Brady Road Landfill. The results with applied proxy correction (CH4 / CO2) are shown in the left column, and the single CH4 column results are shown in the right column. Non-stray-light-corrected results are shown in the top row, and stray-light-corrected results in the bottom row. The blue arrow marks the wind direction. The map underneath is provided by Google Earth (Image © Airbus 2025, © Maxar Technologies 2025).

columns is significantly improved after the stray light correction. However, after the proxy correction, the stray light correction has no significant impact on the column noise. When comparing the standard deviations of the single CH4 column with the stray light correction to the proxy corrected column, the noise of the single CH4 column is marginally lower. The increased column noise after the proxy correction is associated with the division of two independent quantities contaminated with random noise. However, the impact of the random noise is already reduced by along-track binning of five measurements. The impact of spatial stray light is depicted through a correlation of the mean retrieved column anomalies with the mean

490

495

Figure 11. Histograms of the retrieved single  $CO_2$  (a) and  $CH_4$  (b) and the proxy  $(CH_4/CO_2)$  corrected (c) column data as profile scaling factors (PSFs). The Distributions show data without (blue) and with (orange) stray light correction.

intensity of a measured frame, as shown in Fig. 12. The intensity of each wavelength-calibrated and dark-current-corrected spectrum is derived from the as the mean intensity of the continuum between  $1620.5 \,\mathrm{nm}$  and  $1623.0 \,\mathrm{nm}$ , in digital numbers [DN]. Similar to the column noise in Fig. 11, the correlation of the mean column enhancements with the mean intensities is corrected by the proxy method. However, after the stray light correction, the correlation in the single  $\mathrm{CO}_2$  and  $\mathrm{CH}_4$  columns decreases significantly. The effectiveness of the stray light correction differs between the  $\mathrm{CH}_4$  and  $\mathrm{CO}_2$  columns, impacting the

shown correlation of the proxy and the stray-light-corrected data. This variance may be linked to the  $\overrightarrow{OFSL}$  Correction outlined in Sect. 3.1. Due to the location of the used fit-window (1575 nm - 1677.5 nm) on the detector, the CH4 band is more affected by the  $\overrightarrow{OFSL}$  OBSL than the CO2 band. The position of the CO2 and CH4 bands are marked in Fig. 9.

500

505

Figure 12. 2D-histograms (color) of the average retrieved single  $CO_2$  and  $CH_4$  and the proxy  $(CH_4/CO_2)$  corrected column data dependent of the profile scaling factors per frame on the y-axis and the average intensity of the frame on the x-axis. (a) mean  $CO_2$  PSF with no applied stray light (SL) correction shows a strong correlation with the mean intensity. After the stray light correction in (b), the correlation vanishes. The correlation is also visible in the mean  $CH_4$  PSF data in (c) and vanishes after the stray light correction (d). (e) and (f) show the correlation for the proxy-corrected data, where the stray light correction has only a minor impact compared to the single columns.

**4.4.2 Column Comparison of single read-out column noise in-with simulated data**

The column noise in Fig. 11 after the stray light correction and after the proxy correction stays in the same range of 0.34 with along-track binning to get quadratic ground scenes. Further, the histograms have a normal-distribution-like shape, which is an indication that the total noise is dominated by random noisesquare ground scenes stays in the same range of 0.34%. To have the possibility to separate the stray light introduced error from random noise error sourcesdifferent stray light contributors, synthetic spectra were generated and contaminated with different error signals, from stray light, including OBSL and OFSL, and random noise, as described in Sect. Appendix E.

The different cases and resulting single read-out column noise values are shown in Fig. 13. The first two cases are for comparing the real measured results with the simulated results and show a close correlation regarding the stray light introduced error, which is discussed in detail in Appendix E2. The differences between the real non-stray-light corrected data and the synthetic frames with artificially added stray light and random noise contamination highlight that the simulated data is close but not fully accurate. However, a more accurate model would require precise knowledge of the scene's acrosol scenario, spectral surface reflectance and the out-of-field signal. Nevertheless, the synthetic frames can show the impact of the different error contributors uncertainties are 1-\sigma, estimated using a bootstrap method as the standard deviation from randomly selecting 10% of the datasets 1000 times. In all cases, an applied stray light correction leads to an increased column precision of the single columns compared to the proxy correction. The first three cases show the column noise of the retrieved single-measured cases, depending on the applied stray light correction.

For the stray-light-corrected measured column anomalies, the column noise of the single  $CH_4$  column is  $\sim 13\%$  smaller compared to the column noise after proxy correction. This is related to the division of two random noise-contaminated values. The stray light correction for the simulated data is Considering the OBSL during the stray light correction of the real measured data, with the differences described in the following. The stable stray light is corrected with a perfectly known stable kernel, from Fig. B1. The spatial OFSL is neglected, and the spectral OFSL is perfectly known during the shows no significant impact, i.e., within the uncertainties, on the single columns and proxy-corrected measured data. In the simulated data, where the OFSL, OBSL, and random noise are considered in the contamination, considering OBSL in the stray light correction leads to a minor improvement of 6.5% in the proxy corrected and 2.3% in the single  $CO_2$  concentration data compared to the case of non-considered OBSL in the correction. Due to the spatial shift, see Fig. 7 (b), of the reflected stray light, the lower two fibers are not considered in the column noise estimation of the  $CH_4$  band on the detector, considering the OBSL improved the  $CH_4$  column noise by 18.6%.

The stray light correction for the simulated data with random noise contamination shows a similar impact as for the real measurement; the noise of the single-In the simulated spectra, the OFSL is randomly added but not considered in the correction. When comparing the stray light corrected case with the noise-only contaminated case, the leftover OFSL increases the column noise by 7.1% in the proxy corrected column and 9.1% and columns is significantly reduced, and the proxy corrected column is only marginally affected by the stray light correction. In general, the proxy method is effective for reducing the stray-light-introduced pseudo-noise, which can be seen in the stray-light-only contaminated data. The column noise of the proxy method is higher than 10.7% for the single and CH4 columns in non-stray-light-contaminated data (whether after stray light correction or a priori without stray light) when random noise was added to the data. This is expected due to the division of two independent noise-contaminated values, and CO2 column.

By contaminating the synthetic spectra only with the random noise, the resulting proxy single read-out column noise limit is at  $\sim 0.32\%$ . This is the theoretically achievable column precision for the analyzed measurement. Without considering the spectral OFSL in the stable stray light correction, the column noise is  $\sim 0.22\%$ . With perfect knowledge of the spectral OFSL, this value decreases to  $\sim 0.11\%$ . The leftover column noise is related to the spatial OFSL. For the real measured data, the spectral OFSL is approximated, and the impact is discussed in detail in Appendix E3. The proxy-corrected single read-out

column noise of the real-measured data is  $\sim 0.46$  %, which is  $\sim 44$  % higher than the theoretically achievable minimum. The lower column noise in the real single and columns, compared to the proxy column noise, is a hint that the discrepancy with the simulation is caused by discrepancy between the measured and the simulated data is caused by not considering other (pseudo-)noise originating from sources in the simulation, which are, 
[revised manuscript text omitted]
. 17. Further, the length of the blackened fixed slit aperture blocks the origin of the OFSL. In this section, the amount of stray light after the hardware improvement (SLHWI) is compared to the stray light levels in the CoMet 2.0 configuration with (SLComet.corr) and without applied stray light correction (SLComet.nocorr).

This reduction The amount of stray light was determined by single spot measurements,—where the stray light cone is fully imaged at the detector, as illustrated in Fig. 17 (a) and (c). Both the non-stray-light-related horizontal laser artifact and the vertical line originating from in front of the fiber bundle were masked for the comparison, see Sect. 3.1. Furthermore, a noise threshold was applied, To reduce the random noise in the stray light measurement data while preserving the stray light

Python package (version 0.18.1)) was applied. The denoising weight was estimated from the standard deviation (SD) of the signal in an illumination-free area ( $SD(S_{dark})$ ). Following the denoising, non-stray-light-related signals were excluded via an additional threshold, calculated by the mean and the standard deviation in an (SD) in the illumination-free area ( $Signal_{dark}$  of the denoised image ( $S_{dark,dn}$  and  $SD(S_{dark,dn})$ ). All values below the noise signal threshold were set to zero. The prepared image was normalized to the integrated signal of the entire FPA. The stray light was separated from the origin spot with a threshold value relative to the maximum intensity of the frame. The spot-size threshold is the average relative minimum value of the instrumental response functions ( $\overline{ISRF}_{min}$ ), as described in Appendix 12. The relative stray light is the ratio of the integrated stray light to the total integrated signal. The resulting stray light levels are shown in Tab. 1. The total uncertainty was calculated by the quadratic addition of the uncertainties for the spot size, the noise threshold, and the size and position of the horizontal and vertical masks. The single uncertainties were calculated by disturbing the variables corresponding contributing parameter by the values stated in Tab. 2.

**Table 1.** Relative stray light levels of MAMAP2D-Light in the CoMet 2.0 (Comet) configuration, with applied stray light correction  $(SL_{Comet,corr})$  and without applied stray light correction  $(SL_{Comet,nocorr})$  and for the post-campaign hardware improvement  $(SL_{HWI})$ . The total error is calculated by a quadratic addition of the single components.

| Case                | relative stray light level |  |
|---------------------|----------------------------|--|
| $SL_{Comet,nocorr}$ | $(5.6 \pm 0.39)\%$         |  |
| $SL_{Comet,corr}$   | $(0.9 \pm 0.25)\%$         |  |
| $SL_{HWI}$          | $(2.1 \pm 0.47)\%$         |  |

Table 2. Parameters for stray light quantification and absolute uncertainties for the relative stray light of MAMAP2D-Light in the Comet CoMet 2.0 (Comet) configuration, with applied stray light correction ( $SL_{Comet.corr}$ ) and without applied stray light correction ( $SL_{Comet.corr}$ ), and for the post-campaign hardware improvement ( $HWISL_{HWI}$ ). The total error is calculated by a quadratic addition of the single components.  $\sigma$ -SD represents the standard deviation.

| Uncertainty source     | start-Initial value                                                                            | disturbance Disturbance                           | $\Delta_{Comet}$ $\Delta_{SL_{Convet,nace}}$ |
|------------------------|------------------------------------------------------------------------------------------------|---------------------------------------------------|----------------------------------------------|
| Spot size threshold    | $\overline{ISRF_{min}}$                                                                        | $\pm 3\sigma_{ISRF_{min}} \pm 3SD(ISRF_{min})$    | ±0.104 % ±0.099 %                            |
| Noise Denoising weight | $2SD(S_{dark})$                                                                                | $\pm 1 SD(S_{dark})$                              | ±0.185 %                                     |
| Signal threshold       | $\overline{Signal_{dark}} + 3\sigma_{Signal_{dark}} \overline{S_{dark,dn}} + 3SD(S_{dark,dn})$ | $\pm \sigma_{Signat_{dark}} \pm 1SD(S_{dark,dn})$ | $\pm 0.293\% \pm 0.319\%$                    |
| Mask size and position | defined by hand                                                                                | $\pm 1$ pixel                                     | ±0.076 % ±0.094 %                            |

The stray light in the Comet 2.0 configuration, shown in Fig. 17 (a), was  $(3.9 \pm 0.32)$ %. The stray light after the hardware improvement, as depicted in Fig. 17 (c), was  $(1.0 \pm 0.28)$ %. level for  $SL_{Comet,nocorr}$  is close to the estimate from the generated

correction kernels in Appendix B, calculated as

$$\sum_{k,l} (\mathbf{K}_{far})_{k,l} + \overline{\mathbf{E}_{refl}} = 5.8\%$$
(13)

from the far field of the stable stray light  $\mathbf{K}_{far}$  and the mean value of the relative intensity variability of the ghost spot  $\overline{\mathbf{E}_{refl}}$ . The total reduction in stray light from the hardware improvement is approximately  $\frac{74\pm10}{63\pm10}$ %. The post-flight stray light correction applied to the data set before the hardware improvement is reducing the stray light level by approximately  $(84\pm6)$ %.

The uncertainties in Table Tab. 2 show a primary influence of the noise signal threshold on the total uncertainty. This is directly correlated to with the weak stray light signal, particularly in the case of the stray light measurement with the hardware improvement.

Figure 17. Different spectral spots for stray light characterization. (a) and (b) images show two spots similar to Fig. 6 without hardware optimization. (c) and (d) images show measurement results after a blackened fixed slit aperture was inserted in front of the spectrometer's entrance slit design. The sharp ghost vanishes nearly completely, and the stable stray light cone is decreased significantly. The images (a) + (c) are measured at  $\sim 1628 \, \mathrm{nm}$  and (b) + (d) at  $\sim 1661 \, \mathrm{nm}$ . The spectral offset is related to a turned grating during a readjustment of the MAMAP2D-Light system.

The stray light of the hardware-improved design was characterized at five different points at the FPA. From the characterization measurements, a stable stray light kernel was derived and used to contaminate the simulated spectra as described in Appendix E2. The resulting single read-out column noise after retrieving the stray light and random noise contaminated spectra is compared to the results for the stray light in the Comet 2.0 configuration in Tab. 3. The column noise for the single columns is reduced by  $\sim 50\%$ . The column noise after the proxy correction is reduced by  $\sim 15\%$ . An additional stray light correction in the hardware-improved design could reduce the single-column noise by 33% - 37% in the simulated case.

**Table 3.** Simulated single read-out column noise (CN) for different stray light scenarios of MAMAP2D-Light and added random noise; in the CoMet 2.0 (Comet) configuration, for the post-campaign hardware improvement (HWI) and an applied stray light correction (SL corr) for the HWI case.

|                | CH 4 | $CO_2$ | Proxy                    |
|----------------|-----------------|--------|--------------------------|
| CN Comet       | 0.78 %          | 0.63%  | $\underbrace{0.39\%}_{}$ |
| CN HWI  | 0.39%           | 0.32%  | $\underbrace{0.33\%}_{}$ |
| CN HWI SL corr | 0.26 %          | 0.20 % | 0.32%                    |

**665 7 Conclusions**

670

675

680

685

690

Stray light is causing a varying additive error signal in the spectra measured with the push-broom imaging spectrometer MAMAP2D-Light. In the WFM-DOAS retrieval, this varying error leads to pseudo-noise in the retrieved GHG columns. Based on stray light characterization measurements, a stray light correction was applied to a stray-light-contaminated campaign dataset. This allowed insights into the impact The amount of stray light in MAMAP2D-Light in the Comet 2.0 configuration is estimated as  $(5.6 \pm 0.39)$  %, which is in the same order of magnitude of the correctable stray light of 4.4 % in the SWIR channel of TROPOMI (Tol et al., 2018). The applied stray light correction for MAMAP2D-Light in the CoMet 2.0 configuration reduces the stray light level to  $(0.9 \pm 0.25)$  %.

In most cases of the proxy-corrected CH4 column anomalies, the proxy correction performs as well as the stray light correction based on the estimated column noise. This demonstrates the robustness of the commonly used proxy method in the 1.6 µm-band against stray light. However, the whole processing chain, from measured spectra to retrieved GHG fluxes. The proxy method is affected by ghost reflections in high-contrast scenes. In the case shown, the stray light characterization measurements were performed with a relatively low-cost measurement setup, with a Metcalf/Littman laser as a tunable monochromatic light source with insufficient side-mode suppression. To apply the measured data in the stray light correction, occurring detector effects and occurring side modes from the laser system were corrected or interpolated, correction was effective in preventing false column enhancements linked to a sharp-imaged ghost in the proxy corrected column anomalies. This highlights the need for end-to-end stray light characterization.

The In the non-proxy-corrected retrieved  $CH_4$  column anomalies, the stray light correction showed a substantial improvement in the column precision of the retrieved single shows a significant improvement of the column noise for along-track-binned measurements from  $\sim 0.64\%$  to  $\sim 0.33\%$ . For MAMAP2D-Light and instruments, that are using the proxy method in the  $1.6\,\mu\text{m}$ -band, the stray light correction enables to distinguish between mixed  $CH_4$  and column concentration anomalies. Within the  $/CO_2$  anomalies and potentially estimate emissions in such scenes, since the  $CH_4/CO_2$  proxy method is only feasible assuming a constant  $CO_2$  column. Furthermore, the single-readout column noise is reduced in the stray-light-corrected single columns compared to the proxy-corrected data, the column anomalies from  $\sim 0.47\%$  to  $\sim 0.41\%$ , thereby improving the instrument's detection limit. Furthermore, the strongly affected single columns highlight that applying a stray light correction had an impact on single scenes with high-intensity contrast and strong varying spectral surface reflectance. However, in is

mandatory for instruments that are not able to apply a  $\mathrm{CH_4/CO_2}$  proxy correction with both trace gas concentrations being retrieved from the same spectral band. This is the case for TROPOMI, or instruments with additional channels to improve the GHG concentration measurements in the majority of the seenes, the proxy method was able to correct stray-light-related errors. Analyses on artificial spectra showed that the column precision for stray-light-corrected near infrared or the shortwave infrared within the  $2\,\mu\mathrm{m}$ -band, as it is planned for CO2M (Sierk et al., 2021), CAMAP (Gerilowski et al., 2025), and already in use, Sentinel-5 (Landgraf et al., 2019), GOSAT-GW (Tanimoto et al., 2025) and MethaneAIR (Staebell et al., 2021). The derived  $\mathrm{CH_4}$  emissions from the single  $\mathrm{CH_4}$  column anomalies were highly under- or overestimated ( $-55\,\%$  or proxy-corrected data is limited by random noise sources,  $+61\,\%$ ) by the false column anomaly-pattern introduced by stray light in the cases studied in this paper. Applying the proxy method results in no significant change in the estimated emission rates relative to the stray-light-corrected cases.

695

700

705

710

Within the flux estimates for two measured landfill emissions, the stray-light- and the proxy-corrected concentrations provided similar emission rates. However, A significant amount of stray light ( $\sim 63\%$ ) in MAMAP2D-Light originated from reflective critical surfaces in the object plane of the spectrometer. These comprised the ferrule of the fiber-based 2D slit homogenizer and a non-blackened adjustable slit in the COMET 2.0 configuration. Here, the non-stray-light and non-proxy corrected data show error patterns, which are highly affecting the flux estimates. In fiber-based 2D slit homogenizer plays an important role, since the fibers have to be mounted in a ferrule, which is in the object plane and difficult to treat for non-reflectivity by a coating. The concept of the fiber-based 2D slit homogenizer has also been applied to the CO2I instrument of the CO2M mission, where the fibers are mounted in a silica ferrule (Hummel et al., 2022; Dussaux et al., 2025). For MAMAP2D-Light, stray light correction is a crucial step to eliminate the need for the proxy method and, therefore, to differentiate between individual components in mixed plumes. However, the proxy method is also utilized for light path and other instrumental corrections, which must be taken into account during data interpretation. The proxy method is only effective against stray light if both retrieved concentrations are measured in a hardware improvement was inserted, reducing the amount of stray light to  $(2.1 \pm 0.47 \%)$ , which is close to the same optical path and the trace-gas bands are closely spaced, as is the case for and in the 1.6 µm channel. However, for passive remote sensing instruments that offer additional spectral channels, e.g., NIR or SWIR-2, reducing and correcting stray light is essential to retrieve reliable atmospheric data. 2.4 % of stray light in the MetahneAIR SWIR channel (Staebell et al., 2021) . The hardware improvement reduced the stray-light-induced pseudo-noise by  $\sim 50\%$ . However, for the single columns, an additional stray light correction could reduce the column noise further by  $\sim 33 \,\%$ .

The impact of stray light was analyzed based on the WFM-DOAS retrieval. For other retrieval algorithms the impact of stray light might be different, since there are retrieval algorithms like FOCAL (Fast atmOspheric traCe gAs retrieval) (Reuter et al., 2017a, b), UoL-FP (The University of Leicester Full Physics) (Cogan et al., 2012) and the CH4 retrieval for the MethaneAIR instrument (Chan Miller et al., 2024), which consider an a constant additive offset in their atmospheric state vector.

Data availability. All level 1 and level 2 data can be provided by the corresponding authors upon request.

**Appendix A: Stray light terminology**

730

745

725 For this work, the stray light terminology is adapted from Fest (2013). Stray light is a collective term for unwanted redirected radiation that reaches the focal plane of an optical instrument. It occurs in all optical systems and can only be mitigated by design and manufacturing processes or corrected based on exact calibration measurements. The types of stray light can be described by their physical origin mechanisms.

*Ghost reflections* occur due to reflections and refraction, whose light paths obey Snell's law or the grating equation. Depending on the divergence of the resulting light path, ghost reflections can occur as sharply focused images.

Scatter stray light results from scattering on rough or particulate contaminated surfaces; since there are no perfectly smooth surfaces, all surfaces scatter light. Scatter stray light is described by the Bidirectional Scatter Distribution Function (BSDF), which is often referred to in terms of the scatter direction as the Bidirectional Reflection Distribution Function (BRDF) or the Bidirectional Transmittance Distribution Function (BTDF). The most common way to describe the BSDF of one or a series of surfaces is the Harvey model (described, e.g., in Peterson (2004) and Fest (2013)), which uses two to three parameters to describe a surface. Depending on the accuracy of the analytical model, it is rather complex to describe those surface parameters. Internal stray light, also called thermal background, originates from the thermal emission of the optical system itself. This becomes crucial in infrared applications, where the thermal radiation of the instrument results in stray light at the focal plane. The internal stray light is corrected by subtracting a background measurement, which is recorded with a turned-off or blocked intended light source.

Out-of-field stray light originates (OFSL) and out-of-band stray light (OBSL) originate from sources outside of the intended light path. However, the resulting stray light reaches the focal plane and contaminates the measured irradiances at the focal plane. In the spectrometer setup, the OFSL is defined in the spatial direction and the OBSL in the spectral direction.

[revised manuscript text omitted]

**Appendix G: Parameters for WFM-DOAS and flux retrieval**

Table G1. Parameters for RTM simulation, WFM-DOAS and flux retrieval for landfill scene in Fig. 10 and 14.

| Date                       | 11.09.2022                                                    |
|----------------------------|---------------------------------------------------------------|
| Time                       | 15:30 - 16:00 UTC                                             |
| Wavelength                 | 1500 nm - 1750 nm                                             |
| Wavelength resolution      | $0.01\mathrm{nm}$                                             |
| Flight altitude            | 29000 feet                                                    |
| Background CH 4 | 1906 ppb                                                      |
| Background $\mathrm{CO}_2$ | 413.7 ppm                                                     |
| Sun zenith angle           | 55.7°                                                         |
| Surface evaluation         | 172 m - 305 m                                                 |
| Albedo                     | 0.20                                                          |
| Aerosol scenario           | urban                                                         |
| Wind speed                 | $5.3\mathrm{ms^{-1}}$ (GoC, 2025)                             |
| Wind direction             | 209°                                                          |
| Fit window                 | <del>1575 nm</del> 1580.3 nm - <del>1677.5 nm</del> 1677.0 nm |
| Mean $c_f 	ext{ CH}_4$     | 0.80                                                          |
| Mean $c_f CO_2$            | 0.76                                                          |
| Spatial resolution         | $\sim 120 \times 120 \mathrm{m}^2$                            |
| Plume area                 | 1.5 km from source                                            |
| Background area            | 2 km from plume area                                          |
|                            |                                                               |

Table G2. Parameters for RTM simulation and WFM-DOAS for oil sand scene in Fig. 8.

| XXX 11                     | 1555 1500                                                                   |
|----------------------------|-----------------------------------------------------------------------------|
| Wavelength                 | $1555\mathrm{nm}$ - $1730\mathrm{nm}$                                       |
| Wavelength resolution      | $0.01\mathrm{nm}$                                                           |
| Flight altitude            | 26000 feet                                                                  |
| Background $\mathrm{CH}_4$ | 1894 ppb                                                                    |
| Background $\mathrm{CO}_2$ | 411.7 ppm                                                                   |
| Sun zenith angle           | 46.3°                                                                       |
| Surface evaluation         | $185 - 839 \mathrm{m}$                                                      |
| Albedo                     | 0.20                                                                        |
| Aerosol scenario           | urban                                                                       |
| Fit window                 | <del>1575 nm</del> 1580.3 nm - <del>1677.5 nm</del> 1677.0 nm |
| Mean $c_f 	ext{ CH}_4$     | 0.78                                                                        |
| Mean $c_f CO_2$            | 0.75                                                                        |
| Spatial resolution         | $\sim 120 \times 120 \mathrm{m}^2$                                          |

**Appendix H: Entrance slit Slit aperture exchange**

During the CoMet 2.0 mission, the adjustable slit uncoated adjustable slit aperture shown in Fig. H1 (a) with uncoated blades 3 was installed in the spectrometer in front of the entrance fiber ferrule, in Fig. 1. The variable slit uncoated adjustable slit aperture was exchanged with a fixed 200 µm slit (THORLABS S200ULK) consisting of blackened stainless steel, shown in Fig. H1(b) and (c). The slit. The blackened fixed slit aperture was wider than the fiber and is therefore acting as an additional aperture, as blackening the ferrule was not feasible. The slit blackened fixed slit aperture was glued on an anodized aluminum support. The side that shows in the direction of the optics is painted with NEXTEL Velvet Coating 811-21.

**940 Appendix I: Characterization measurements**

935

945

In order to retrieve trace gas column enhancements from the measured spectra, it is necessary to have a very good characterization of the instrument. The wavelength calibration and the instrumental spectral response function (ISRF) were measured with a Littman/Metcalf laser system (Lion System, by Sacher Germany, (Stry et al., 2006)), with a tunable wavelength from  $1600 \, \mathrm{nm} - 1750 \, \mathrm{nm}$  at a precision of  $0.05 \, \mathrm{nm}$  and a power of  $\sim 20 \, \mathrm{mW}$ . The actual wavelength of the laser was observed with a laser wavelength meter (671A, by Bristol) with an accuracy of  $\pm 0.2 \, \mathrm{pm}$  at  $1000 \, \mathrm{nm}$  for the NIR range from  $520 \, \mathrm{nm} - 1700 \, \mathrm{nm}$ . Flat field-Flat-field corrections were applied to account for PRNU errorsand losses from the grating. These corrections were performed using a broadband Quartz Tungsten Halogen lamp as a WLS. To achieve a homogeneous illumination, all sources were connected to an input port of an integrating sphere, with an inner diameter of  $5.3^{\circ}$  (Ophir, IS6-C, by Ophir).

Figure H1. (a) adjustable Blackened fixed 200 µm slit, which was assembled during aperture as replacement for the CoMet 2.0 mission at the entrance fiber ferrule (ferrule 2, uncoated adjustable slit aperture in Fig. 1) of the spectrometer3. (ba) side of the blackened fixed 200 µm slit as replacement for (a) aperture showing in direction of the ferrule. (eb) side of blackened fixed slit, that shows aperture showing in the direction of the ferrule collimator lens.

[revised manuscript text omitted]

Techniques, Environmental Science and Technology Letters, ISSN 2328-8930, https://doi.org/10.1021/acs.estlett.4c01063, 2025.

- Chan Miller, C., Roche, S., Wilzewski, J. S., Liu, X., Chance, K., Souri, A. H., Conway, E., Luo, B., Samra, J., Hawthorne, J., Sun, K., Staebell, C., Chulakadabba, A., Sargent, M., Benmergui, J. S., Franklin, J. E., Daube, B. C., Li, Y., Laughner, J. L., Baier, B. C., Gautam, R., Omara, M., and Wofsy, S. C.: Methane retrieval from MethaneAIR using the CO2 proxy approach: a demonstration for the upcoming MethaneSAT mission, Atmospheric Measurement Techniques, 17, 5429–5454, ISSN 1867-8548, https://doi.org/10.5194/amt-17-5429-2024, 2024.
- Chapman, J. W., Thompson, D. R., Helmlinger, M. C., Bue, B. D., Green, R. O., Eastwood, M. L., Geier, S., Olson-Duvall, W., and Lundeen, S. R.: Spectral and Radiometric Calibration of the Next Generation Airborne Visible Infrared Spectrometer (AVIRIS-NG), Remote Sensing, 11, 2129, ISSN 2072-4292, https://doi.org/10.3390/rs11182129, 2019.
- 1000 Clermont, L., C.Michel, Chouffart, Q., and Zhao, Y.: Going beyond hardware limitations with advanced stray light calibration for the Metop-3MI space instrument, Scientific Reports, 14, ISSN 2045-2322, https://doi.org/10.1038/s41598-024-68802-z, 2024.
  - Cogan, A. J., Boesch, H., Parker, R. J., Feng, L., Palmer, P. I., Blavier, J. L., Deutscher, N. M., Macatangay, R., Notholt, J., Roehl, C., Warneke, T., and Wunch, D.: Atmospheric carbon dioxide retrieved from the Greenhouse gases Observing SATellite (GOSAT): Comparison with ground-based TCCON observations and GEOS-Chem model calculations, Journal of Geophysical Research: Atmospheres, 117, ISSN 0148-0227, https://doi.org/10.1029/2012id018087, 2012.
  - Dussaux, A., Bazalgette Courrèges-Lacoste, G., Gaucel, J.-M., Garnier, T., Gaudin-Delrieu, ., Fayret, J.-P., Ouslimani, H., Charnier, J.-Y., Delclaud, Y., Lesschaeve, S., Spilling, D., te Hennepe, F., Förster, U., Strauss, S., Huber, G., Komadina, J., Reijnset, R., Pachot, C., Durand, Y., Pasquet, A., Chanumolu, A., Martinez Fernandez, M., Caleno, M., Meijer, Y., and Fernandez, V.: Copernicus CO2M: the mission for monitoring anthropogenic carbon dioxide from space: status of the payload at the start of the AIT phase, in: International Conference on Space Optics ICSO 2024, edited by Bernard, F., Karafolas, N., Kubik, P., and Minoglou, K., p. 38, SPIE, https://doi.org/10.1117/12.3072498, 2025.
  - Fest, E.: Stray Light Analysis and Control, SPIE, ISBN 9780819493262, https://doi.org/10.1117/3.1000980, 2013.
  - Gerilowski, K., Tretner, A., Krings, T., Buchwitz, M., Bertagnolio, P. P., Belemezov, F., Erzinger, J., Burrows, J. P., and Bovensmann, H.: MAMAP a new spectrometer system for column-averaged methane and carbon dioxide observations from aircraft: instrument description

- and performance analysis, Atmospheric Measurement Techniques, 4, 215–243, ISSN 1867-8548, https://doi.org/10.5194/amt-4-215-2011, 2011.
- Gerilowski, K., Windpassinger, R., Bovensmann, H., Borchardt, J., Krautwurst, S., Huhs, O., Richrath, M., Franke, J., Ohlendorf, J.-H., Thomssen, W., Burrows, J. P., Boesch, H., Meijer, Y., and Fehr, T.: CAMAP: a CO2 and methane airborne imaging spectrometer, in: International Conference on Space Optics ICSO 2024, edited by Bernard, F., Karafolas, N., Kubik, P., and Minoglou, K., p. 52, SPIE, https://doi.org/10.1117/12.3072751, 2025.
  - GoC: Government of Canada, Hourly Data Report for September 11, 2022, WINNIPEG A CS MANITOBA, https://climate.weather.gc.ca/climate\_data/hourly\_data\_e.html?hlyRange=2013-12-10/2025-04-06&dlyRange=1996-10-01/2025-04-05&mlyRange=1996-10-01/2007-11-01&StationID=27174&Prov=MB&urlExtension=\_e.html&searchType=stnProv&optLimit=specDate&StartYear=2022&EndYear=2022&selRowPerPage=25&Line=76&lstProvince=MB&timeframe=1&time=UTC&time=UTC&Year=2022&Month=9&Day=11#, last access 07.04.2025, 2025.

[revised manuscript text omitted]

- Rozanov, V., Rozanov, A., Kokhanovsky, A., and Burrows, J.: Radiative transfer through terrestrial atmosphere and ocean: Software package SCIATRAN, Journal of Quantitative Spectroscopy and Radiative Transfer, 133, 13–71, ISSN 0022-4073, https://doi.org/10.1016/j.jqsrt.2013.07.004, 2014.
- Sierk, B., Fernandez, V., Bézy, J.-L., Meijer, Y., Durand, Y., Bazalgette Courrèges-Lacoste, G., Pachot, C., Löscher, A., Nett, H., Minoglou, K., Boucher, L., Windpassinger, R., Pasquet, A., Serre, D., and te Hennepe, F.: The Copernicus CO2M mission for monitoring anthropogenic carbon dioxide emissions from space, in: International Conference on Space Optics ICSO 2020, edited by Sodnik, Z., Cugny, B., and Karafolas, N., p. 128, SPIE, https://doi.org/10.1117/12.2599613, 2021.
- 1075 Staebell, C., Sun, K., Samra, J., Franklin, J., Chan Miller, C., Liu, X., Conway, E., Chance, K., Milligan, S., and Wofsy, S.: Spectral calibration of the MethaneAIR instrument, Atmospheric Measurement Techniques, 14, 3737–3753, ISSN 1867-8548, https://doi.org/10.5194/amt-14-3737-2021, 2021.
  - Stry, S., Thelen, S., Sacher, J., Halmer, D., Hering, P., and Mürtz, M.: Widely tunable diffraction limited 1000 mW external cavity diode laser in Littman/Metcalf configuration for cavity ring-down spectroscopy, Applied Physics B, 85, 365–374, ISSN 1432-0649, https://doi.org/10.1007/s00340-006-2348-1, 2006.
  - Tanimoto, H., Matsunaga, T., Someya, Y., Fujinawa, T., Ohyama, H., Morino, I., Yashiro, H., Sugita, T., Inomata, S., Müller, A., Saeki, T., Yoshida, Y., Niwa, Y., Saito, M., Noda, H., Yamashita, Y., Ikeda, K., Saigusa, N., Machida, T., Frey, M. M., Lim, H., Srivastava, P., Jin, Y., Shimizu, A., Nishizawa, T., Kanaya, Y., Sekiya, T., Patra, P., Takigawa, M., Bisht, J., Kasai, Y., and Sato, T. O.: The greenhouse gas observation mission with Global Observing SATellite for Greenhouse gases and Water cycle (GOSAT-GW): objectives, conceptual framework and scientific contributions, Progress in Earth and Planetary Science, 12, ISSN 2197-4284, https://doi.org/10.1186/s40645-025-00684-9, 2025.
  - Tol, P. J., van Kempen, T. A., van Hees, R. M., Krijger, M., Cadot, S., Snel, R., Persijn, S. T., Aben, I., and Hoogeveen, R. W. M.: Characterization and correction of stray light in TROPOMI-SWIR, Atmospheric Measurement Techniques, 11, 4493–4507, ISSN 1867-8548, https://doi.org/10.5194/amt-11-4493-2018, 2018.

- van Hees, R. M., Tol, P. J. J., Cadot, S., Krijger, M., Persijn, S. T., van Kempen, T. A., Snel, R., Aben, I., and Hoogeveen, R. M.: Determination of the TROPOMI-SWIR instrument spectral response function, Atmospheric Measurement Techniques, 11, 3917–3933, ISSN 1867-8548, https://doi.org/10.5194/amt-11-3917-2018, 2018.
- Veefkind, J., Aben, I., McMullan, K., Förster, H., de Vries, J., Otter, G., Claas, J., Eskes, H., de Haan, J., Kleipool, Q., van Weele, M., Hasekamp, O., Hoogeveen, R., Landgraf, J., Snel, R., Tol, P., Ingmann, P., Voors, R., Kruizinga, B., Vink, R., Visser, H.,
   and Levelt, P.: TROPOMI on the ESA Sentinel-5 Precursor: A GMES mission for global observations of the atmospheric composition for climate, air quality and ozone layer applications, Remote Sensing of Environment, 120, 70–83, ISSN 0034-4257, https://doi.org/10.1016/j.rse.2011.09.027, 2012.
  - Zong, Y., Brown, S. W., Johnson, B. C., Lykke, K. R., and Ohno, Y.: Simple spectral stray light correction method for array spectroradiometers, Applied Optics, 45, 1111, ISSN 1539-4522, https://doi.org/10.1364/ao.45.001111, 2006.
- 1100 Zong, Y., Brown, S. W., Meister, G., Barnes, R. A., and Lykke, K. R.: Characterization and correction of stray light in optical instruments, in: Sensors, Systems, and Next-Generation Satellites XI, edited by Meynart, R., Neeck, S. P., Shimoda, H., and Habib, S., vol. 6744, p. 67441L, SPIE, ISSN 0277-786X, https://doi.org/10.1117/12.737315, 2007.

---

## Author Comment (AC3)

Author comment on "Impact of stray light on greenhouse gas concentration retrievals and emission estimates as observed with the passive airborne remote sensing imager MAMAP2D-Light" by Oke Huhs et al., Anonymous Referee #1

We want to thank Referee #1 for the comments regarding our manuscript. In this document, we provide our reply to the comments. The original comments made by the referee are typeset in red. The authors' answers are typeset in black and *italic*. A revised version of the manuscript, with highlighted changes, will be sent with this reply. The changes in the revised manuscript regarding the referee's comments are highlighted. If not stated differently, the figure numbers refer to those of the initial manuscript.

Huhs et al. present measurements from a new airborne hyperspectral pushbroom spectrometer targeting CO2 and CH4 in the 1.6 micron band. During their maiden campaign they discover that their data suffers from substantial stray light contamination, caused by reflections from the entrance slit. They implement a correction using the TROPOMI stray light correction algorithm to salvage their observations. The subject matter is within the scope of AMT. I believe it is of use to publish as a reference for future campaigns where the instrument is deployed. I think the organisation could be improved and the paper could be made more concise. Since the stray light correction method is not new, I would prefer the authors to shift the focus more towards the stray-light impacts on the retrieval, as weirdly the novel aspect could be the unintended experiment created by the adjustable slit. I am surprised by the reaction from the other reviewers - the paper is similar in quality to an average paper I have seen in AMT.

A general clarification of the scope of the manuscript has been added to the replies to the comments of Referee #2. We thank Referee #1 for the recommendations to improve the manuscript. Based on this, the following adaptations are performed to achieve an improved organisation as well as conciseness:

The analysis of retrieved data, which is the focus of the manuscript, has been highlighted in the introduction and conclusions.

Furthermore, the description of the stray light correction for the stray light components already addressed in the TROPOMI correction is moved to the Appendix.

The structure of the manuscript is improved by shifting Sect. 3 to Sect. 6.

**Sect. 6.3.2 is condensed to show more concise results.**

stated explicitly to avoid confusion.

surfaces in the spectrometer's object plane.

**Comments**

Pg 5 L 96: I believe the first reviewer (*Referee #2 in the following*) stated that the slit configuration was different during the campaign than during the stray light measurements. But it does say later that the measurements later were taken using the CoMet campaign conditions (Pg 7, L162), which I assume means the slit was in its fully open position. If this is the case perhaps it should be

Indeed, the stray light measurements for the applied stray light correction were performed in the CoMet 2.0 configuration. We think that Referee #2 is treating the adjustable slit as the specific component that causes the majority of stray light. We have clarified in the revised manuscript and in the answers to Referee #2 that the issue is generally caused by critical

To avoid confusion regarding the different usage of the term "slit," we define the various slits as:

- Entrance slit: describes the entrance of the imaging spectrometer in the theoretical design.
- Uncoated adjustable slit aperture: describes the specific component built in, but not in use during the CoMet 2.0 campaign
- Blackened fixed slit aperture: describes the specific low reflective component inserted for the post-campaign hardware improvement
- entrance slit unit (ESU) as a term for the spectrometers' setup in the entrance slit.

Pg 6 L132: The notation of Eq. 1,2 is a bit weird. First, since the error term appears on the RHS, I believe the LHS of Eq 1 actually corresponds to the authors  $R^{\text{mea}}_{\text{lambda}}$ . Secondly the function  $R^{\text{mea}}_{\text{lambda}}$  appears on the LHS and RHS. The one on the right should be denoted something like  $R^{\text{RTM}}_{\text{lambda}}$  or something like that because it corresponds to the simulated spectrum from the radiative transfer simulation. Then  $R^{\text{mod}}_{\text{lambda}}$  would be equation 1 without the error term, making the objective function in Equation 2 make sense. A normal way to write equation 2 would be

$$\underset{\mathbf{x}, \mathbf{c}}{\operatorname{arg \; min} \; } ||\ln R_{\lambda}^{mea} - \ln R_{\lambda}^{mod}(\mathbf{c}, \mathbf{a})||^{2}$$

**Agree, Equation 2 is adapted**

Pg. 6 ln 136: Is there any particular reason why the measured radiances are log-transformed? In terms of radiance, the noise errors are approximately normally distributed, so the errors in this case will be log-normally distributed. This means that the least squares estimate is no longer the maximum likelihood one.

The measured spectra are log-transformed, which allows for linearizing the changes in concentration based on Lambert-Beer's law and adds robustness against multiplicative errors using broadband polynomials (Buchwitz, Rozanov, & Burrows, 2000). This also allows us to perform the least-squares fit to the model, since the parameters  $\mathbf{c}$  and  $\mathbf{a}$ , which are optimized in the WFMD approximation shown in Eq. (1), are in the exponent of the Lambert-Beer law. This is based on the DOAS method as described in detail in (Platt, 2008).

Indeed, the log-transformed radiances lead to a log-normal distributed random error. However, as long as the errors ( $\Delta R$ ) are minor compared to the radiance (R), the logarithm  $ln(R+\Delta R)$  can be linearized by a Taylor expansion to  $ln(R) + \Delta R/R$ .

Pg 9, L206 - It might be helpful to add some arrows to Figure 5 labeling the features due to the sources of stray light discussed in Section 4.1

**Agree, labels are inserted to Fig. 5 and Fig. 18 (Fig. 6 and Fig. 17 in the revised manuscript)**

Pg 12, L278 - There seems to be quite a bit more variability in the intensity of the ghost spot than what is modeled using the first order polynomial fit. It also looks real, in the sense that there is correlation between the calibration measurements. Is there a reason to believe this is an error that needs to be smoothed, or could some non-parametric method like radial basis function interpolation be used instead? It might be worth testing if something like this would improve the test case in Fig. 9.

Tests with a higher-order 2D polynomial lead to significantly decreased quality in the test case, due to the unknown edge areas. However, tests using a non-parametric method based on the suggestion have shown no improvement in the scene. As stated in the manuscript in Line 289 (Line 605 in the revised manuscript), the available data for the Eref estimation should have been optimized for a larger dynamical range to increase the SNR in the measurements.

Pg 14, L325: PSF is already being used for point spread function. The acronym for profile scaling factor should be changed to something else.

Since the term Profile Scaling Factor is used more frequently, the Point Spread Function will be used without the acronym and will be combined with the ISRF to form the PRF (point response function).

Pg 16, L372: As a general rule it is best to organize figures sequentially in the order that they are first referenced in the paper.

**Agree, the position of the figure is adapted in the revised version of the manuscript.**

Pg 17, L380: The proxy uncertainty seems too close to the CH4 uncertainty in the stray light corrected case. If the error is completely uncorrelated with the CO2 column it should be ~0.44. 0.34 implies that the errors in CH4 and CO2 columns retrieved are almost perfectly correlated. That is really weird to me - is the instrument random noise really that low? It could be a typo as the value for the ratio looks to be about 0.44 in Fig. 13.

The noise is indeed lower; in Fig. 11, the noise is estimated from along-track binned concentration data (five measurements), whereas the data shown in Fig.13 is for a single read-out. We have clarified the difference more explicitly in the title of Sect 6.3.2 (Sect. 4.4.2 of the revised manuscript).

For the single-read-out simulated case, where the spectra are contaminated with random noise, the quadratic summation of the single-column errors is equal to the error of the proxy, as expected for uncorrelated data.

Pg 23: Perhaps Fig. 14/15 and Fig 16/17 can be combined into two panel figures

It was decided to move Fig. 14 (Fig. 10 in the revised manuscript) to Sect. 6.3, where the scene is mentioned first. Therefore, the figures are kept as single figures.

**Minor Corrections**

L53: building -> building

**Corrected in revised version**

L80: "and thereby, allowing to operate the instrument at ambient temperature." -> and thereby allowing the instrument to operate at ambient temperature.

**Corrected in revised version**

L107: adpated -> adapted

**Corrected in revised version**

L 115 and L399:: Quadratic sounds weird in this context. I think saying "square" is fine.

**Agree, changed in revised version**

L 402 Sect. D -> Appendix D.

**Corrected in revised version**

Summarized and further applied changes in the revised manuscript:

- Rewritten the introduction and conclusions with a focus on the retrieved data
  Highlight similarities with CO2M, MethaneAIR, GOSAT-GW, Sentinel 5 and TANGO
- Differentiate clearly between OFSL and OBSL
- Description of the stray light correction for the stray light components already addressed in the TROPOMI correction, as well as minor adaptations regarding detector effects, and laser side-modes, are moved to the Appendix. The newly developed approach for OBSL stays in the main text.
- The structure of the manuscript is improved by shifting Sect. 3 to Sect. 6 (Sect. 4.1 in the revised manuscript) and combining the stray light characterization with the correction to Sect. 3 in the revised manuscript.
- Changed "real" measurements to measurements
- Made Sect. 6.3.2 more consistent to clear results regarding single read-out column noise, OFSL and OBSL, including Fig. 13 (14 in the revised manuscript).
- Added analysis of the impact of the stray light level after hardware improvement on the simulated data, in Sect. 8, Sect. 6 in the revised manuscript. Furthermore, the handling of the noise in the stray light level estimation in Sect. 8 has been improved, which leads to increased stray light levels. We added an estimation of the remaining stray light level after the applied stray light correction for the CoMet2.0 configuration to this section.
- A Bug in the stray light correction code was found and eliminated, which is leading to non-significant minor differences in Fig. 13, 15,17 (in the revised manuscript: Fig. 13, 15, 16)

**References:**

Buchwitz, M., Rozanov, V. V., & Burrows, J. P. (June 2000). A near-infrared optimized DOAS method for the fast global retrieval of atmospheric CH4, CO, CO2, H2O, and N2O total column amounts from SCIAMACHY Envisat-1 nadir radiances. *Journal of Geophysical Research:*Atmospheres, 105, 15231–15245. doi:10.1029/2000jd900191

Platt, U. (2008). *Differential optical absorption spectroscopy.* (J. Stutz, Hrsg.) Berlin, [Germany] ;: Springer Berlin Heidelberg. doi:10.1007/978-3-540-75776-4

**Impact of stray light on greenhouse gas concentration retrievals and emission estimates as observed with the passive airborne remote sensing imager MAMAP2D-Light**

Oke Huhs1, Jakob Borchardt1, Sven Krautwurst1, Konstantin Gerilowski1, Heinrich Bovensmann1, Hartmut Bösch1, and John P. Burrows1

1University of Bremen, Institute of Environmental Physics, Otto-Hahn-Allee 1, 28359 Bremen

**Correspondence:** Oke Huhs (oke.huhs@iup.physik.uni-bremen.de)

**Abstract.** MAMAP2D-Light is an airborne passive remote sensing imaging push-broom spectrometer developed at the Institute for Environmental Physics at the University of Bremen to measure determine atmospheric methane (CH4) and carbon dioxide (CO2) column anomalies to quantify point-source emissions in the 1.6 µm-band to quantify point-source emissions. In its initial version, as flown in 2022 in Canada, a significant stray light level of 4\% 5.6\% of the measured signal has been observed post-campaign, causing apparent error patterns in the retrieved CO2 and CH4 column anomalies. In this paper, we report the successful application of a stray light correction developed for the instrument. Measurement data collected during an airborne campaign in 2022 in Canada offer the unique opportunity to investigate the end-to-end impact of stray light and its correction on the retrieved CO2 and CH4 column anomalies, as well as the retrieved derived emission rates. Stray light caused apparent error patterns in the retrieved columnanomaly maps We successfully developed and applied a straylight correction to the instrument and investigated its impact on the CH4/CO2 proxy method, the CH4 column, and derived point-source emissions. In nearly all cases, applying the  $\frac{CH_4/CO_2}{CH_4/CO_2}$  proxy method reduced the stray-light-related column errors below the column noise, leading to comparable final emission rate estimates for proxy-only and stray-light-corrected data. The derived emission rates for the proxy-retrieval with and without straylight corrected spectra are comparable, proving for the first time the capabilities of the  $CH_4/CO_2$  proxy method to correct stray light-related artifacts. In this paper, we additionally investigate the special scene contrast conditions impact on the CH4 total column retrieval for a high contrast scene condition under which the correction by applying the proxy method is no longer sufficient. Following the initial campaign in 2022, the stray light was reduced by  $\sim$  75% by the implementation of post-campaign stray light characterization and analysis revealed that a significant fraction of stray light was attributed to reflective surfaces in the object plane of the spectrometer. Based on these findings, the total stray light was reduced by  $\sim 63\%$  by implementing a hardware modification from 2023 onward.

**20 1 Introduction**

Passive remote sensing has become one of the cornerstones for monitoring the most critical greenhouse gases (GHGs), carbon dioxide (CO2) and methane (CH4), in the Earth's atmosphere to determine anthropogenic and natural GHG emissions. The spectral absorption features of the GHGs in reflected sunlight are exploited to retrieve the corresponding atmospheric GHG

concentrations. However, depending on the instrument's spatial and spectral resolution, the distance from the source, and the source area, surface emissions introduce only minor changes in the measured absorption features compared to the absorption features due to the accumulated background concentrations in the total atmospheric column. For Therefore, the spectra have to be measured very precisely to enable accurate emission estimates, therefore, which is translated into strict instrument-dependent specifications of for the accuracy of the spatial and spectral calibration accuracy of the measured spectra required, allowing to retrieve the according atmospheric GHG columns with the required accuracy and precision.

For an instrument with a given spatial and spectral resolution, the required column precision is determined by the detection limit required for the envisaged emission estimates (Jacob et al., 2022; Pandey et al., 2023). For example, the CH4 column single-measurement precision for SCIAMACHY, the first instrument dedicated applying solar backscatter absorption spectroscopy to remote sensing of GHGs from space aboard the EnVISAT ENVISAT satellite, was planned to achieve 1% (Bovensmann et al., 1999). For its successor, TROPOMI (TROPOspheric Monitoring Instrument), the goal precision was tightened to 0.6% for a single measurement (Veefkind et al., 2012). To achieve this precision, high-quality instrument characterizations, minimizing and correcting radiometric errors, are required the calibration measurements, therefore, need to characterize radiometric errors precisely to implement corrections minimizing their impact on the measured spectra.

A significant contributor to the radiometric error is stray light, which arises from reflections and scattering processes that are not intended in the optical design. The definition and terminology of stray light are adapted from Fest (2013). Stray light distorts the measured spectra with a continuum-dependent error (Tol et al., 2018) and is most prominent in high-contrast scenes, e.g., in mixed scenes with dark land surfaces and bright clouds. However, the stray-light-induced error signal depends on the overall intensity distribution of the light paths within the system. The spectrally, spatially, and intensity-dependent error signal introduces error patterns in the retrieved concentrations and can be misinterpreted as column enhancements and, in certain cases, even as plumesemission plumes from point source emitters. Therefore, it is essential to mitigate stray light within the optical system. Effective mitigation of stray light involves minimizing it through an optimized optical design, usually via simulations during the design phase of an instrument, and correcting it during data processing. The latter uses so-called based on stray light kernels estimated from stray light characterization measurements.

Various methods for For the stray light correctionhave been developed. A widely used approach for non-imaging spectrometers, described by Zong et al. (2006), exploits matrix operations to correct both spectrally and spatially stable and variant stray lightsimultaneously. This method requires dense characterization measurements or interpolations across the entire focal plane array. For imaging spectrometers—, the presence of an additional spatial axis increases the size of the correction matrix by the squared number of spatial pixels and therefore the required computational resources massively (Zong et al., 2007). For the SWIR channel of TROPOMI, it is essential to analyze the origin of the stray light. The origin and behaviour is highly dependent on the instrument type (Clermont et al., 2024; Baumgartner et al., 2025). For grating spectrometers with a relatively narrow spectral range, Tol et al. (2018) introduced a tailored method that separates stable and variant stray light components. The stable component is corrected using an iterative deconvolution method, which has also been applied for the MethaneAIR instrument (Staebell et al., 2021). The variant component is addressed through a spatial transformation based on the variability of the stray light depending on the origin. This method leverages the fast Fourier transform (FFT) for deconvolution, enabling

50

55

a near-real-time the globally stable stray light correction. from the variable stray light due to, e.g., ghosts.

This work focuses on spectra measured with For this work, the stray light correction from Tol et al. (2018) is adapted to MAMAP2D-Light (Methane Airborne MAPper 2D Light), a lightweight airborne remote sensing push-broom imaging grating spectrometer built at the Institute for Environmental Physics (IUP) Bremen. Besides satellite-based instruments, airborne remote sensing spectrometers offer spatially high-resolution measurements with similar spectral specifications. This allows intercomparison between both measurement platforms, airborne and spaceborne, for the used GHG concentration retrieval algorithms and the final emission rate retrievals. In contrast to satellite-based instruments, airborne spectrometers can be recalibrated and improved during their lifetime.MAMAP2D-Light is buildung built on concepts established with the MAMAP (Methane Airborne MAPper) instrument (Gerilowski et al., 2011) and. It is designed to measure CH4 and CO2 column anomalies in the 1.6 µm band, exploiting the CO2 (Krings et al., 2011) or the CH4 (Krings et al., 2013) proxy method. The data set used in this study was collected during the CoMet 2.0 Arctic mission, which took place in summer 2022 in Canada1. The GHG concentrationswere, which is also established for MethaneAIR (Chan Miller et al., 2024), and planned for the GOSAT-GW (Observing SATellite for Greenhouse gases and Water cycle) (Tanimoto et al., 2025) and Sentinel-5 (Landgraf et al., 2019) missions.

For remote sensing of GHGs, airborne remote sensing spectrometers provide smaller ground scenes compared to satellite-based observations with similar spectral properties. This offers the opportunity to distinguish between real enhancements and concentration anomalies introduced by instrument, atmospheric, or surface-related error sources (Gerilowski et al., 2011). Therefore, airborne demonstrators, such as the MAMAP2D (Methane Airborne MAPper 2D) and the CAMAP (CO2 And Methane Airborne maPper) (Gerilowski et al., 2025) instrument currently developed as the airborne demonstrator for the CO2M-Mission (Sierk et al., 2021), are a valuable complement for satellite missions to understand the impact of instrumental error signals, as stray light, on the retrieved data products with real measurements.

This work focusses on CH4 concentrations, retrieved using the WFM-DOAS (Weighting Function Modified Differential Absorption Spectroscopy) method Krings et al. (2011), in combination with the CH4/CO2 proxy method, in the following abbreviated as proxy method, which has been proven to deliver reliable CH4 and column anomalies on the local scale in the past (Krings et al., 2011, 2013; Krautwurst et al., 2017, 2021, 2024). A post-campaign stray light characterization with a Littman/Metcalf tunable laser (Stry et al., 2006) revealed a significant stray light contamination in the campaign dataset.

Therefore, a post-flight (Krings et al., 2013; Krautwurst et al., 2017, 2021, 2024; Borchardt et al., 2025). For airborne remote sensing, the sensitivities of the proxy method to deviations in the atmospheric state and observation geometry have been analyzed through simulations by Krings et al. (2011). Besides the advantages of the proxy method, it is only feasible if the proxy concentration (in this case, the CO2) remains constant; otherwise, it either underestimates or overestimates the concentrations by definition.

In this work, the adapted stray light correction was applied, exploiting the previously performed characterization measurements.

The stray-light-contaminated campaign data is applied to measured spectra collected with MAMAP2D-Light during the CoMet

<sup>1https://comet2arctic.de/ last access: 11.02.2025

- 2.0 Arctic mission, which took place in summer 2022 in Canada1. The campaign dataset is contaminated with 5.6% of stray light, which is only slightly higher than the estimated correctable stray light of 4.4% in the TROPOMI SWIR channel (Tol et al., 2018).
- Applying the stray light correction to the CoMet 2.0 data set provides a unique opportunity to investigate the impact of the stray light correction on real measured data and the capabilities of the proxy method in the presence of stray light in the stray light and its correction in the entire processing chain from the measured spectra to the retrieved GHG concentrations and the derived emission rate estimates -with and without the applied proxy method. This especially examines the capabilities of the proxy method to correct stray-light-induced errors in the single CH4 column.
- In Sect. 2, the instrument design of MAMAP2D-Light is introduced. The impact of stray light on the used WFM-DOAS retrieval is described in Sect. 4.1. The stray light stray light characterization measurements of MAMAP2D-Light are summarized in Sect. 3.1. From the characterization measurements, a and the applied stray light correction algorithm is applied are summarized in Sect. 3.13. The correction is applied to real-measured and simulated spectra, and the impact on the WFM-DOAS method as well as the impact on the retrieved concentration maps is analyzed in Sect. 4. From the concentration maps, the resulting CH4 emission rates of two measured landfill plumes are analyzed based on in Sect. 5 with respect to the impact of the applied stray light correction and the proxy correctionin Sect. 5. With the post-flight stray light characterization measurements, the origin of the majority of the stray light has been localized and mitigated finally improved by a hardware improvement shown modification summarized in Sect. 6. The expected error in column noise after the hardware improvement is determined.

**2 MAMAP2D-Light instrument**

MAMAP2D-Light is an airborne passive remote sensing instrument for observing atmospheric CO2 and CH4 columns using 110 infrared spectroscopy solar backscatter absorption spectroscopy in the short-wave infrared around 1.6 µm. The MAMAP2D-Light instrument, shown in Fig. 1, is a push broom imaging spectrometer with a planar reflective grating. It weighs approximately 43 kg and fits into an underwing pod of a motor glider aircraft (e.g., Diamond HK 36-TTC ECO, (Borchardt et al., 2025)). MAMAP2D-Light covers the wavelength range from 1559 nm to 1690 nm with a spectral resolution of approximately 1.1 nm. It comprises a front optic, an optical fiber bundle (see Fig. 2), an entrance slit unit (ESU), two different lenses, serving as 115 collimator and camera optics, a planar reflection grating and an infrared detector (AIM SWIR384). The detector deployed has a Mercury Cadmium Telluride (MCT) focal plane array (FPA) comprising 384 pixel × 288 pixel and a pixel pitch of  $24 \, \mu \text{m} \times 24 \, \mu \text{m}$ . Within the spectrometer, the FPA is oriented in a way that the spectral axis is along the 384 pixel axis, which results in a spectral oversampling of  $\sim 3$  pixel, while the spatial axis is along the 288 pixel axis. The FPA is cooled to approximately 150 K with a linear single-piston cooler to reduce the internal thermal dark current of the MCT. The spectral cut-off 120 was adapted from  $\sim 2.5\,\mu m$  to  $\sim 1.8\,\mu m$  by the manufacturer to reduce the sensitivity to thermal background radiation from the optical bench and mounting elements and thereby, allowing to operate the instrument allowing the instrument to operate at ambient temperature.

<sup>1https://comet2arctic.de/ last access: 11.02.2025

**Figure 1.** Schematic optical setup of the MAMAP2D-Light instrument. The reflected sunlight reflected from the Earth's surface is imaged by the front optic on onto the input of a fiber bundle with 36 fibers stacked in Ferrule 1, where each fiber corresponds to a single spatial ground scene. The radiation enters the spectrometer block through the fibers in Ferrule 2. The fibers are stacked perpendicular to the optical bench of the spectrometer block. To adjust the linewidth of the spectrometer, a Slit aperture is placed in the entrance focal plane. The radiation is dispersed and imaged at the 2D detector with the two optics and the grating. The area of a single fiber together with the focal length of the front optics defines the instantaneous field of view (IFOV). 28 fibers are imaged at the detector, determining the field of view (FOV).

MAMAP2D-light measures scattered sunlight from the Earth's surface, which is imaged via the front optical lens onto an optical fiber bundle with 36 rectangular single fibers stacked in a ferrule, see Fig. 2, acting as a 2D-slit-homogenizer (Hummel et al., 2022; Gerilowski et al., 2025). Each fiber has a fiber core of  $\sim 300 \, \mu \text{m} \times 100 \, \mu \text{m}$  in spatial and spectral direction, respectively. The outer dimensions of the fibers with cladding are  $\sim 315 \, \mu \text{m} \times 175 \, \mu \text{m}$ . Due to the orientation of the detector, only 28 of the 36 fibers are imaged at onto the detector, resulting in 28 across-track ground scenes observed by the instrument. The entrance slit unit ESU of the spectrometer comprises the ferrule on the fiber bundle's second end, an adjustable slit, an uncoated adjustable slit aperture (Acton Research, Model SPS-716-1S), an 1500 nm cut-on optical order sorting filter and a shutter unit. The light entering the spectrometer is collimated by a lens collimator system (the collimator) with a focal length of  $F_c = 300 \, \text{mm}$  and an aperture of F/N = 3.5. The dispersed collimated light from the grating is then focused on the detector by the camera lens optics with  $F_o = 200 \, \text{mm}$  and F/N = 2.4. The angle of the optical axes between the lenses is  $32^\circ$ . The grating deployed in MAMAP2D-Light is a ruled plane grating with  $300 \, \text{lines mm}^{-1}$  and a nominal blaze angle of  $17.5^\circ$ , which is operated at the  $-1^{\text{st}}$  order.

125

130

135

During the CoMet 2.0 campaign, the installed slit aperture in Fig. 1 was adjustable. The slit, an uncoated adjustable slit

Figure 2. Image of a section of aligned fibers within the aluminum ferrule from Fig. 1. The fiber core dimension is  $\sim 300 \, \mu m \times 100 \, \mu m$ .

aperture, as shown in Fig. H1 (a), consists 3, consisting of two uncoated steel blades. Initially, it was intended to adjust the ISRF and the spectral oversampling with the slit aperture. However, due to misaligned fibers (see Fig. 2) in the entrance ferrule, the slit was not used and was therefore uncoated adjustable slit aperture was left open to its maximum using only the fiber geometry as the entrance slit.

140

The swath of the instrument is defined by the focal length of the input front objective  $F_f = 25 \,\mathrm{mm}$ ; in combination with the

**Figure 3.** Uncoated adjustable slit aperture as mounted during the CoMet 2.0 mission at the entrance fiber ferrule (ferrule 2, in Fig. 1) of the spectrometer. (a) side of uncoated adjustable slit aperture showing in direction of the ferrule. (b) side of uncoated adjustable slit aperture showing in direction of the collimator lens.

FPA spatial pixel count, the pixel size and the imaging ratio  $F_o/F_c$ . The total swath is defined by the fully imaged fibers on the FPA since the fiber bundle length is larger than the detector width. The resulting across-track field of view (FOV) for the full detector is about 23.3°. However, the exact field of view is defined by the length of the input fibers fully imaged on the detector. This leads to a real FOV of 22.6°. For the CoMet 2.0 campaign (this work), MAMAP2D-Light was integrated on a Gulfstream G 550 (HALO, High Altitude and LOng Range Research Aircraft, operated by the DLR, Deutsches Zentrum für Luft- und Luft- Raumfahrt). With a flight altitude of  $\sim 8 \, \mathrm{km}$  above ground level, the FOV of 22.6° led to a swath width of  $\sim 3.5 \, \mathrm{kmat}$  a above ground level, with a sampling of 28 spatial fibers, corresponding to an across-track spatial resolution of  $\sim 120 \, \mathrm{m}$ . The along-track ground scene size is dependent on the flight speed, the integration exposure time, and the number of binned ground scenes, and was adpated adapted to  $\sim 120 \, \mathrm{m}$  by binning  $\sim 5 \, \mathrm{single}$  measurements for the flights in Canada during CoMet 2.0. Due to MAMAP2D-Lights' compact dimensions and weight of approximately 43 kg, the system also fits

into an underwing pod of a motor glider aircraft (e.g., Diamond HK 36-TTC ECO) and was successfully deployed in this configuration for an airborne campaign in Australia (Borchardt et al., 2025).

**3 The impact of stray light in the WFM-DOAS retrieval**

165

175

The GHG anomalies are retrieved from the measured spectra using the WFM-DOAS method. This method does not consider any corrections for an additive error signal, which is the expected type of error resulting from stray light contamination. This section describes the WFM-DOAS retrieval and the impact of an additive offset within the WFM-DOAS retrieval.MAMAP2D-Light measures the spectra of the sunlight passing through the atmospheric column. Depending on the depth of the absorption bands of the corresponding GHG, the anomalies of GHG concentrations are retrieved from the spectra using the WFM-DOAS method. The WFM-DOAS retrieval was initially developed for the spaceborne SCIAMACHY instrument by Buchwitz et al. (2000). The algorithm was later adapted for the airborne measurement geometry by Krings et al. (2011) for the MAMAP instrument. Krautwurst et al. (2024) describe the retrieval algorithm's latest version as applied to MAMAP2D-Light data.Based on Lambert Beer's law, a calculated RTM at a wavelength  $R_{\lambda}^{mod}(\bar{\mathbf{c}})$  for a state of the atmosphere represented by the model state vector  $\bar{\mathbf{c}}$ , can be modulated to get the RTM at the state  $\bar{\mathbf{c}}$  of the measurement  $R_{\lambda}^{mod}(\bar{\mathbf{c}})$ . The weighting functions,  $W_{\lambda,c_j}$ , describe the change of radiance due to a change of the respective parameter j. An additional low-order polynomial  $P_{\lambda}$  with a free parameter vector  $\bar{\mathbf{c}}$  approximates slow spectral variations due to scattering or spectral surface reflectance, which have to be considered but are not quantified. This results in the following equation:

$$\ln R_{\lambda}^{mod}(\mathbf{c},\mathbf{a}) = \ln R_{\lambda}^{mod}\left(\overline{\mathbf{c}}\right) + \sum_{j} W_{\lambda,\overline{c}_{j}} \frac{c_{j} - \overline{c}_{j}}{\overline{
[revised manuscript text omitted]
 PRF area of MAMAP2D-Light is larger than in the TROPOMI SWIR spectral band, leading to larger areas of saturation during the stray light characterization, surrounded by a 1-pixel wide area of blooming2 around the saturated pixels. To get reliable data in the saturation area, the exposure time was increased in specifically adapted

<sup>2Blooming occurs due to photogenerated charges within a saturated pixel, which are not fully collected by the pixel's read-out electronics. The leftover charges are then collected by the neighbouring pixels.

**Figure 5.** (a) Photograph of the entrance aluminium fiber ferrule, Ferrule 2 in Fig. 1, with a black anodized mount. A single fiber is illuminated with the setup shown in Fig. 4 and a white light source. The illuminated fiber is visible as a red dot. (b) zoom in on the fiber stack.

smaller steps. The dark signal level, increasing linearly with the exposure time due to thermal radiation, constrained the highest exposure time to 3000 ms. The dark signal for each point was measured for each exposure time after a complete set of exposure times with illumination, by shutting off the laser. The measurements were flat-field corrected, where the fibers' cladding areas (displayed as dark lines in the spectral direction in Fig. E1) were interpolated by fitting a 2-dimensional 3rd-order polynomial to the fiber core signal. The dark current corrected data showed patterns related to a detector effect, which were most prominent for higher exposure times with increased saturation. The patterns were corrected using a data-driven approach, which is shown in detail in Appendix D.

The measured signals at one position for all exposure times were merged into a single two-dimensional frame —by the following procedure: Therefore At each exposure time, saturated and blooming-contaminated pixels had to be were filtered out. Blooming³ occurred in the measurements at higher exposure times due to the saturation of the directly illuminated pixels. Due to the CMOS-based read-out electronics of the detector, only the directly neighbouring pixels of a saturated pixel are affected by blooming. For merging, each non-saturated and non-blooming-contaminated pixel value at the highest exposure time was selected. The full merged frame was finally normalized to the integral of the signal over all pixels. The merged frames of the stray light characterization measurements for MAMAP2D-Light in the CoMet 2.0 configuration for four different spot positions are shown in Fig. 6. The measurements revealed several stray light and non-stray-light-related artifacts which are discussed in the next section.

**3.2 Stray light contributors components**

245

250

The stray light contributors were separated into a spectrally and spatially *invariant* part independent of and a spatially *variable* part depending on the position of the illuminated spot. Within the measurements, the relative spectral position of the *variable* stray light was constant. The description for the stray light sources uses the terminology defined in Appendix A, where the stray light is classified in different orders. With each stray light process (e.g., scattering, reflection, etc.) in a light path, the

<sup>3Blooming occurs due to photogenerated charges within a saturated pixel, which are not fully collected by the pixel's read-out electronics. The leftover charges are then collected by the neighbouring pixels.

Figure 6. Different spectral and spatial spots from the stray light characterization of MAMAP2D-Light in the CoMet 2.0 configuration. (a) and (c) were recorded at  $\sim 1628$  nm,(b) and (d) at  $\sim 1661$  nm. The horizontal line at the right-hand side of the illuminated spot is due to the not completely suppressed laser side modes (LSM) of the laser used. The vertical line through the illuminated spots is caused by light from outside the instrument entering the spectrometer through the fibers. A sharp ghost appears spatially mirrored but spectrally in a constant offset from the initial spot. Further, the spectral and spatial invariant stray light (stable stray light; SSL) cone around the illuminated spot is shown in all images.

order is increased, starting with the intended light path as the 0th order.

260

265

The observed stray light was separated into two components based on their position relative to the illuminated spot, dependent on the position of the illuminated spot on the detector: The spectrally and spatially *invariant* stray light part always occurred at the same relative position, while the spatially *variable* part changed its relative position. Nevertheless, the relative spectral position of the *variable* stray light was constant.

[revised manuscript text omitted]

The reflection Kernel  $\mathbf{K}_{refl}$  was determined from the relative positions of the ghost spot to the originally illuminated spot, see Fig. 6. In the spectral direction, the relative offset  $x_{refl}$  was constant. In the spatial direction, the ghost spot was mirrored and shifted by  $y_{refl}$  from the center. A spot search algorithm defined  $x_{refl}$  and  $y_{refl}$  based on the relative distances between ghost and origin spots' barycenters.  $\mathbf{K}_{refl}$  shifted the frame to the ghost position. Since the ghost spot was a sharp image,  $\mathbf{K}_{refl}$  would be ideally a single pixel with the value 1 at  $x_{refl}$  and  $y_{refl}$ . However, due to floating values, the signal pixel was initially set to the nearest integer value and afterward shifted by the decimal points using the "shift" function of the "scipy.ndimage" package (version 1.13.1) in python. Thus, the signal in  $\mathbf{K}_{refl}$  had an area of 2 pixels  $\times$  2 pixels. The relative intensity variability of the ghost spot and the origin spot is represented by  $\mathbf{E}_{refl}$  and was generated from the wavelength grid, instrumental response function (Appendix II and I2) and the stray light characterization measurements using the equation:

$$\mathbf{E}_{refl} = \frac{\mathbf{S}_{refl}^{R}(x - x_{refl}, y - y_{refl})}{\mathbf{S}_{origin}(x, y)}.$$

335  $S_{origin}$  represents the signal of the origin spot and  $S_{refl}$  is the corresponding signal of the reflected spot, which is shifted by the corresponding  $x_{refl}$  and  $y_{refl}$  values. The respective signal levels within a fiber were determined by the mean intensity of

the spot, defined by a half-maximum threshold. The R-operator is mirroring the y-axis. Due to the sparse data availability for  $\mathbf{E}_{reft}$ , a two-dimensional first-order polynomial fit was deployed to fill the data gaps, shown in Fig. B2. Higher orders in the fit function led to a stronger variability of the values in the unknown edges. The RMS of the relative fit residuals was  $\sim 8\,\%$ . A more accurate  $\mathbf{E}_{reft}$  estimation would either require a denser grid of stray light measurements or, e.g., wavelength grid measurements with an increased dynamical range, as done for the stray light characterization measurements, see Sect. 3.1. The second term in Eq. B2  $(\mathbf{K}_{reft}*(\mathbf{E}_{reft}-\mathbf{J})^R)$  represents the amount of reflected stray light in the frame. However, the slit was not perfectly aligned vertically, and due to the smile effect3 slightly curved. This distortion needed to be corrected before the mirroring operation was performed and reversed before subtraction. The correction was achieved by shifting each row by a value  $x_{smile}$ , row. This value was determined by the difference between the barycenter of each row from a measurement of a full slit and the median of all barycenters from the same measurement. The resulting  $\mathbf{x}_{smile}$  array for all rows was the median for each row from the wavelength grid and instrumental response function (refer to Appendix II and I2) measurements. The correction is only valid due to the relatively small wavelength dependency of the diffraction angle defined by the groove frequency of the grating with 300 lines mm $^{-1}$ .

**3.2 Out-of-field stray light in MAMAP2D-Light**

340

345

350

355

360

365

Due to the extension of  $K_{stable}$  and the offset of  $K_{reft}$  from the center, the out-of-field stray light (OFSL) contaminated the Out-of-field stray light (OFSL) and OBSL contaminated the spatial and spectral edges of the measured frameframes. To consider the spectral OFSL (or out-of-band stray light) OBSL in the correction, the spectral axis of the measured frame was extrapolated with an extended RTM, as used in Sect. 4. The RTM was fitted to each row of the dark current and flat field corrected frame and scaled with a polynomial ( $3^{rd}$ -order) and spectral shift parameter within the spectral range of MAMAP2D-Light. The extended spectra were then derived from scaling and shifting the full RTM range with the derived fit parameters. This method of extrapolation gives only an estimation of the signal level of the spectral OFSL provides only an estimate of the OBSL signal level, and the expected impact of estimation uncertainty is discussed in Appendix E3. It is important to note that the surface spectral reflectance and the aerosol scenario have an impact on the signal level of the spectral OFSL OBSL and would affect the correction quality even with in a perfectly characterized system.

[revised manuscript text omitted]

$$\quad \ln R_{\lambda}^{mod}(\mathbf{c}, \mathbf{a}) = \ln R_{\lambda}^{mod}(\overline{\mathbf{c}}) + \sum_{j} W_{\lambda, \overline{c}_{j}} \frac{c_{j} - \overline{c}_{j}}{\overline{c}_{j}} + P_{\lambda}(\mathbf{a}) + \varepsilon_{\lambda}$$

$$\tag{1}$$

The values of the parameters j (e.g. the GHG concentrations) building the state vector of interest c are retrieved from a measured spectrum  $R_{\lambda}^{mea}$  by a least squares fit with the fit parameters c and a. To separate the noise contribution of the stray light from other noise sources, simulated synthetic measurements are included in the analysis

$$\underset{\mathbf{a}, \mathbf{c}}{\operatorname{arg\,min}} \left\| \ln R_{\lambda}^{mea} - \ln R_{\lambda}^{mod}(\mathbf{c}, \mathbf{a}) \right\|^{2} \tag{2}$$

Stray light is radiation deviating from the intended light path and illuminating the FPA at unintended positions. The position of the intended path in the focal plane is called the origin position, and the unintended position is called the target position (For terminology, see again Appendix A). Stray light causes an additive error signal (or zero-level offset) *e* at the focal plane. The error signal occurs in the target spectrum and, by being absent, also in the origin spectrum. The fitting in Eq. 2 is then performed to a measured spectrum of

410
$$\ln R_{\lambda}^{mea,real} = \ln(R_{\lambda}^{mea} + e).$$
 (3)

While the polynomials  $P_{\lambda}(\mathbf{a})$  are introduced to catch, among others, instrumental error signals, they are in fact additive components to the logarithm of the radiance in Eq. 1, and therefore, scalable multiplicative factors of the radiance. Consequently,

in WFM-DOAS, the additive offset e is tried to be compensated for by a multiplicative scaling factor of the polynomial. This introduces a signal level-dependent scaling error, which leads, in the case of a positive error signal, to a shrinking of the absorption line depths relative to the continuum. The corresponding fitting parameter  $c_i$  then "sees" shallower trace gas absorption bands, which leads to an underestimation of the retrieved column anomaly. Therefore, an additive offset can not be observed in the spectral residuals  $e_{\lambda}$  of the fit, except for areas in the spectral window without any trace gas-related absorption bands, e.g., pure Fraunhofer-Lines.

MAMAP2D-Light is designed to quantify GHG anomalies relative to the background concentrations. As the normalization of the retrieved columns to the background is performed in the post-processing, described in detail in Sect. 4.2, a constant additive offset would not impact the precision of the retrieved column anomalies. However, the impact of stray light depends on the radiation of the source and the amount of the intended radiation within the target spectrum. Thus, scenes with inhomogeneous albedo, spectral surface reflectance, or aerosol scenario result in decreased precision in the retrieved column anomalies.

**425 4.2 Data processing**

430

The column anomalies were retrieved with the airborne WFM-DOAS method, which is described briefly in Sect. 4.1 and in detail by Krautwurst et al. (2024). The retrieval delivers column anomalies from the trace gases of interest as profile scaling factors (PSF) of atmospheric profiles at the mean state of the atmosphere during the measurements using an RTM calculated with SCIATRAN 3.8 (Rozanov et al., 2014). The spectra were dark current corrected, radiometric calibrated by a calibrated sphere measurement, see Sect. 3.1, and wavelength calibrated. The retrieved data was filtered using a root-mean-squared (RMS, see in Sect. 4.1) threshold of the fit residuals to assess the quality of the fit. To account for signal intensities exceeding the linearity range of the detector and to keep a sufficient signal-to-noise ratio, a maximum and minimum signal threshold was applied.

The retrieved column data showed a nonlinear dependency on the detector filling. This phenomenon has already been observed for MAMAP data and is discussed by Krautwurst et al. (2017). For MAMAP2D-Light, the nonlinear dependency for each spatial sample was corrected with a data-driven approach analogous to that developed for MAMAP. A low-order polynomial (2nd - 3rd order) was fitted to the column data over the detector filling for one spatial fiber over a single flight leg. The column data was then normalized by the fit result.

Typically (Krings et al., 2013; Krautwurst et al., 2017, 2024), the proxy method is used to minimize the impact of light-path errors, like multi-scattering or instrumental error. The CH4 proxy is the ratio of the retrieved CH4-PSF and the CO2-PSF, assuming a constant CO2 concentrations over the measurements area:

$$CH_{4,proxy} = CH_{4,psf}/CO_{2,psf}. (4)$$

However, the proxy method underestimates mixed plume signals either underestimates or overestimates plume signals if the  $CO_{2,psf}$  is not constant, e.g., due to  $CO_2$  emissions nearby or background changes due to large-scale gradients in the  $CO_2$  concentration. Therefore, in this work, the non-proxy corrected single columns are also analyzed in more detail.

Depending on the altitude at which the  $\mathrm{CH_4}$  plume and therefore the concentration perturbation is located, the WFM-DOAS retrieval has varying sensitivities. This sensitivity is described by the altitude-dependent averaging kernel AK(z) (Krings et al., 2011). For the CoMet 2.0 data, it was computed for each ground scene, considering its respective surface elevation and assuming that all enhancements are located below the aircraft. Based on the AK(z), conversion factors  $c_f$  were derived used for correction of the retrieved PSFs:

$$CH_{4,rel} = (CH_{4,psf} - 1) \cdot c_f \tag{5}$$

The column data was georeferenced using the aircraft position and attitude and the surface elevation,—. The procedure is described in detail by in Krautwurst et al. (2024).

**4.3 Stray light in high contrast scenes column anomalies**

450

465

Initial results of the CoMet 2.0 campaign dataset revealed an error pattern in the proxy-corrected CH4 column anomalies for the scene shown in Fig. 8. The data was processed using the retrieval, RTM, and orthorectification parameters shown in Tab. G2. In the non-stray-light-corrected concentration map in Fig. 8 (c), significantly enhanced CH4 column anomalies are shown. The intensity map in Fig. 8 (b) revealed a high contrast scene, where the surfaces consist of highly reflective sand, low-reflective vegetation, and a nearly non-reflective lake. The CH4 column anomaly pattern resembles the mirrored sand surface, which aligns with the mirrored ghost seen in Fig. 6. After the stray light correction, the structures in the CH4 column anomalies were reduced, but not erased. This is related to the not accurately known reflection intensity distribution Erefl shown in Fig. B2 and Seet.-discussed in Appendix B2.

The stray light correction also reduced further negative column anomalies, which were located in at ground scenes with low intensity compared to the across-track neighbouring ground scenes. In this scene, with applied proxy correction, the total column noise was reduced from 0.40% to 0.33% by the stray light correction.

The impact of the stray light mainly depends on the signal level and distribution of the origin and the signal level of the target spectrum. In Fig. 9, the reflected and the stable error signal for a target spectrum are shown. The reflected stray light introduces a more structured and different curved error signal, whereas the stable stray light is smoother and follows the curve of the target spectrum. The proxy method is unable to correct imbalanced error contamination in the  $CO_2$  and  $CH_4$  bands. Due to the different absorption line depths, a general imbalance of the sensitivity to a zero-level offset is given; if the zero-level offset varies spectrum, the imbalance can be compensated or amplified. The shown target spectrum is the corresponding synthetic spectrum, which is generated as described in Appendix E, of an enhanced pixel in Fig. 8 (b), which is caused by the contamination of the reflected stray light.

**4.4 Stray light as source for pseudo-noise in column anomalies**

The stray light introduced stray-light-introduced error patterns in the concentration maps can be observed as pseudo-noise in the column noise estimate of the real measured retrieved column anomalies. Therefore, in the following, the variation of the column anomalies is analyzed based on a flight leg, shown in Fig. 10, over an area dominated by urban and agricultural

**Figure 8.** Measured scene over high reflective sand and low reflective vegetation. The surface in RGB is shown in (a). (b) shows the intensity in the SWIR measured with MAMAP2D-Light. The non-stray-light-corrected and proxy-corrected processed data is shown in (c), with a column noise of 0.40 %. The stray-light-corrected data in (d) has a reduced column noise of 0.33 %. The RGB map is provided by © OpenStreetMap, accessed using Cartopy.

surfaces, where plume signatures were masked. A plume signal extending from a landfill was masked for the calculation of the column noise. The flight leg was chosen due to the strong variations in surface reflectance. Further, based on the real measured frames, synthetic frames were generated and artificially contaminated with stray light and random noise to simulate the different error types individually in the processing chain. The concentration anomalies were retrieved using the parameters shown in Tab. G1.

**4.4.1 Column noise in real measured data**

480

485

The column noise of the real measured column anomalies was estimated from the standard deviation of the source-free background area. In Fig. 11, the distribution and the column noise of the non-proxy-corrected single columns and proxy-corrected columns with and without applied stray light correction are shown. The column noise of the non-proxy-corrected single

Figure 9. Separated stray light (SL) error signal from the sharp ghost reflex (reflected SL) and the stable kernel (stable SL) with y-axis on the left for a target spectrum with y-axis on the right in a simulated frame.

**Figure 10.** Retrieved CH4 anomalies at the Brady Road Landfill. The results with applied proxy correction (CH4 / CO2) are shown in the left column, and the single CH4 column results are shown in the right column. Non-stray-light-corrected results are shown in the top row, and stray-light-corrected results in the bottom row. The blue arrow marks the wind direction. The map underneath is provided by Google Earth (Image © Airbus 2025, © Maxar Technologies 2025).

columns is significantly improved after the stray light correction. However, after the proxy correction, the stray light correction has no significant impact on the column noise. When comparing the standard deviations of the single CH4 column with the stray light correction to the proxy corrected column, the noise of the single CH4 column is marginally lower. The increased column noise after the proxy correction is associated with the division of two independent quantities contaminated with random noise. However, the impact of the random noise is already reduced by along-track binning of five measurements. The impact of spatial stray light is depicted through a correlation of the mean retrieved column anomalies with the mean

490

495

Figure 11. Histograms of the retrieved single  $CO_2$  (a) and  $CH_4$  (b) and the proxy  $(CH_4/CO_2)$  corrected (c) column data as profile scaling factors (PSFs). The Distributions show data without (blue) and with (orange) stray light correction.

intensity of a measured frame, as shown in Fig. 12. The intensity of each wavelength-calibrated and dark-current-corrected spectrum is derived from the as the mean intensity of the continuum between  $1620.5 \,\mathrm{nm}$  and  $1623.0 \,\mathrm{nm}$ , in digital numbers [DN]. Similar to the column noise in Fig. 11, the correlation of the mean column enhancements with the mean intensities is corrected by the proxy method. However, after the stray light correction, the correlation in the single  $\mathrm{CO}_2$  and  $\mathrm{CH}_4$  columns decreases significantly. The effectiveness of the stray light correction differs between the  $\mathrm{CH}_4$  and  $\mathrm{CO}_2$  columns, impacting the

shown correlation of the proxy and the stray-light-corrected data. This variance may be linked to the  $\overline{OFSLOBSL}$  correction outlined in Sect. 3.1. Due to the location of the used fit-window (1575 nm - 1677.5 nm) on the detector, the CH4 band is more affected by the  $\overline{OFSLOBSL}$  than the CO2 band. The position of the CO2 and CH4 bands are marked in Fig. 9.

500

505

Figure 12. 2D-histograms (color) of the average retrieved single  $CO_2$  and  $CH_4$  and the proxy ( $CH_4/CO_2$ ) corrected column data dependent of the profile scaling factors per frame on the y-axis and the average intensity of the frame on the x-axis. (a) mean  $CO_2$  PSF with no applied stray light (SL) correction shows a strong correlation with the mean intensity. After the stray light correction in (b), the correlation vanishes. The correlation is also visible in the mean  $CH_4$  PSF data in (c) and vanishes after the stray light correction (d). (e) and (f) show the correlation for the proxy-corrected data, where the stray light correction has only a minor impact compared to the single columns.

**4.4.2 Column Comparison of single read-out column noise in with simulated data**

The column noise in Fig. 11 after the stray light correction and after the proxy correction stays in the same range of 0.34 with along-track binning to get quadratic ground scenes. Further, the histograms have a normal-distribution-like shape, which is an indication that the total noise is dominated by random noise square ground scenes stays in the same range of 0.34%. To have the possibility to separate the stray light introduced error from random noise error sourcesdifferent stray light contributors, synthetic spectra were generated and contaminated with different error signals from stray light, including OBSL and OFSL, and random noise, as described in Sect. Appendix E.

The different cases and resulting single read-out column noise values are shown in Fig. 13. The first two cases are for comparing the real measured results with the simulated results and show a close correlation regarding the stray light introduced error, which is discussed in detail in Appendix E2. The differences between the real non-stray-light corrected data and the synthetic frames with artificially added stray light and random noise contamination highlight that the simulated data is close but not fully accurate. However, a more accurate model would require precise knowledge of the scene's acrosol scenario, spectral surface reflectance and the out-of-field signal. Nevertheless, the synthetic frames can show the impact of the different error contributors uncertainties are 1-\sigma, estimated using a bootstrap method as the standard deviation from randomly selecting 10% of the datasets 1000 times. In all cases, an applied stray light correction leads to an increased column precision of the single columns compared to the proxy correction. The first three cases show the column noise of the retrieved single-measured cases, depending on the applied stray light correction.

For the stray-light-corrected measured column anomalies, the column noise of the single  $CH_4$  column is  $\sim 13\%$  smaller compared to the column noise after proxy correction. This is related to the division of two random noise-contaminated values. The stray light correction for the simulated data is Considering the OBSL during the stray light correction of the real measured data, with the differences described in the following. The stable stray light is corrected with a perfectly known stable kernel, from Fig. B1. The spatial OFSL is neglected, and the spectral OFSL is perfectly known during the shows no significant impact, i.e., within the uncertainties, on the single columns and proxy-corrected measured data. In the simulated data, where the OFSL, OBSL, and random noise are considered in the contamination, considering OBSL in the stray light correction leads to a minor improvement of 6.5% in the proxy corrected and 2.3% in the single  $CO_2$  concentration data compared to the case of non-considered OBSL in the correction. Due to the spatial shift, see Fig. 7 (b), of the reflected stray light, the lower two fibers are not considered in the column noise estimation of the  $CH_4$  band on the detector, considering the OBSL improved the  $CH_4$  column noise by 18.6%.

The stray light correction for the simulated data with random noise contamination shows a similar impact as for the real measurement; the noise of the single In the simulated spectra, the OFSL is randomly added but not considered in the correction. When comparing the stray light corrected case with the noise-only contaminated case, the leftover OFSL increases the column noise by 7.1% in the proxy corrected column and 9.1% and columns is significantly reduced, and the proxy corrected column is only marginally affected by the stray light correction. In general, the proxy method is effective for reducing the stray-light-introduced pseudo-noise, which can be seen in the stray-light-only contaminated data. The column noise of the proxy method is higher than 10.7% for the single and CH4 columns in non-stray-light-contaminated data (whether after stray light correction or a priori without stray light) when random noise was added to the data. This is expected due to the division of two independent noise-contaminated values, and CO2 column.

By contaminating the synthetic spectra only with the random noise, the resulting proxy single read-out column noise limit is at  $\sim 0.32\,\%$ . This is the theoretically achievable column precision for the analyzed measurement. Without considering the spectral OFSL in the stable stray light correction, the column noise is  $\sim 0.22\,\%$ . With perfect knowledge of the spectral OFSL, this value decreases to  $\sim 0.11\,\%$ . The leftover column noise is related to the spatial OFSL. For the real measured data, the spectral OFSL is approximated, and the impact is discussed in detail in Appendix E3. The proxy-corrected single read-out

column noise of the real-measured data is  $\sim 0.46$  %, which is  $\sim 44$  % higher than the theoretically achievable minimum. The lower column noise in the real single and columns, compared to the proxy column noise, is a hint that the discrepancy with the simulation is caused by discrepancy between the measured and the simulated data is caused by not considering other (pseudo-)noise originating from sources in the simulation, which are, 
[revised manuscript text omitted]
. 17. Further, the length of the blackened fixed slit aperture blocks the origin of the OFSL. In this section, the amount of stray light after the hardware improvement (SLHWI) is compared to the stray light levels in the CoMet 2.0 configuration with (SLComet.corr) and without applied stray light correction (SLComet.nocorr).

This reduction The amount of stray light was determined by single spot measurements,—where the stray light cone is fully imaged at the detector, as illustrated in Fig. 17 (a) and (c). Both the non-stray-light-related horizontal laser artifact and the vertical line originating from in front of the fiber bundle were masked for the comparison, see Sect. 3.1. Furthermore, a noise threshold was applied. To reduce the random noise in the stray light measurement data while preserving the stray light

Python package (version 0.18.1)) was applied. The denoising weight was estimated from the standard deviation (SD) of the signal in an illumination-free area (SD(Sdark)). Following the denoising, non-stray-light-related signals were excluded via an additional threshold, calculated by the mean and the standard deviation in an (SD) in the illumination-free area (Signal arr of the denoised image (Sdark dn and SD(Sdark dn)). All values below the noise signal threshold were set to zero. The prepared image was normalized to the integrated signal of the entire FPA. The stray light was separated from the origin spot with a threshold value relative to the maximum intensity of the frame. The spot-size threshold is the average relative minimum value of the instrumental response functions (TSRFmin), as described in Appendix 12. The relative stray light is the ratio of the integrated stray light to the total integrated signal. The resulting stray light levels are shown in Tab. 1. The total uncertainty was calculated by the quadratic addition of the uncertainties for the spot size, the noise threshold, and the size and position of the horizontal and vertical masks. The single uncertainties were calculated by disturbing the variables corresponding contributing parameter by the values stated in Tab. 2.

**Table 1.** Relative stray light levels of MAMAP2D-Light in the CoMet 2.0 (Comet) configuration, with applied stray light correction  $(SL_{Comet,corr})$  and without applied stray light correction  $(SL_{Comet,nocorr})$  and for the post-campaign hardware improvement  $(SL_{HWI})$ . The total error is calculated by a quadratic addition of the single components.

| Case                | relative stray light level |
|---------------------|----------------------------|
| $SL_{Comet,nocorr}$ | $(5.6 \pm 0.39)\%$         |
| $SL_{Comet,corr}$   | $(0.9 \pm 0.25)\%$         |
| $SL_{HWI}$          | $(2.1 \pm 0.47)\%$         |

Table 2. Parameters for stray light quantification and absolute uncertainties for the relative stray light of MAMAP2D-Light in the Comet CoMet 2.0 (Comet) configuration, with applied stray light correction ( $SL_{Comet.corr}$ ) and without applied stray light correction ( $SL_{Comet.corr}$ ), and for the post-campaign hardware improvement ( $HWISL_{HWI}$ ). The total error is calculated by a quadratic addition of the single components.  $\sigma$ -SD represents the standard deviation.

| Uncertainty source     | start-Initial value                                                                            | disturbance Disturbance                           | $\Delta_{Comet}$ $\Delta_{SL_{Comet,new}}$ |
|------------------------|------------------------------------------------------------------------------------------------|---------------------------------------------------|--------------------------------------------|
| Spot size threshold    | $\overline{ISRF_{min}}$                                                                        | $\pm 3\sigma_{ISRF_{min}} \pm 3SD(ISRF_{min})$    | ±0.104 % ±0.099 %                          |
| Noise Denoising weight | $2SD(S_{dark})$                                                                                | $\pm 1SD(S_{dark})$                               | ±0.185 %                                   |
| Signal threshold       | $\overline{Signal_{dark}} + 3\sigma_{Signal_{dark}} \overline{S_{dark,dn}} + 3SD(S_{dark,dn})$ | $\pm \sigma_{Signat_{dark}} \pm 1SD(S_{dark,dn})$ | $\pm 0.293\% \pm 0.319\%$                  |
| Mask size and position | defined by hand                                                                                | $\pm 1$ pixel                                     | ±0.076 % ±0.094 %                          |

The stray light in the Comet 2.0 configuration, shown in Fig. 17 (a), was  $(3.9 \pm 0.32)$  %. The stray light after the hardware improvement, as depicted in Fig. 17 (c), was  $(1.0 \pm 0.28)$  %. level for  $SL_{Comet,nocorr}$  is close to the estimate from the generated

correction kernels in Appendix B, calculated as

$$\sum_{k,l} (\mathbf{K}_{far})_{k,l} + \overline{\mathbf{E}_{refl}} = 5.8\%$$
(13)

from the far field of the stable stray light  $\mathbf{K}_{far}$  and the mean value of the relative intensity variability of the ghost spot  $\overline{\mathbf{E}_{refl}}$ . The total reduction in stray light from the hardware improvement is approximately  $\frac{74\pm10}{63\pm10}$ %. The post-flight stray light correction applied to the data set before the hardware improvement is reducing the stray light level by approximately  $(84\pm6)$ %.

The uncertainties in Table Tab. 2 show a primary influence of the noise signal threshold on the total uncertainty. This is directly correlated to with the weak stray light signal, particularly in the case of the stray light measurement with the hardware improvement.

Figure 17. Different spectral spots for stray light characterization. (a) and (b) images show two spots similar to Fig. 6 without hardware optimization. (c) and (d) images show measurement results after a blackened fixed slit aperture was inserted in front of the spectrometer's entrance slit design. The sharp ghost vanishes nearly completely, and the stable stray light cone is decreased significantly. The images (a) + (c) are measured at  $\sim 1628$  nm and (b) + (d) at  $\sim 1661$  nm. The spectral offset is related to a turned grating during a readjustment of the MAMAP2D-Light system.

The stray light of the hardware-improved design was characterized at five different points at the FPA. From the characterization measurements, a stable stray light kernel was derived and used to contaminate the simulated spectra as described in Appendix E2. The resulting single read-out column noise after retrieving the stray light and random noise contaminated spectra is compared to the results for the stray light in the Comet 2.0 configuration in Tab. 3. The column noise for the single columns is reduced by  $\sim 50\%$ . The column noise after the proxy correction is reduced by  $\sim 15\%$ . An additional stray light correction in the hardware-improved design could reduce the single-column noise by 33% - 37% in the simulated case.

**Table 3.** Simulated single read-out column noise (CN) for different stray light scenarios of MAMAP2D-Light and added random noise; in the CoMet 2.0 (Comet) configuration, for the post-campaign hardware improvement (HWI) and an applied stray light correction (SL corr) for the HWI case.

|                | CH 4          | $CO_2$ | Proxy                    |
|----------------|--------------------------|--------|--------------------------|
| CN Comet       | 0.78 %                   | 0.63%  | $\underbrace{0.39\%}_{}$ |
| CN HWI  | $\underbrace{0.39\%}_{}$ | 0.32%  | $\underbrace{0.33\%}_{}$ |
| CN HWI SL corr | 0.26%                    | 0.20 % | 0.32%                    |

**665 7 Conclusions**

670

675

680

685

690

Stray light is causing a varying additive error signal in the spectra measured with the push-broom imaging spectrometer MAMAP2D-Light. In the WFM-DOAS retrieval, this varying error leads to pseudo-noise in the retrieved GHG columns. Based on stray light characterization measurements, a stray light correction was applied to a stray-light-contaminated campaign dataset. This allowed insights into the impact The amount of stray light in MAMAP2D-Light in the Comet 2.0 configuration is estimated as  $(5.6 \pm 0.39)$  %, which is in the same order of magnitude of the correctable stray light of 4.4 % in the SWIR channel of TROPOMI (Tol et al., 2018). The applied stray light correction for MAMAP2D-Light in the CoMet 2.0 configuration reduces the stray light level to  $(0.9 \pm 0.25)$  %.

In most cases of the proxy-corrected CH4 column anomalies, the proxy correction performs as well as the stray light correction based on the estimated column noise. This demonstrates the robustness of the commonly used proxy method in the 1.6 µm-band against stray light. However, the whole processing chain, from measured spectra to retrieved GHG fluxes. The proxy method is affected by ghost reflections in high-contrast scenes. In the case shown, the stray light characterization measurements were performed with a relatively low-cost measurement setup, with a Metcalf/Littman laser as a tunable monochromatic light source with insufficient side-mode suppression. To apply the measured data in the stray light correction, occurring detector effects and occurring side modes from the laser system were corrected or interpolated, correction was effective in preventing false column enhancements linked to a sharp-imaged ghost in the proxy corrected column anomalies. This highlights the need for end-to-end stray light characterization.

The In the non-proxy-corrected retrieved  $CH_4$  column anomalies, the stray light correction showed a substantial improvement in the column precision of the retrieved single shows a significant improvement of the column noise for along-track-binned measurements from  $\sim 0.64\%$  to  $\sim 0.33\%$ . For MAMAP2D-Light and instruments, that are using the proxy method in the  $1.6\,\mu\text{m}$ -band, the stray light correction enables to distinguish between mixed  $CH_4$  and column concentration anomalies. Within the  $/CO_2$  anomalies and potentially estimate emissions in such scenes, since the  $CH_4/CO_2$  proxy method is only feasible assuming a constant  $CO_2$  column. Furthermore, the single-readout column noise is reduced in the stray-light-corrected single columns compared to the proxy-corrected data, the column anomalies from  $\sim 0.47\%$  to  $\sim 0.41\%$ , thereby improving the instrument's detection limit. Furthermore, the strongly affected single columns highlight that applying a stray light correction had an impact on single scenes with high-intensity contrast and strong varying spectral surface reflectance. However, in is

mandatory for instruments that are not able to apply a  $\mathrm{CH_4/CO_2}$  proxy correction with both trace gas concentrations being retrieved from the same spectral band. This is the case for TROPOMI, or instruments with additional channels to improve the GHG concentration measurements in the majority of the seenes, the proxy method was able to correct stray-light-related errors. Analyses on artificial spectra showed that the column precision for stray-light-corrected near infrared or the shortwave infrared within the  $2\,\mu\mathrm{m}$ -band, as it is planned for CO2M (Sierk et al., 2021), CAMAP (Gerilowski et al., 2025), and already in use, Sentinel-5 (Landgraf et al., 2019), GOSAT-GW (Tanimoto et al., 2025) and MethaneAIR (Staebell et al., 2021). The derived  $\mathrm{CH_4}$  emissions from the single  $\mathrm{CH_4}$  column anomalies were highly under- or overestimated ( $-55\,\%$  or proxy-corrected data is limited by random noise sources,  $+61\,\%$ ) by the false column anomaly-pattern introduced by stray light in the cases studied in this paper. Applying the proxy method results in no significant change in the estimated emission rates relative to the stray-light-corrected cases.

695

700

705

710

Within the flux estimates for two measured landfill emissions, the stray-light- and the proxy-corrected concentrations provided similar emission rates. However, A significant amount of stray light ( $\sim 63\%$ ) in MAMAP2D-Light originated from reflective critical surfaces in the object plane of the spectrometer. These comprised the ferrule of the fiber-based 2D slit homogenizer and a non-blackened adjustable slit in the COMET 2.0 configuration. Here, the non-stray-light and non-proxy corrected data show error patterns, which are highly affecting the flux estimates. In fiber-based 2D slit homogenizer plays an important role, since the fibers have to be mounted in a ferrule, which is in the object plane and difficult to treat for non-reflectivity by a coating. The concept of the fiber-based 2D slit homogenizer has also been applied to the CO2I instrument of the CO2M mission, where the fibers are mounted in a silica ferrule (Hummel et al., 2022; Dussaux et al., 2025). For MAMAP2D-Light, stray light correction is a crucial step to eliminate the need for the proxy method and, therefore, to differentiate between individual components in mixed plumes. However, the proxy method is also utilized for light path and other instrumental corrections, which must be taken into account during data interpretation. The proxy method is only effective against stray light if both retrieved concentrations are measured in a hardware improvement was inserted, reducing the amount of stray light to  $(2.1 \pm 0.47 \%)$ , which is close to the same optical path and the trace-gas bands are closely spaced, as is the case for and in the 1.6 µm channel. However, for passive remote sensing instruments that offer additional spectral channels, e.g., NIR or SWIR-2, reducing and correcting stray light is essential to retrieve reliable atmospheric data. 2.4 % of stray light in the MetahneAIR SWIR channel (Staebell et al., 2021) . The hardware improvement reduced the stray-light-induced pseudo-noise by  $\sim 50\%$ . However, for the single columns, an additional stray light correction could reduce the column noise further by  $\sim 33 \,\%$ .

The impact of stray light was analyzed based on the WFM-DOAS retrieval. For other retrieval algorithms the impact of stray light might be different, since there are retrieval algorithms like FOCAL (Fast atmOspheric traCe gAs retrieval) (Reuter et al., 2017a, b), UoL-FP (The University of Leicester Full Physics) (Cogan et al., 2012) and the CH4 retrieval for the MethaneAIR instrument (Chan Miller et al., 2024), which consider an a constant additive offset in their atmospheric state vector.

Data availability. All level 1 and level 2 data can be provided by the corresponding authors upon request.

**Appendix A: Stray light terminology**

730

745

725 For this work, the stray light terminology is adapted from Fest (2013). Stray light is a collective term for unwanted redirected radiation that reaches the focal plane of an optical instrument. It occurs in all optical systems and can only be mitigated by design and manufacturing processes or corrected based on exact calibration measurements. The types of stray light can be described by their physical origin mechanisms.

*Ghost reflections* occur due to reflections and refraction, whose light paths obey Snell's law or the grating equation. Depending on the divergence of the resulting light path, ghost reflections can occur as sharply focused images.

Scatter stray light results from scattering on rough or particulate contaminated surfaces; since there are no perfectly smooth surfaces, all surfaces scatter light. Scatter stray light is described by the Bidirectional Scatter Distribution Function (BSDF), which is often referred to in terms of the scatter direction as the Bidirectional Reflection Distribution Function (BRDF) or the Bidirectional Transmittance Distribution Function (BTDF). The most common way to describe the BSDF of one or a series of surfaces is the Harvey model (described, e.g., in Peterson (2004) and Fest (2013)), which uses two to three parameters to describe a surface. Depending on the accuracy of the analytical model, it is rather complex to describe those surface parameters. Internal stray light, also called thermal background, originates from the thermal emission of the optical system itself. This becomes crucial in infrared applications, where the thermal radiation of the instrument results in stray light at the focal plane. The internal stray light is corrected by subtracting a background measurement, which is recorded with a turned-off or blocked intended light source.

Out-of-field stray light originates (OFSL) and out-of-band stray light (OBSL) originate from sources outside of the intended light path. However, the resulting stray light reaches the focal plane and contaminates the measured irradiances at the focal plane. In the spectrometer setup, the OFSL is defined in the spatial direction and the OBSL in the spectral direction.

[revised manuscript text omitted]

with  $J_0$  as the measured contaminated frame and  $K_{far}$  the far-field of the  $K_{stable}$ , see Sect. Appendix B1.

The stray light resulting from the sharp ghost reflection was considered as described in Eq. B2. For the synthetic frames, the two-dimensional fit for  $\mathbf{E}_{refl}$ , shown in Fig. B2, was expanded to the full frame shown in Fig. E1.

880 Stray light is causing a pseudo-noise in the retrieved column anomalies. However, there is also random noise, which is introduced by the shot noise  $N_{phot}$  of the measured photons and the read-out noise  $N_{ro}$  of the detector electronics, which had to be considered in the synthetic spectra, too. The shot noise is introduced by the intended signal photons as well as from the unwanted thermal photons in the background correction and was calculated by the signal in electrons  $S_{el}$  with  $N_{phot} = \sqrt{S_{el}}$ . The thermal signal was estimated from the background measurements, dependent on the exposure time. The slope of a first-order polynomial fit of a pixel value per exposure time was used as the background signal introduced by thermal photons. The pixel values for the thermal and the intended signal were converted with the fraction of the detector's full-well-capacity  $(0.34\,\mathrm{Me^-})$  and the corresponding bit-depth  $(16\,\mathrm{bit})$  to the signal in electrons. The total noise  $N_{full}$  is calculated by:

$$N_{full} = \sqrt{S_{el,intended} + S_{el,thermal} + N_{ro}^2}.$$
 (E2)

The synthetic frames were contaminated by a noise frame, containing a noise value for each frame pixel. The value for each pixel was a random normal distributed value with a standard deviation of the calculated noise converted into binary units. Two noise frames are generated, one for the non-stray-light-contaminated synthetic frame and one for the full-stray-light-contaminated synthetic frame.

The column anomalies were retrieved from the synthetic spectra as described in Sect. 4.1 for the real-measured spectra. In Fig. E3, the resulting CO2, CH4 and the proxy corrected (CH4/CO2) columns are compared. The overall column noise for all three columns of the real-measured and the simulated data is very similar. Differences are expected due to several factors, namely the not perfectly matched fitting of the low-order polynomial to adapt the simulated spectra to the measured spectra, see Fig. E2, pseudo-noise introduced by a more complex structure than a low-order polynomial, spectral surface reflectance, the unknown real out-of-field signal, and residual uncertainties in the measured stray light kernels. However, a Pearson correlation factor Pearson correlation coefficients of 0.80 in the for CO2 and 0.77 in the for CH4 column was were calculated in the direct comparisons. The correlation factor coefficient is close to zero after the proxy correction.

**E3 Spectral out-of-field Out-of-band stray light extrapolation**

885

895

900

In the stray light correction in Sect. 3.1, the measured spectra were extrapolated using a 3rd-order polynomial to scale an extended RTM. In reality, the signal beside the FPA, and therefore the OFSL and OBSL, is unknown. However, the simulated spectra provide the opportunity to apply the extrapolation with different orders (1st to 4th) of the polynomial. In Fig. E4, the standard deviations of the retrieved column anomalies are compared to the case where the spectral OFSL OBSL is fully known and with no considered OFSL, similar to Fig. 13. However, due to the presence of spatial OFSL, only data from the middle fiber is analyzed. The CH4-band is close to the border of the FPA, see Fig. 9. Therefore, it is expected that the spectral OFSL OBSL has the biggest impact on the CH4-band. The introduced pseudo noise is decreased by ~ 90 % to

Figure E3. Retrieved profile scaling factors (PSF) for the  $CO_2$  (a, b),  $CH_4$  (c, d) and the proxy corrected ( $CH_4/CO_2$ ) (e, f) columns for simulated and real-measured data. The simulated data is artificially contaminated with stray light and random noise. The left column shows histograms representing the column noise. The right column shows the correlation of the simulated and real-measured column anomalies. The red line shows a linear fit through throug the data, and the dashed black line marks a Pearson correlation coefficient (R) of 1.

which is usually not fully covered by a low-order polynomial for a wider spectral range. Further, the higher-order fits for extrapolation can highly under- or overestimate the signal level in the areas for extrapolation.

**Figure E4.** Single read-out column noise for retrieved column anomalies for the CO2 (blue triangle), CH4 (orange triangle) and the proxy corrected (CH4/CO2) (green-grey diamond) column, for different cases (y-axis). The first two cases are from Fig. 13 and only the middle fiber (number 14) is considered.

**915 Appendix F: Stable Kernel optimization**

920

The measured stable stray light in Fig. 6 in Sect. 3.1 shows contamination of the used laser, which is related to insufficient side-mode suppression. For the stable kernel creation, shown in Fig. B1, those areas were corrected. Therefore, the horizontal signal of the raw stable kernel was masked. Afterward, the stable kernel was defined in several sections, depending on the surface type in the entrance slit object plane. The stray light in the outer regions from spectral pixels 75 - 116 and 267 - 303 was reflected from a steel surface from the adjustable slits bladesuncoated adjustable slit aperture. Within the areas from spectral pixel 117 - 158 and 225 - 268, the light was reflected from the blade edges; due to the angle, the light was not reflected into the optical path of the useful signal. The area from the spectral pixel 159 - 224 was reflected from the aluminum ferrule and the aligned fibers. The stray light signal from the relay optics used for single fiber illumination can not be separated into instrumental and non-instrumental stray light, and therefore, the fiber area was set to zero.

925 The signal of each region was remapped into the polar coordinate space using "warp\_polar" function of the "skimage" Python package (version 0.18.1). The rows of the resulting 2D image represented the rotation angles, and the columns the radii. The signal in dependency of the radius for all angles is shown in Fig. F1. At this point, a generalized scattering theory, like the Harvey scatter model described by Peterson (2004), could be fitted. However, due to the signal steps in the aluminum and steel areas, the fitting did not describe the kernel sufficiently. Therefore, the median value along the rotation angle axis was used as a numerical function to describe the scattering. By rotating the function, the observational gaps were filled.

Figure F1. Signal of the scattering surfaces from the stable kernel in order of the radius after a polar coordinate transform.

**Appendix G: Parameters for WFM-DOAS and flux retrieval**

Table G1. Parameters for RTM simulation, WFM-DOAS and flux retrieval for landfill scene in Fig. 10 and 14.

| Date                       | 11.09.2022                                                    |
|----------------------------|---------------------------------------------------------------|
| Time                       | 15:30 - 16:00 UTC                                             |
| Wavelength                 | 1500 nm - 1750 nm                                             |
| Wavelength resolution      | $0.01\mathrm{nm}$                                             |
| Flight altitude            | 29000 feet                                                    |
| Background $\mathrm{CH}_4$ | 1906 ppb                                                      |
| Background $\mathrm{CO}_2$ | 413.7 ppm                                                     |
| Sun zenith angle           | 55.7°                                                         |
| Surface evaluation         | 172 m - 305 m                                                 |
| Albedo                     | 0.20                                                          |
| Aerosol scenario           | urban                                                         |
| Wind speed                 | $5.3\mathrm{ms^{-1}}$ (GoC, 2025)                             |
| Wind direction             | 209°                                                          |
| Fit window                 | <del>1575 nm</del> 1580.3 nm - <del>1677.5 nm</del> 1677.0 nm |
| Mean $c_f 	ext{ CH}_4$     | 0.80                                                          |
| Mean $c_f CO_2$            | 0.76                                                          |
| Spatial resolution         | $\sim 120 \times 120 \mathrm{m}^2$                            |
| Plume area                 | 1.5 km from source                                            |
| Background area            | 2 km from plume area                                          |
|                            |                                                               |

Table G2. Parameters for RTM simulation and WFM-DOAS for oil sand scene in Fig. 8.

| Wavelength                 | 1555 nm - 1730 nm                                             |
|----------------------------|---------------------------------------------------------------|
| Wavelength resolution      | $0.01\mathrm{nm}$                                             |
| Flight altitude            | 26000 feet                                                    |
| Background CH 4 | 1894 ppb                                                      |
| Background $\mathrm{CO}_2$ | 411.7 ppm                                                     |
| Sun zenith angle           | 46.3°                                                         |
| Surface evaluation         | $185 - 839 \mathrm{m}$                                        |
| Albedo                     | 0.20                                                          |
| Aerosol scenario           | urban                                                         |
| Fit window                 | <del>1575 nm</del> 1580.3 nm - <del>1677.5 nm</del> 1677.0 nm |
| Mean $c_f 	ext{ CH}_4$     | 0.78                                                          |
| Mean $c_f CO_2$            | 0.75                                                          |
| Spatial resolution         | $\sim 120 \times 120 \mathrm{m}^2$                            |

**Appendix H: Entrance slit Slit aperture exchange**

During the CoMet 2.0 mission, the adjustable slit uncoated adjustable slit aperture shown in Fig. H1 (a) with uncoated blades 3 was installed in the spectrometer in front of the entrance fiber ferrule, in Fig. 1. The variable slit uncoated adjustable slit aperture was exchanged with a fixed 200 µm slit (THORLABS S200ULK) consisting of blackened stainless steel, shown in Fig. H1(b) and (c). The slit. The blackened fixed slit aperture was wider than the fiber and is therefore acting as an additional aperture, as blackening the ferrule was not feasible. The slit-blackened fixed slit aperture was glued on an anodized aluminum support. The side that shows in the direction of the optics is painted with NEXTEL Velvet Coating 811-21.

**940 Appendix I: Characterization measurements**

935

945

In order to retrieve trace gas column enhancements from the measured spectra, it is necessary to have a very good characterization of the instrument. The wavelength calibration and the instrumental spectral response function (ISRF) were measured with a Littman/Metcalf laser system (Lion System, by Sacher Germany, (Stry et al., 2006)), with a tunable wavelength from  $1600 \, \mathrm{nm} - 1750 \, \mathrm{nm}$  at a precision of  $0.05 \, \mathrm{nm}$  and a power of  $\sim 20 \, \mathrm{mW}$ . The actual wavelength of the laser was observed with a laser wavelength meter (671A, by Bristol) with an accuracy of  $\pm 0.2 \, \mathrm{pm}$  at  $1000 \, \mathrm{nm}$  for the NIR range from  $520 \, \mathrm{nm} - 1700 \, \mathrm{nm}$ . Flat field-Flat-field corrections were applied to account for PRNU errorsand losses from the grating. These corrections were performed using a broadband Quartz Tungsten Halogen lamp as a WLS. To achieve a homogeneous illumination, all sources were connected to an input port of an integrating sphere, with an inner diameter of  $5.3^{\circ}$  (Ophir, IS6-C, by Ophir).

Figure H1. (a) adjustable Blackened fixed 200 µm slit, which was assembled during aperture as replacement for the CoMet 2.0 mission at the entrance fiber ferrule (ferrule 2, uncoated adjustable slit aperture in Fig. 1) of the spectrometer3. (ba) side of the blackened fixed 200 µm slit as replacement for (a) aperture showing in direction of the ferrule. (eb) side of blackened fixed slit, that shows aperture showing in the direction of the ferrule collimator lens.

[revised manuscript text omitted]

Techniques, Environmental Science and Technology Letters, ISSN 2328-8930, https://doi.org/10.1021/acs.estlett.4c01063, 2025.

- Chan Miller, C., Roche, S., Wilzewski, J. S., Liu, X., Chance, K., Souri, A. H., Conway, E., Luo, B., Samra, J., Hawthorne, J., Sun, K., Staebell, C., Chulakadabba, A., Sargent, M., Benmergui, J. S., Franklin, J. E., Daube, B. C., Li, Y., Laughner, J. L., Baier, B. C., Gautam, R., Omara, M., and Wofsy, S. C.: Methane retrieval from MethaneAIR using the CO2 proxy approach: a demonstration for the upcoming MethaneSAT mission, Atmospheric Measurement Techniques, 17, 5429–5454, ISSN 1867-8548, https://doi.org/10.5194/amt-17-5429-2024, 2024.
- Chapman, J. W., Thompson, D. R., Helmlinger, M. C., Bue, B. D., Green, R. O., Eastwood, M. L., Geier, S., Olson-Duvall, W., and Lundeen, S. R.: Spectral and Radiometric Calibration of the Next Generation Airborne Visible Infrared Spectrometer (AVIRIS-NG), Remote Sensing, 11, 2129, ISSN 2072-4292, https://doi.org/10.3390/rs11182129, 2019.
- 1000 Clermont, L., C.Michel, Chouffart, Q., and Zhao, Y.: Going beyond hardware limitations with advanced stray light calibration for the Metop-3MI space instrument, Scientific Reports, 14, ISSN 2045-2322, https://doi.org/10.1038/s41598-024-68802-z, 2024.
  - Cogan, A. J., Boesch, H., Parker, R. J., Feng, L., Palmer, P. I., Blavier, J. L., Deutscher, N. M., Macatangay, R., Notholt, J., Roehl, C., Warneke, T., and Wunch, D.: Atmospheric carbon dioxide retrieved from the Greenhouse gases Observing SATellite (GOSAT): Comparison with ground-based TCCON observations and GEOS-Chem model calculations, Journal of Geophysical Research: Atmospheres, 117, ISSN 0148-0227, https://doi.org/10.1029/2012id018087, 2012.
  - Dussaux, A., Bazalgette Courrèges-Lacoste, G., Gaucel, J.-M., Garnier, T., Gaudin-Delrieu, ., Fayret, J.-P., Ouslimani, H., Charnier, J.-Y., Delclaud, Y., Lesschaeve, S., Spilling, D., te Hennepe, F., Förster, U., Strauss, S., Huber, G., Komadina, J., Reijnset, R., Pachot, C., Durand, Y., Pasquet, A., Chanumolu, A., Martinez Fernandez, M., Caleno, M., Meijer, Y., and Fernandez, V.: Copernicus CO2M: the mission for monitoring anthropogenic carbon dioxide from space: status of the payload at the start of the AIT phase, in: International Conference on Space Optics ICSO 2024, edited by Bernard, F., Karafolas, N., Kubik, P., and Minoglou, K., p. 38, SPIE, https://doi.org/10.1117/12.3072498, 2025.
  - Fest, E.: Stray Light Analysis and Control, SPIE, ISBN 9780819493262, https://doi.org/10.1117/3.1000980, 2013.
  - Gerilowski, K., Tretner, A., Krings, T., Buchwitz, M., Bertagnolio, P. P., Belemezov, F., Erzinger, J., Burrows, J. P., and Bovensmann, H.: MAMAP a new spectrometer system for column-averaged methane and carbon dioxide observations from aircraft: instrument description

[revised manuscript text omitted]

- Rozanov, V., Rozanov, A., Kokhanovsky, A., and Burrows, J.: Radiative transfer through terrestrial atmosphere and ocean: Software package SCIATRAN, Journal of Quantitative Spectroscopy and Radiative Transfer, 133, 13–71, ISSN 0022-4073, https://doi.org/10.1016/j.jqsrt.2013.07.004, 2014.
- Sierk, B., Fernandez, V., Bézy, J.-L., Meijer, Y., Durand, Y., Bazalgette Courrèges-Lacoste, G., Pachot, C., Löscher, A., Nett, H., Minoglou, K., Boucher, L., Windpassinger, R., Pasquet, A., Serre, D., and te Hennepe, F.: The Copernicus CO2M mission for monitoring anthropogenic carbon dioxide emissions from space, in: International Conference on Space Optics ICSO 2020, edited by Sodnik, Z., Cugny, B., and Karafolas, N., p. 128, SPIE, https://doi.org/10.1117/12.2599613, 2021.
- 1075 Staebell, C., Sun, K., Samra, J., Franklin, J., Chan Miller, C., Liu, X., Conway, E., Chance, K., Milligan, S., and Wofsy, S.: Spectral calibration of the MethaneAIR instrument, Atmospheric Measurement Techniques, 14, 3737–3753, ISSN 1867-8548, https://doi.org/10.5194/amt-14-3737-2021, 2021.
  - Stry, S., Thelen, S., Sacher, J., Halmer, D., Hering, P., and Mürtz, M.: Widely tunable diffraction limited 1000 mW external cavity diode laser in Littman/Metcalf configuration for cavity ring-down spectroscopy, Applied Physics B, 85, 365–374, ISSN 1432-0649, https://doi.org/10.1007/s00340-006-2348-1, 2006.
  - Tanimoto, H., Matsunaga, T., Someya, Y., Fujinawa, T., Ohyama, H., Morino, I., Yashiro, H., Sugita, T., Inomata, S., Müller, A., Saeki, T., Yoshida, Y., Niwa, Y., Saito, M., Noda, H., Yamashita, Y., Ikeda, K., Saigusa, N., Machida, T., Frey, M. M., Lim, H., Srivastava, P., Jin, Y., Shimizu, A., Nishizawa, T., Kanaya, Y., Sekiya, T., Patra, P., Takigawa, M., Bisht, J., Kasai, Y., and Sato, T. O.: The greenhouse gas observation mission with Global Observing SATellite for Greenhouse gases and Water cycle (GOSAT-GW): objectives, conceptual framework and scientific contributions, Progress in Earth and Planetary Science, 12, ISSN 2197-4284, https://doi.org/10.1186/s40645-025-00684-9, 2025.
  - Tol, P. J., van Kempen, T. A., van Hees, R. M., Krijger, M., Cadot, S., Snel, R., Persijn, S. T., Aben, I., and Hoogeveen, R. W. M.: Characterization and correction of stray light in TROPOMI-SWIR, Atmospheric Measurement Techniques, 11, 4493–4507, ISSN 1867-8548, https://doi.org/10.5194/amt-11-4493-2018, 2018.

- van Hees, R. M., Tol, P. J. J., Cadot, S., Krijger, M., Persijn, S. T., van Kempen, T. A., Snel, R., Aben, I., and Hoogeveen, R. M.: Determination of the TROPOMI-SWIR instrument spectral response function, Atmospheric Measurement Techniques, 11, 3917–3933, ISSN 1867-8548, https://doi.org/10.5194/amt-11-3917-2018, 2018.
- Veefkind, J., Aben, I., McMullan, K., Förster, H., de Vries, J., Otter, G., Claas, J., Eskes, H., de Haan, J., Kleipool, Q., van Weele, M., Hasekamp, O., Hoogeveen, R., Landgraf, J., Snel, R., Tol, P., Ingmann, P., Voors, R., Kruizinga, B., Vink, R., Visser, H.,
   and Levelt, P.: TROPOMI on the ESA Sentinel-5 Precursor: A GMES mission for global observations of the atmospheric composition for climate, air quality and ozone layer applications, Remote Sensing of Environment, 120, 70–83, ISSN 0034-4257, https://doi.org/10.1016/j.rse.2011.09.027, 2012.
  - Zong, Y., Brown, S. W., Johnson, B. C., Lykke, K. R., and Ohno, Y.: Simple spectral stray light correction method for array spectroradiometers, Applied Optics, 45, 1111, ISSN 1539-4522, https://doi.org/10.1364/ao.45.001111, 2006.
- 1100 Zong, Y., Brown, S. W., Meister, G., Barnes, R. A., and Lykke, K. R.: Characterization and correction of stray light in optical instruments, in: Sensors, Systems, and Next-Generation Satellites XI, edited by Meynart, R., Neeck, S. P., Shimoda, H., and Habib, S., vol. 6744, p. 67441L, SPIE, ISSN 0277-786X, https://doi.org/10.1117/12.737315, 2007.